# De novo analysis of bulk RNA-seq data at spatially resolved single-cell resolution

Jie Liao [1,2,10], Jingyang Qian[1,10], Yin Fang [3,4,10], Zhuo Chen [3,4,10], Xiang Zhuang[3,4,10], Ningyu Zhang[3,4], Xin Shao [1,2], Yining Hu[1], Penghui Yang[1], Junyun Cheng[1,5], Yang Hu[1,5], Lingqi Yu[1], Haihong Yang[3,4], Jinlu Zhang[1,3], Xiaoyan Lu[1,5], Li Shao[6], Dan Wu [7], Yue Gao [8] ✉, Huajun Chen [3,4] ✉ & Xiaohui Fan [1,2,5,9] ✉

Uncovering the tissue molecular architecture at single-cell resolution could help better understand organisms' biological and pathological processes. However, bulk RNA-seq can only measure gene expression in cell mixtures, without revealing the transcriptional heterogeneity and spatial patterns of single cells. Herein, we introduce Bulk2Space (https://github.com/ZJUFanLab/bulk2space), a deep learning framework-based spatial deconvolution algorithm that can simultaneously disclose the spatial and cellular heterogeneity of bulk RNA-seq data using existing single-cell and spatial transcriptomics references. The use of bulk transcriptomics to validate Bulk2Space unveils, in particular, the spatial variance of immune cells in different tumor regions, the molecular and spatial heterogeneity of tissues during inflammation-induced tumorigenesis, and spatial patterns of novel genes in different cell types. Moreover, Bulk2Space is utilized to perform spatial deconvolution analysis on bulk transcriptome data from two different mouse brain regions derived from our in-house developed sequencing approach termed Spatial-seq. We have not only reconstructed the hierarchical structure of the mouse isocortex but also further annotated cell types that were not identified by original methods in the mouse hypothalamus.

Tissue complexity is portrayed by the spatial diversity and heterogeneity of cells[1]. Advances in spatially resolved transcriptomics[2–4] have made it possible to understand the cell composition, molecular architecture, and functional details of tissues at unanticipated spatial levels[5,6]. State-of-the-art experimental technologies, including image-based methods[7–10], spatial barcoding RNA-seq methods[11–14], and laser capture microdissection-based methods[15,16], have been developed to address either high throughput measuring of cells, unbiased detection of mRNA species, or single-cell resolution. To investigate the molecular variation during biological and pathological processes at a higher resolution, each sample is encouraged to be analyzed by spatially resolved single-cell transcriptomics, which is not yet fully achieved and

[1]College of Pharmaceutical Sciences, Zhejiang University, 310058 Hangzhou, China. [2]Future Health Laboratory, Innovation Center of Yangtze River Delta, Zhejiang University, 314100 Jiaxing, China. [3]College of Computer Science and Technology, Zhejiang University, 310027 Hangzhou, China. [4]Hangzhou Innovation Center, Zhejiang University, 310058 Hangzhou, China. [5]Innovation Center in Zhejiang University, State Key Laboratory of Component-Based Chinese Medicine, 310058 Hangzhou, China. [6]Institute of Translational Medicine, The Affiliated Hospital of Hangzhou Normal University, Hangzhou Normal University, 310015 Hangzhou, China. [7]College of Biomedical Engineering and Instrument Science, Zhejiang University, 310013 Hangzhou, China. [8]Department of Pharmaceutical Sciences, Beijing Institute of Radiation Medicine, 100850 Beijing, China. [9]Westlake Laboratory of Life Sciences and Biomedicine, 310024 Hangzhou, China. [10]These authors contributed equally: Jie Liao, Jingyang Qian, Yin Fang, Zhuo Chen, Xiang Zhuang. ✉e-mail: gaoyue@bmi.ac.cn; huajunsir@zju.edu.cn; fanxh@zju.edu.cn

is time-consuming, costly, and difficult to scale up[17]. Meanwhile, with investment for almost two decades, RNA-seq has been extensively applied in transcriptome analysis[18], with many large projects having been carried out, such as the Encyclopedia of DNA Elements (ENCODE)[19], The Cancer Genome Atlas (TCGA)[20], and projects of the International Cancer Genome Consortium (ICGC)[21]. A wealth of bulk RNA-seq data has become a legacy for biological and clinical research[22]. Thus, reanalysis of the enormous amount of bulk data to explain both cellular diversity and spatial expression patterns is a challenging but consequential task.

In silico methods have great potential to predict spatial heterogeneity from bulk RNA-seq data at single-cell resolution by integrating cutting-edge technologies. Several approaches such as CPM[23], CIBERSORT[24], and MuSiC[25] can only extrapolate proportions of cell types from bulk RNA-seq data and have failed to further generate single-cell data, let alone map them to tissue coordinates. The emergence of an approach that can efficiently decompose bulk RNA-seq data into spatially resolved single-cell expression profiles is expected to reveal the cell diversity of complex tissue and the spatial expression variation simultaneously.

Herein, we introduce Bulk2Space, a spatial deconvolution algorithm based on deep learning frameworks, which generates spatially resolved single-cell expression profiles from bulk transcriptomes using existing high-quality scRNA-seq data and spatial transcriptomics as references. We hypothesize that the process of scRNA-seq is similar to sampling cells from bulk tissue, and each selected cell is labeled with a unique barcode. Consequently, bulk transcriptome data can be used as a weighted collection of single-cell expression data in a defined clustering space of cells. Bulk2Space first generates single-cell transcriptomic data within the clustering space to find a set of cells whose aggregated data is proximate to the bulk data. Next, the generated single cells were allocated to optimal spatial locations using a spatial transcriptome reference. For this step, we chose as a spatial reference one of the two most commonly used spatially resolved transcriptomics technologies. One is an image-based method with limited target genes[7–9], and the other is a spatial barcoding method without single-cell resolution[11–14]. Taken all, Bulk2Space showed a robust performance across multiple datasets and conditions and is an open-access algorithm on GitHub (https://github.com/ZJUFanLab/bulk2space).

## Results
### Design concept of Bulk2Space

The overall design of the Bulk2Space algorithm is illustrated in Fig. 1, which is divided into two steps, deconvolution, and spatial mapping. Although bulk RNA-seq data are obtained by sequencing a mixture of cells, whereas scRNA-seq labels individual cells in advance, both methods share comparable cell types and states[23–26]. Thus, we assume that bulk transcriptomics data can be decomposed into single-cell transcriptomics data by a well-designed deconvolution algorithm (Step 1). Subsequently, in the second step, the above-mentioned two prevailing spatially resolved transcriptomics methods, though, failed in either achieving single-cell resolution or delineating the whole transcriptome, can provide reference locations for single cells generated in the first step of Bulk2Space based on the similarity of their

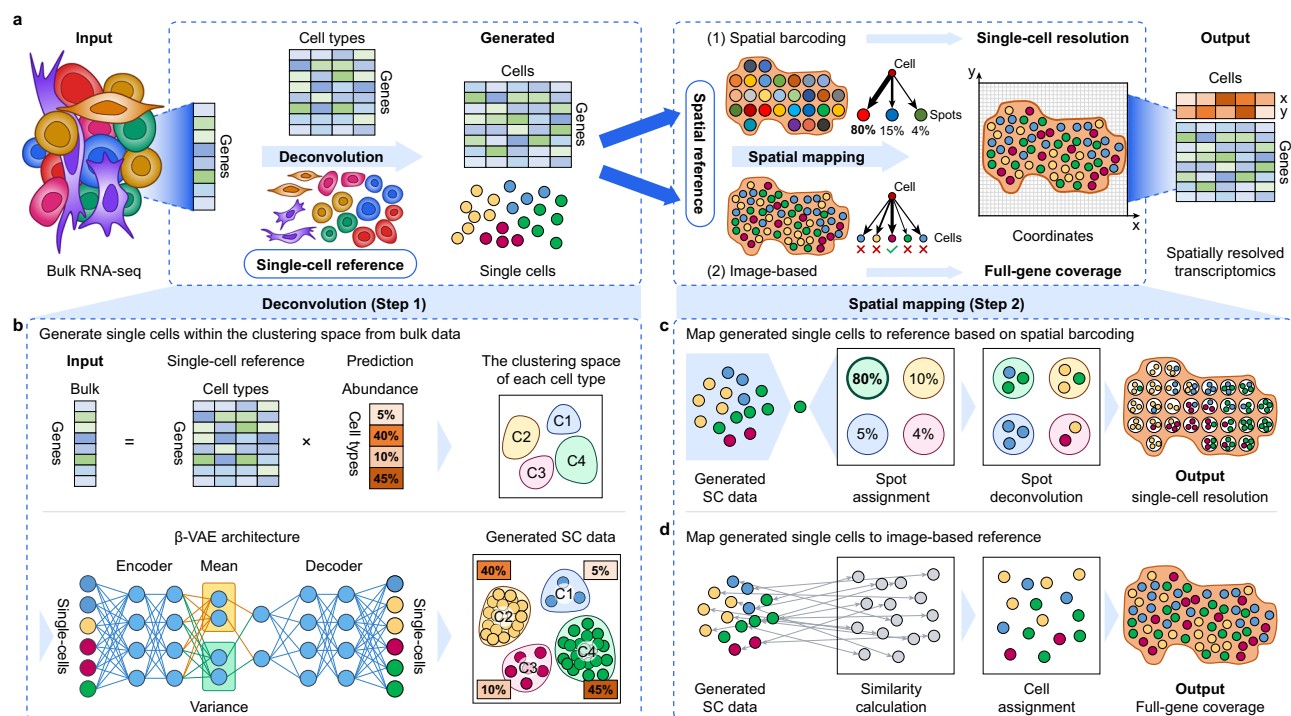

**Fig. 1 | Workflow of Bulk2Space. a** Overview of the design concept of Bulk2Space. Bulk transcriptome data is taken as the input, and a single-cell profile is used as the reference for characterizing the clustering space of the heterogeneous tissue. After deconvolution, input bulk data is deconvolved into single-cell transcriptomics data. Then, either of the two spatially resolved transcriptomics, spatial barcoding-based RNA-seq or image-based in-situ hybridization, is used as the spatial reference. Generated single cells are assigned to the corresponding coordinates based on the spatial reference. The output is a set of generated single-cell profiles with specified x and y spatial coordinates. **b** Detailed deconvolution procedure. The input vector of bulk tissue is equal to the production of the expression matrix of cell types and the proportion vector of each cell type. The calculated proportion of all cell types is employed for the subsequent single-cell generation. The single-cell reference is used to characterize the clustering space of the tissue, and a deep learning model generates single-cell profiles within the clustering space of each cell type. **c, d** demonstrate the strategies for spatial mapping based on the two mainly used spatially resolved transcriptomics approaches. **c** For spatial barcoding-based reference, each generated single cell is assigned to the spot with the highest gene expression correlation until the aggregation of cells within the spot is close enough to the exact expression value. **d** For image-based reference, each generated cell is assigned to the location where the cell on tissue has the highest similarity with the given cell.

expression profiles (Step 2)[27,28]. As a consequence, it could be rationally conceived that the combination of single-cell profiles and spatial transcriptomics is expected to overcome technical bottlenecks and accomplish the spatially resolved single-cell deconvolution of bulk transcriptomes (Fig. 1a).

For the deconvolution of bulk transcriptomics data, we hypothesize that the process of single-cell sequencing identifies with the process of sampling from the tissue, while the ratio between different cell types is related to the zone of the tissue, the ways of single-cell capturing, and the size of the cell itself. However, even if the ratio between cell types changes, the state of each cell type still fluctuates within a relatively stable high-dimensional space, which is namely the clustering space of cell types. This hypothesis has been supported by several studies[29–31]. As shown in Fig. 1b, a single-cell reference can be utilized to characterize the clustering space and average the gene expression for each cell type[31,32]. The expression vector of the bulk transcriptome is taken as the input, which is equal to a product of the average gene expression matrix of cell types and their abundance vector. By solving this nonlinear equation, the proportion of each cell type is determined. Then, the solved proportion of each cell type is taken as a control parameter to generate the corresponding number of single cells. Specifically, a deep generative model, termed beta variational autoencoder (β-VAE)[33], is employed to simulate a given number of single cells within the characterized clustering space of each cell type. The simulation stops when the training loss no longer reduces, thus deconvolves the heterogeneous bulk transcriptome into single-cell transcriptome data. Instead of only calculating the abundance values of cell types, as performed by other methods, such as Cell Population Mapping (CPM)[23], CIBERSORT[24], and ImmuCC[26], Bulk2Space can generate biologically feasible single-cell expression profiles.

For spatial mapping, the single-cell profiles generated in the first step are assigned to the reference tissue section based on an optimization strategy. In the second step, we consider two of the most commonly used spatially resolved transcriptomics approaches as the spatial reference, namely, spatial barcoding-based RNA-seq methods and image-based targeted methods.

Spatial barcoding-based methods employ arrayed reverse transcription primers with unique positional barcodes to preserve the spatial information of mRNA via in situ complementary DNA (cDNA) synthesis. Such methods could provide the whole transcriptome as well as the location of each sequenced spot after decoding the spatial barcodes. However, these methods are unable to achieve single-cell resolution, because the spots are customized in shape and size, such as ST[14] (100 μm in diameter), Visium (55 μm in diameter), Slide-seq[13] (10 μm in diameter), etc. As shown in Fig. 1c, for spatial barcoding-based references, each barcoded spot is regarded as a mixture of several cells. Similar to existing decomposition methods such as RCTD[34], SpatialDWLS[35], stereoscope[36], and SPOTlight[37], we first calculate the cell-type composition of each spot. The difference is that we then map the generated single cells into the recommended spots based on the similarity of their cell expression profiles and ensure that the proportions of cell types in each spot are consistent with the calculated results. Since Bulk2Space generates spatially resolved single-cell transcriptomics data from bulk RNA-seq, the spatial heterogeneity of individual cells can be analyzed, which cannot be realized by RCTD, SpatialDWLS, stereoscope, and SPOTlight.

As illustrated in Fig. 1d, another strategy is to map the single-cell data generated in the first step onto image-based targeted references capable of measuring hundreds to thousands of RNA species[38], such as MERFISH[7], STARmap[9], seqFISH[10], etc. The pairwise similarity between cells is calculated based on genes shared by both datasets, and eventually, each generated single cell is robustly mapped to an optimized coordinate of the target cell in the spatial reference. The resulting spatially resolved single-cell RNA-seq data can provide unbiased transcriptomes of individual cells and improve the gene coverage.

Taken all, unlike other computational methods that focus only on either of the two spatially resolved transcriptomics approaches, Bulk2Space takes into account both techniques to greatly increase the scalability, applicability, and reference data of the algorithm.

## Performance evaluation of Bulk2Space using simulated and biological datasets

To demonstrate the robustness of the deconvolution step of Bulk2Space, a benchmark test was performed using 30 paired simulations (both bulk and single-cell data were generated from the same datasets) from 10 different high-quality single-cell datasets across human blood, brain, kidney, liver, and lung, and mice brain, kidney, lung, pancreas, and testis (Fig. S1a), and 12 unpaired simulations (both bulk and single-cell data were generated from different datasets of mice pancreas) from 8 single-cell RNA-seq data of human pancreas (Fig. S1b). All datasets used in this study were listed in Supplementary Data 1.

First, for 30 paired simulations, each single-cell transcriptome dataset was divided into two parts, one for reference and the other for the synthesis of bulk data. The procedure for bulk data synthesis is shown in Fig. S1c. For each single-cell dataset, we changed the proportion of cell types and synthesized 3 corresponding bulk transcriptomics data with different cell compositions. Three generative deep learning models, namely, the beta variational autoencoder (β-VAE)[33], generative adversarial networks (GAN)[39], and conditional GAN (CGAN)[40], were introduced here for the benchmarking of different methods using 30 paired single-cell datasets. The gene expression correlation between the generated single-cell data and input bulk data was calculated to evaluate and compare the three candidate algorithms. As illustrated in Fig. 2a, β-VAE performed better than the other algorithms in both Pearson and Spearman correlations with a lower root mean squared error (RMSE). The Pearson correlation, Spearman correlation, and RMSE were calculated by combining the gene expression across all cell types and averaging these metrics for each cell type. Ten examples of different species and tissues are shown in Fig. S2. The results demonstrated that the clustering space of the generated single-cell data was similar to that of the test data but varied in the proportions of different cell types (Fig. S2a and Fig. S2b). Furthermore, the correlation analysis of the expression of marker genes for each cell type between the generated and test data was performed for each method. The comparison of the correlation heatmaps between Bulk2Space, GAN, and CGAN showed that Bulk2Space had higher correlations across different simulations (Fig. S2c).

We then benchmarked the deconvolution step of Bulk2Space using 12 unpaired simulation data of mice pancreas. Because other deconvolution methods, such as CPM[23], CIBERSORT[24], and ImmuCC[26], can only predict cell-type proportions instead of gene expression of generated data, we compared Bulk2Space with GAN, CGAN, and a Bayesian deconvolution method termed bMIND[41]. As illustrated in Fig. 2b and Fig. S3a, Bulk2Space outperformed the other three methods with a higher Pearson correlation of gene expression and lower gene expression variation (RMSE). Although CGAN had a comparable performance as Bulk2Space in the single-cell generation, its computing speed was significantly lower than Bulk2Space. The detailed comparison of the cell-type-specific marker gene expression correlations of generated single-cell data and the ground truth between the four candidate methods were shown in Fig. S3b.

To further investigate the robustness of the single-cell generation of Bulk2Space, we introduced two noise mechanisms to test the performance of the algorithm (Fig. S4). One mechanism changes the expression values of certain genes in randomly selected cells (Fig. S4a), and the other alters the cell type labels of selected cells (Fig. S4b). Bulk2space showed robust performance and could effectively avoid the over-fitting phenomenon (Fig. S4c and Fig. S4d). We also demonstrated that Bulk2Space can deconvolve bulk RNA-seq data with an annotation-free single-cell reference in the supplementary information

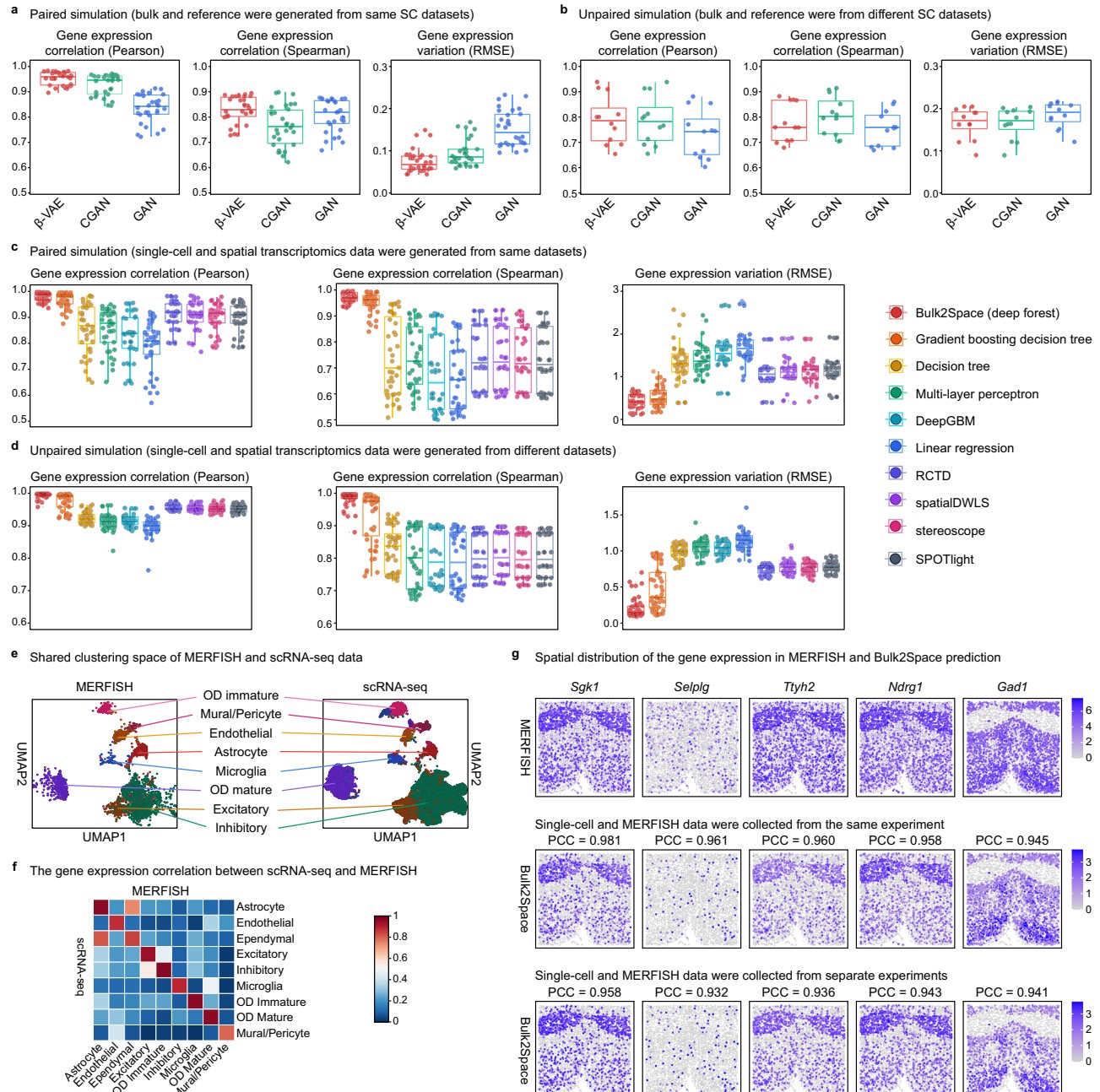

**Fig. 2 | Benchmark test for Bulk2Space. a** Gene expression correlation (Pearson correlations, left, and Spearman correlations, middle) and gene expression variation (RMSE, right) of three methods, namely, Bulk2Space (β-VAE), GAN, and CGAN, using paired simulation data (*n* = 30). Each point represented a simulated dataset. Data are presented as boxplots (minima, 25th percentile, median, 75th percentile, and maxima). **b** Gene expression correlation and gene expression variation of three methods with unpaired simulation data (*n* = 12). Data are presented as boxplots (minima, 25th percentile, median, 75th percentile, and maxima). **c** Benchmark test for 10 spatial mapping methods (from left to right, Bulk2Space (deep forest), gradient boosting decision tree, decision tree, multi-layer perceptron, DeepGBM, linear regression, RCTD, spatialDWLS, stereoscope, and SPOTlight) using paired simulation data (*n* = 50). Data are presented as boxplots (minima, 25th percentile, median,

75th percentile, and maxima). Left, Pearson correlation of gene expression, Middle, Spearman correlation of gene expression, Right, gene expression variation (RMSE). **d** Benchmark test for 10 spatial mapping methods using paired simulation data (*n* = 60). Data are presented as boxplots (minima, 25th percentile, median, 75th percentile, and maxima). **e** The UMAP layout of the MERFISH data (image-based spatially resolved transcriptomics) and the single-cell data. Share cell types were colored with corresponding colors. **f** Expression correlation heatmap of the marker genes of all shared cell types in MERFISH data and scRNA-seq data. **g** Spatial expression of genes in the ground truth and predicted by Bulk2Space using paired and unpaired single-cell RNA-seq data. PCC for each gene was shown. Source data are provided as a Source Data file.

Fig. S5. Since the core generation model of Bulk2Space is β-VAE, a perturbation analysis was conducted to verify the robustness of single-cell generation by Bulk2Space. The perturbation test confirmed that better generation results can be obtained by assuming that the latent vectors follow the Gaussian distribution (Fig. S6). Finally, we validated Bulk2Space with paired biological bulk and single-cell data derived

from different mice liver[42]. As shown in Fig. S7, the deconvolution results indicated that Bulk2Space could be well applied in biological scenarios.

In the second step, Bulk2Space assigns the generated single cells to spatial coordinates based on a spatially resolved transcriptome reference (Fig. 1c, d). There are two types of most common spatially

resolved transcriptomic techniques. One is the spatial barcoding-based RNA-seq technology, which includes 'spatial transcriptomics' (ST)[14], high-definition spatial transcriptomics (HDST)[11], and Slide-seq[12,13], 10 Visium, etc. These methods are based on tissue micro-regions with certain regular shapes (spots), and cannot achieve single-cell resolution. The other is image-based in situ transcriptomics, including MERFISH[7], seqFISH[8,10], and STARmap[9]. These methods can only measure the expression of target genes but cannot cover the whole transcriptome. To overcome the bottlenecks of the two approaches, we designed corresponding spatial assignment strategies.

The first strategy was designed to map single cells to spots in spatially resolved transcriptomics based on spatial barcoding. In this step, we first used paired synthetic data to test the performance of Bulk2Space. The data synthesis procedure is illustrated in Fig. S1c. Similar to the commonly used spatially resolved transcriptomics technologies, such as ST, HDST, 10X Visium, and Slide-seq, we randomly chose 10 cells from each scRNA-seq data and aggregated their gene expression profiles as a spot of pseudo spatial transcriptomics data. A comprehensive comparison between Bulk2Space (deep forest), six machine learning or deep learning approaches, and four published methods, RCTD[34], SpatialDWLS[35], stereoscope[36], and SPOTlight[37] was then conducted using 50 paired simulation datasets. These methods were benchmarked for the optimization of cell-type composition and spatial gene expression patterns. As shown in Fig. 2c, compared with these methods, Bulk2Space showed higher correlations in gene expression and lower RMSEs than other methods. The cell composition and spatial cell-type distribution predicted by Bulk2Space were highly correlated to the ground truth (Fig. S8). Next, we benchmarked the spatial mapping step of Bulk2Space using unpaired simulation data of mice pancreas. As shown in Fig. 2d, Bulk2Space showed the highest correlations in gene expression and the lowest RMSEs. The comparison of the cell-type composition for each spot between Bulk2Space and the spatial reference was shown in Fig. S9.

The second spatial mapping strategy uses image-based targeted methods. A scRNA-seq data ("GSE113576") and MERFISH data (image-based reference) of the mouse hypothalamus tissue from the same experiment[43] were used as paired datasets to test the performance of the Bulk2Space algorithm. Another scRNA-seq data[44] derived from a separated experiment was used as unpaired single-cell data. We wanted to find the shortest path that could shift single cells to optimal locations. First, a nonparametric empirical Bayes network was used to eliminate the expression differences between the single-cell data and spatially resolved transcriptome data. As illustrated in Fig. 2e, the clustering spaces of the two datasets were very similar, with most cell types sharing the distribution in both datasets. The two discrete datasets could be well merged in distribution, and the gene expression of each cell type had a correlation of over 0.8 between the two datasets (Fig. 2f). Of the 150 target genes identified in MERFISH, a leave-one-out test was conducted for the evaluation of the spatial mapping step of Bulk2Space. Specifically, 149 were used as reference genes, and the rest one was used for validation. The results showed that spatially, the gene expression pattern of the matched cells was highly correlated with that of the reference, and the predicted spatial expression patterns across all 150 targeted genes are strongly correlated with the ground truth with averaged Pearson correlation coefficients (PCCs) over 0.9 for both paired and unpaired datasets (Fig. 2g and Fig. S10).

We further verified Bulk2Space's spatial mapping using five-fold cross-validation for image-based methods (Fig. S11). specifically, the 150 targeted genes in MERFISH were split into five folds using R package caret (version 6.0–92), and 80% of genes were used as the reference and the remaining was used for validation of predicted spatial expression. As shown in Fig. S11, the spatial expressions of the top 25 genes predicted by Bulk2Space for the paired and unpaired single-cell data were compared with MERFISH data.

To validate the spatial mapping step of Bulk2Space in the real situation, two biological datasets, a spatial reference sequenced by Slide-seq v2[12], and a single-cell data[45] from separated experiments were used to reconstruct the structure of the mouse hippocampus (Fig. S12). The spatial distribution of single cells predicted by Bulk2Space was consistent with the real pattern of cell types in that region. Notably, the refined subregions in the mouse hippocampus were successfully reconstructed by Bulk2Space, which was confirmed by the spatial expression of cell-type-specific marker genes (Fig. S12e).

Since Bulk2Space employed β-VAE to generate single-cell profiles within the clustering space of cell types, we then investigated whether the randomness of β-VAE could affect the generation of single-cell data and the spatial mapping results. We conducted 100 repetitions on the deconvolution step of Bulk2Space to evaluate the robustness of β-VAE using simulated bulk and single-cell ref. [46]. The results demonstrated that the single-cell generation remained highly robust across 100 repetitions (Fig. S13).

We then used simulated PDAC[47] and melanoma[48,49] bulk data to evaluate the robustness of the spatial mapping step of Bulk2Space using three repetitions of bulk data deconvolution and spatial mapping. As shown in Fig. S14 and Fig. S15, the spatial deconvolution results suggested a robust performance of Bulk2Space with 3 repetitions for PDAC and melanoma bulk data, respectively.

In conclusion, although the single-cell data generated by β-VAE were slightly different each time, the overall prediction results showed robust performance in the spatial distribution of cell types, the cell-type composition and proportion in spots, and the spatial patterns of gene expression.

## Validation of Bulk2Space using biological data

To further verify the performance of the Bulk2Space, two consecutive slices[49] from the same melanoma tissue were used to demonstrate the spatial deconvolution result. As shown in Fig. S16, one slice was used as the spatial reference and the other was used to synthesize the bulk data. The expression of cell-type-specific marker genes between the generated data and the reference were highly correlated and the spatial distribution pattern of generated single cells was consistent with the histological annotations of different regions. Moreover, we found a spatial heterogeneity of generated B cells from different tissue regions, which was associated with the biological functions in the lymph node area and the melanoma region. Because Bulk2Space can predict spatially resolved single-cell transcriptomics data from bulk RNA-seq or scRNA-seq, spatial heterogeneity of the same cell type can be discovered. This was difficult for other spatial deconvolution algorithms to achieve such as RCTD[34], SpatialDWLS[35], stereoscope[36], and SPOTlight[37].

Similarly, the same spatial deconvolution procedure of Bulk2-Space was performed for another two discrete slices from different PDAC[47] tissues to reveal the spatial heterogeneity of bulk transcriptomics data. One slice was used as the spatial reference and the other was used to synthesize the bulk data. As shown in Fig. S17, similar results were retrieved as the validation of melanoma data. The expression of cell-type-specific marker genes between the generated data and the reference were highly correlated and the spatial distribution pattern of generated single cells was consistent with the histological feature of tissue regions.

## Bulk2Space integrates spatial gene expression and histomorphology in PDAC

Based on the success in the spatial deconvolution of PDAC using simulated data and discrete slices, we further verified the performance of the Bulk2Space algorithm using biological bulk RNA-seq data. As shown in Fig. 3a, two bulk RNA-seq data[50] of the pancreatic adjacent (PA) and the PDAC tissues, one scRNA-seq data[47], and one

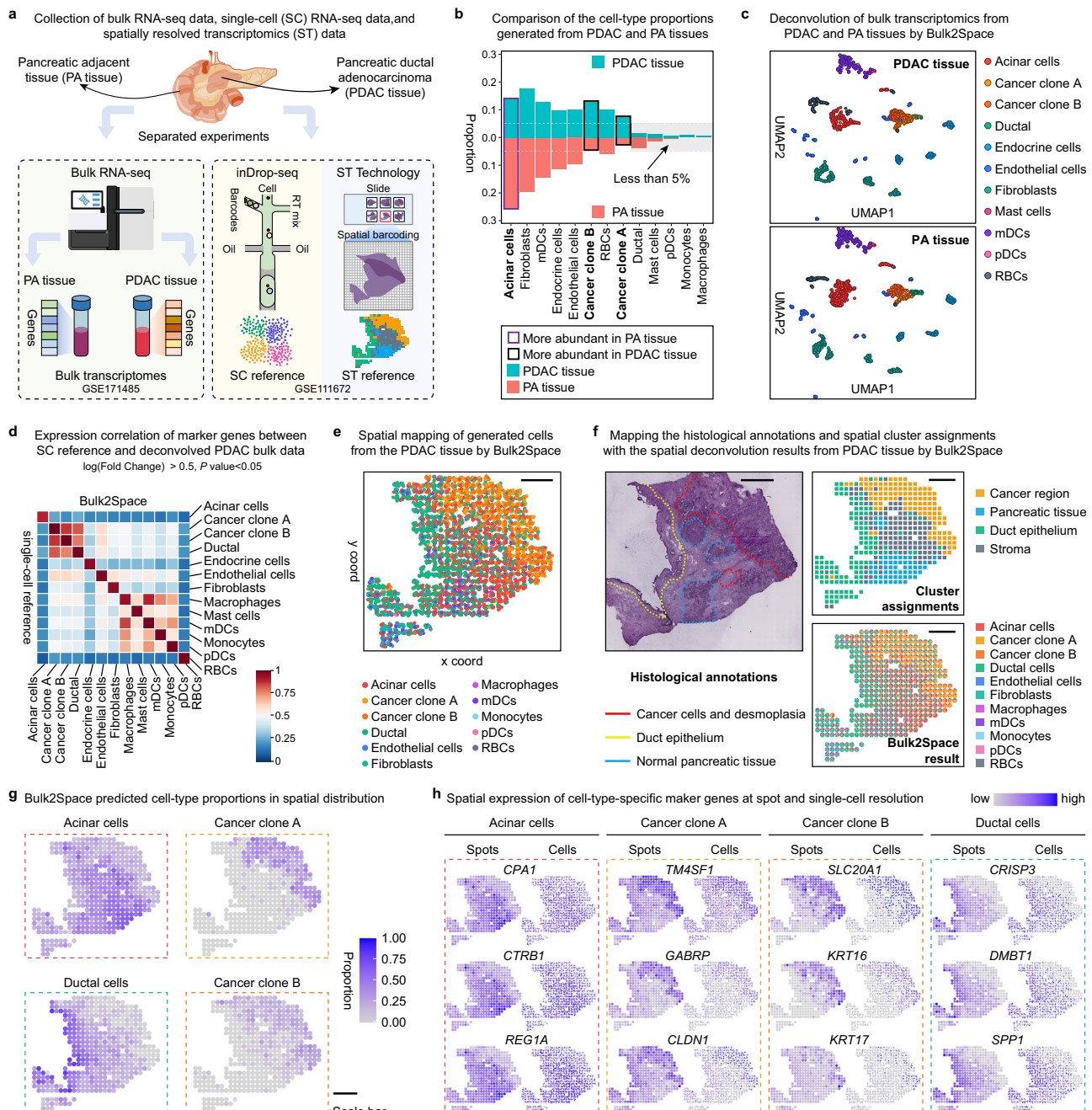

**Fig. 3 | Spatially resolved analysis of PDAC by Bulk2Space. a** Collection of pancreatic adjacent (PA) and PDAC bulk transcriptome data, PDAC scRNA-seq data, and PDAC spatial transcriptomics data. Two bulk data were sequenced by bulk RNA-seq. The scRNA-seq data sequenced by inDrop-seq[51] is used as the single-cell reference. One slice of the sectioned PDAC tissue sequenced by ST[14] was employed as the spatial reference. **b** The cell-type proportions of single cells generated from PDAC and PA tissues by Bulk2Space. Purple box, the proportion of the cell type was higher in PA tissue than that in PDAC tissue. Black box, the proportion of the cell type was higher in PDAC tissue than that in PA tissue. Teal, the PDAC tissue. Red, PA tissue. **c** The clustering space of single cells generated from the PDAC (Top) and PA (Bottom) bulk data by Bulk2Space using UMAP layout. Different colors represented distinct cell types. **d** Pairwise expression correlation of cell-type-specific marker genes between single cells generated by Bulk2Space and the single-cell reference

for PDAC. Marker genes were found by 'FindAllMarkers' function in Seurat. *P* value was calculated with the Wilcoxon Rank Sum test. **e** Spatial mapping of single cells generated from PDAC bulk data. Scale bar, 1 mm. **f** Top, Histological annotation for cancer region (red), duct epithelium (yellow), and normal pancreatic tissue (light blue). Middle, clusters of cancer region (yellow), pancreatic tissue (blue), duct epithelium (green), and stroma (dark gray) from the spatial transcriptomics data. Bottom, spatial deconvolution result of the bulk PDAC data by Bulk2Space with each spot displaying the composition of cell types in a pie chart. Scale bar, 1 mm. **g** The spatial abundance of acinar cells, cancer clone A cells, cancer clone B cells, and ductal cells in each spot on the tissue section predicted by Bulk2Space. Scale bar, 1 mm. **h** The spatial expression of the marker genes in acinar cells, cancer clone A cells, cancer clone B cells, and ductal cells predicted by Bulk2Space at spot (left) and cellular (right) resolution. Source data are provided as a Source Data file.

spatial transcriptomics data[47] were derived from two individual experiments and three different technologies. The scRNA-seq data sequenced by inDrop-seq[51] were used as the single-cell reference for the deconvolution of PA tissue and PDAC bulk data, and

the spatial barcoding-based ST data were used as the spatial reference.

The PA and PDAC bulk data were deconvolved into single-cell data with different cell-type proportions by Bulk2Space. We compared the

cell-type proportions of single cells generated from PA and PDAC bulk data with Bulk2Space. As shown in Fig. 3b, the proportion of acinar cells in PA was significantly higher than that in PDAC, where the proportion of two cancer clone subtypes was higher. The clustering space of single cells generated from PDAC and PA tissue was exhibited in Fig. 3c. The detailed information for the deconvolution of PA tissue was shown in Fig. S19.

Here, we focused on the PDAC bulk data. The expression of cell-type-specific marker genes between generated single-cell profiles by Bulk2Sapce and the reference data was highly correlated (Fig. 3d). Then, each cell was mapped to coordinates based on the spatial reference (Fig. 3e). To further correlate the spatial expression pattern of the tissue with its histological feature, we investigated the molecular architecture of the annotated regions in the spatial reference (Fig. 3f). As shown, the spatial distribution of two cancer clones, acinar cells, and ductal cells predicted by Bulk2Space (Fig. 3e) was consistent with the histological annotation of cancer cells and desmoplasia, normal pancreatic tissues, and duct epithelium, as well as the spatially resolved transcriptomics of the cancer region, pancreatic tissue and stroma, and the duct epithelium, respectively. The spatial cell-type proportions predicted by Bulk2Space were illustrated in Fig. 3g and Fig. S19c. The spatial expression of cell-type-specific marker genes at spot (left) and single-cell (right) resolution for different cell types were shown in Fig. 3h and Fig. S19d. Compared with the spatial distribution of single cells generated by Bulk2Space, the spatial expression of cell-type-specific marker genes exhibited consistent patterns.

## Bulk2Space reconstructs the hierarchical structure of the mouse isocortex region sequenced by Spatial-seq

In addition to linking the histomorphology and transcriptomics in pathological tissues, another application scenario of Bulk2Space is to reconstruct the structure of tissues with spatial patterns. For instance, the spatial organization of the isocortex region of the mouse brain exhibits a layered pattern. Therefore, Bulk2Space was applied to reconstruct the hierarchical structure of the mouse isocortex region through spatial deconvolution of the bulk transcriptomics data. The bulk transcriptomics data of mouse isocortex were sequenced by our in-house developed multiplexed RNA-seq approach, termed Spatial-seq. The detailed information for Spatial-seq was described in the 'Methods' section and illustrated in Fig. S20a. In short, the laser capture microdissection (LCM) was used to isolate regions of interest from the tissue sections and each isolated tissue was collected independently. Then, barcoded beads were used to capture and label the mRNA from the collected samples. Finally, the captured mRNA was pooled together for RNA-seq. The transcriptome data of each sample can be obtained by identifying the sample-specific barcode sequence. Using Spatial-seq, we isolated and sequenced 13 main brain regions from coronal and sagittal sections across the entire mouse brain (Fig. S20b).

As shown in Fig. 4a, the bulk transcriptomics data of the mouse isocortex was obtained by Spatial-seq. The mouse primary visual cortex regions in different coronal sections from anterior to posterior were collected and sequenced by SMART-seq2[52]. The result scRNA-seq data were used as the single-cell reference. A sagittal section of the mouse brain was divided into two parts and sequenced using 10X Visium to obtain spatial transcriptomics data of the isocortex region. The spatial reference data were downloaded from 10X datasets. The detailed information of datasets used in this study was summarized in Supplementary Data 1.

The bulk data were first deconvolved into single-cell RNA-seq data (Fig. 4b). Subsequently, the generated single cells were mapped to spatial locations by Bulk2Space and the spatial distribution of cell types showed a distinct layered pattern (Fig. 4c). The expression of cell-type-specific marker genes was highly correlated between Bulk2-Space results and the single-cell reference data (Fig. 4d). The spatial

distribution of the cell-type proportion in each spot predicted by Bulk2Space was illustrated in Fig. 4e, which corresponded well with the hierarchical structure of the mouse isocortex. The spatial distributions of cell-type proportions predicted by Bulk2Space for seven cell types including Astro, L2/3 IT, L4, L5 IT, L5 PT, L6 CT, and L6 IT cells were shown in Fig. 4f. The results supported the ability of Bulk2Space to reconstruct the spatial organization of tissues. Moreover, the spatial expression patterns of cell-type-specific marker genes predicted by Bulk2Space at single-cell resolution were consistent with the proportion distributions of the corresponding cell types (Fig. 4g).

Different from traditional spatial deconvolution algorithms such as RCTD[34], SpatialDWLS[35], stereoscope[36], and SPOTlight[37], Bulk2Space can generate spatially resolved single-cell transcriptomics data from bulk RNA-seq. Besides, Bulk2Space allowed us to analyze the spatial heterogeneity of individual cells from the spatially resolved transcriptomics data without single-cell resolution.

## Bulk2Space re-annotates ambiguous cells in the mouse hypothalamus

The Bulk2Space algorithm was also used to spatially deconvolve bulk transcriptome data derived from the hypothalamus region of the mouse brain using Spatial-seq, to explain the spatial distribution of single cells and gene expression. The resources of the bulk, single-cell, and spatial data was illustrated in Fig. 5a. The single-cell reference was obtained by profiling ~31000 cells using Drop-seq and the MERFISH data of the mouse hypothalamus were used as the spatial ref. 43. More details were summarized in Supplementary Data 1.

After deconvolution of the bulk RNA-seq data of the mouse hypothalamus by Bulk2Space, the clustering space of the generated single cells was close to that of the MERFISH data (Fig. 5b). The generated single cells were then mapped to spatial coordinates based on the image-based spatial reference (Fig. 5c). As shown in Fig. 5c, the cell-type distributions in MERFISH data and Bulk2Space results were comparable at single-cell resolution. Among the 10 cell types, immature oligodendrocytes were unique to MERFISH, and macrophages were predicted by Bulk2Space, but absent in MERFISH data. The heatmap suggested that 7 of the 8 shared cell types were strongly correlated in both MERFISH and Bulk2Space results with an average pairwise correlation over 0.9. The poor correlation of mural cells between MERFISH data and Bulk2Space results may be related to the few numbers of pericytes. The spatial patterns of the MERFISH targeted genes were consistent with that of Bulk2Space results (Fig. 5d).

Meanwhile, owing to the limited number of measured target genes, some ambiguous cells in MERFISH data could not be further identified according to their expression profiles. However, using Bulk2Space, the generated single cells with unbiased gene expression were assigned to the spatial context, thus further clustering these ambiguous cells using novel genes that were absent in MERFISH data (Fig. 5e). Meanwhile, the spatial distribution of novel genes beyond the targeted RNA species was predicted by Bulk2Space and was consistent with the spatial pattern of corresponding cell types as illustrated in Fig. 5f and Fig. S21.

## Bulk2Space uncovers spatial gene expression dynamics in different stages of the inflammation-induced prostate cancer

Tumor development is linked to chronic infection, dietary factors, obesity, inhaled pollutants, tobacco use, and autoimmunity. The unifying principle underlying these processes is inflammation, which is an aberrantly prolonged form of a protective response to a loss of tissue homeostasis[53]. To explore whether Bulk2Space could identify the cellular and molecular pathways that coordinate the tumor-promoting effects in inflammation-induced cancer, we deconvolved the bulk transcriptome data of prostate cancer and mapped single cells generated by Bulk2Space to the corresponding spatial references, including normal glands, inflammatory tissue, and tumor sites. The

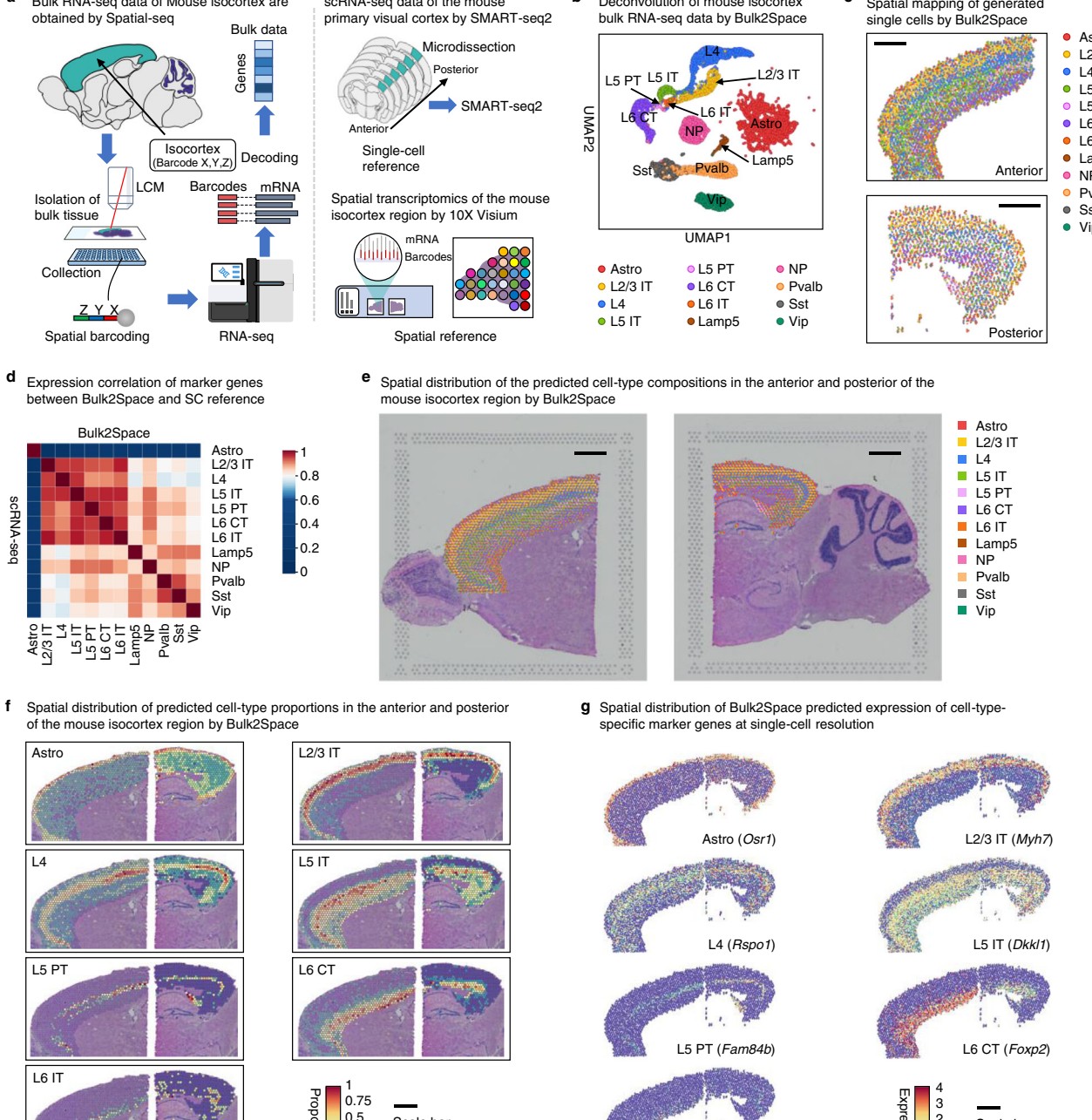

**Fig. 4 | Spatially resolved single-cell analysis of the mouse isocortex bulk data by Bulk2Space.** **a** The resources of the bulk transcriptome data, single-cell reference, and spatial reference. The bulk RNA-seq data of the mouse isocortex was sequenced using our in-house developed Spatial-seq which combines LCM and spatial barcoding strategies. The single-cell reference data is derived from scRNA-seq of a collection of consecutive mouse primary visual cortex sections from anterior to posterior. The spatial reference data is obtained from an open-access database using the 10X Visium approach. **b** The deconvolution results of the mouse isocortex bulk data by Bulk2Space. The UMAP layout showed the clustering space resources of the bulk (from TCGA), single-cell[54], and spatial data[55] were listed in Supplementary Data 1.

of the generated single cells. **c** The spatial mapping of the single cells generated from the mouse isocortex bulk data by Bulk2Space. Scale bar, 1 mm. **d** The pairwise expression correlation of cell-type-specific marker genes between single cells generated by Bulk2Space and single cells in the reference data. **e** The spatial distribution of the cell-type proportion predicted by Bulk2Space. Scale bar, 1 mm. **f** The spatial cell-type abundance of seven cell types with layered structure predicted by Bulk2Space. Scale bar, 1 mm. **g** The spatial expression distribution of cell-type-specific marker genes predicted by Bulk2Space at single-cell resolution. Scale bar, 1 mm. Source data are provided as a Source Data file.

These tissue regions were isolated from the prostate, and three of the sections were subjected to RNA-seq based on spatial barcoding (Fig. 6a). Next, Bulk2Space was used to deconvolve the bulk data into single-cell profiles (Fig. 6b) and mapped the generated single cells to spatial coordinates (Fig. 6c). Notably, the composition and distribution of cells in different tissue regions were quite different and closely related to the state of the tissue (Fig. 6d, e). A comparison of cell distribution between normal, inflammatory, and cancerous tissues (Fig. 6f–h) showed that in normal glands, cancer-associated fibroblasts (CAFs) were the most abundant, but in inflammatory and cancer tissues, the proportions of CAFs were significantly lower (52.4% in normal, 20.5% in inflammation, and 27.5% in cancer). In the areas of

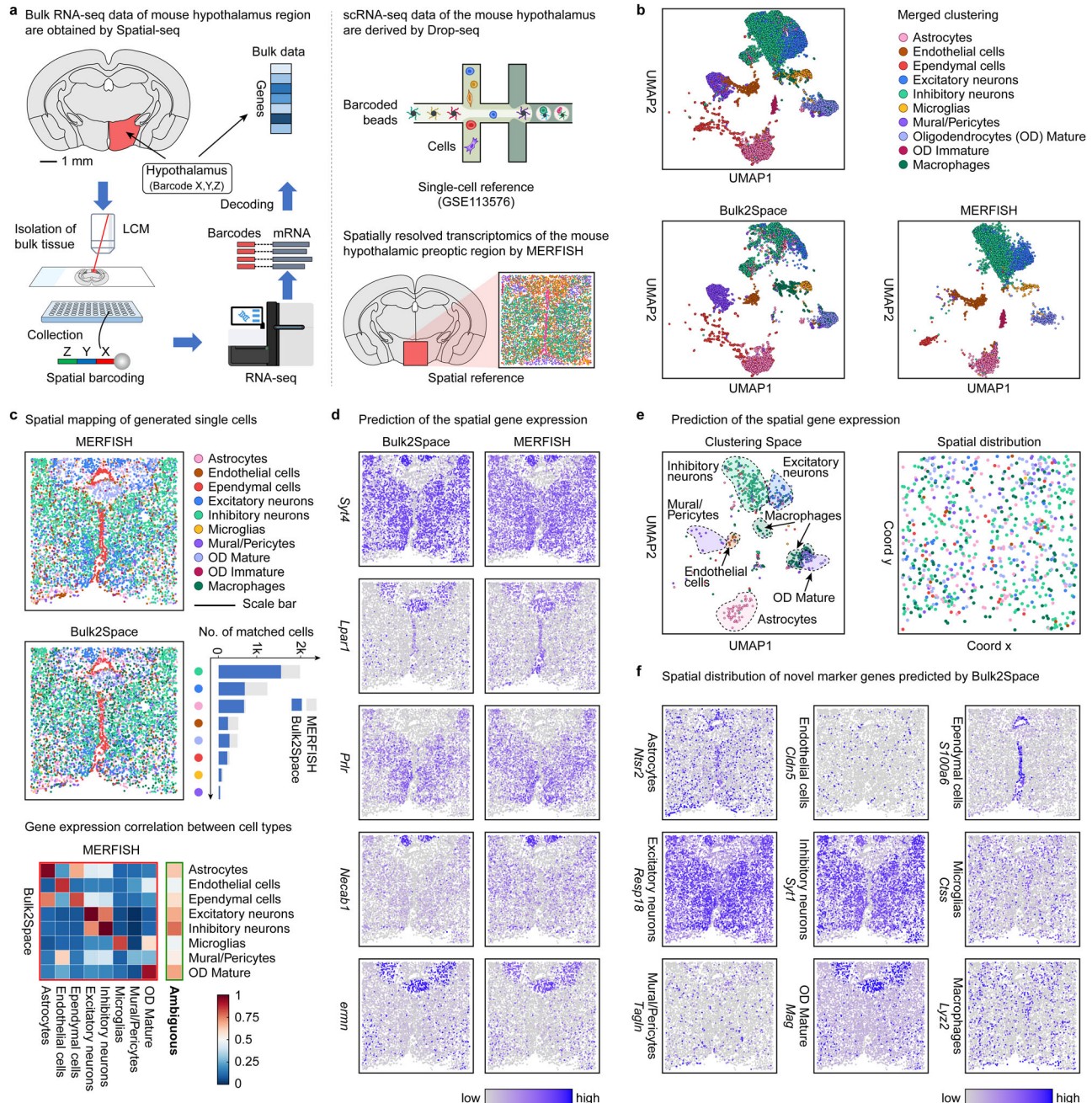

**Fig. 5 | Spatially resolved analysis of the mouse hypothalamus bulk data by Bulk2Space. a** The resources of the bulk transcriptome data, single-cell reference, and spatial reference. The bulk RNA-seq data of the mouse hypothalamus was profiled using our in-house developed Spatial-seq. The single-cell reference data was derived from Drop-seq. The spatial reference data is obtained from MERFISH. **b** The UMAP layout of single-cell profiles in MERFISH data and generated from bulk tissue by Bulk2Space. Top, clustering of all cells in both datasets. Two heterogeneous sets of data were aligned through a joint analysis. Each cell type was represented by a unique color. **c** The spatial distribution of distinct cells in MERFISH and Bulk2Space results. The stacked bar chart showed the cell number of each cell type (gray) and the number of cells predicted by Bulk2Space to the

corresponding cell type at each coordinate (blue). Bottom, Pearson correlation of gene expression between cell types in MERFISH data and Bulk2Space results. The correlation of shared cell types was in red border and the expression correlation between MERFISH ambiguous cells and Bulk2Space predicted cell types was in green border. **d** The spatial expression of genes predicted by Bulk2Space and in the ground truth (MERFISH). **e** Re-annotation of ambiguous cells in MERFISH using Bulk2Space results. Left, clustering of the ambiguous cells. Right, spatial assignment of cells that were colored according to their predicted cell types. **f** Predicted spatial expression patterns of novel marker genes for different cell types. Source data are provided as a Source Data file.

inflammation, cells involved in the inflammatory response, such as immunocompetent B cells, endothelial cells, monocytes, and NK cells (Fig. 6g), were more abundant than in normal or tumor tissues (13.4% in inflammation, 2.0% in normal, and 1.7% in cancer). In cancer with a Gleason score (Gs) of 3 + 4, the proportion of cancer epithelial cells was significantly higher (53.0% in cancer, 6.3% in normal, and 4.3% in

inflammation) than that in normal and inflammatory areas (Fig. 6h), while the proportion of the periventricular layer cells was significantly lower (17.9% in cancer, 39.3% in normal, and 44.1% in inflammation).

During the progress of inflammation-induced prostate cancer, CAFs accumulated in normal glands at the early stage, resulting in high expression of inflammatory factors and promoting the occurrence of

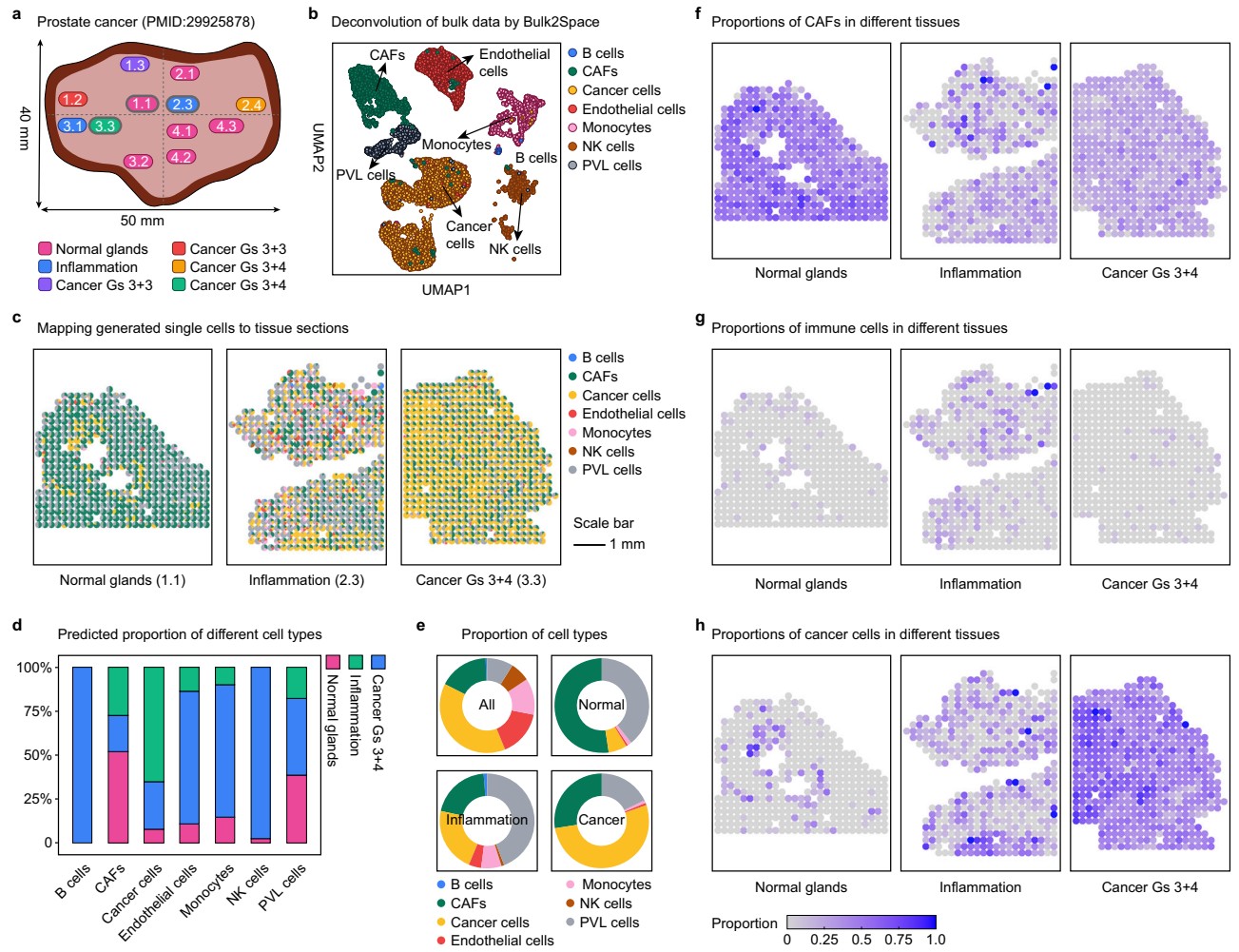

**Fig. 6 | Spatial gene expression dynamics analysis of inflammation-induced prostate cancer by Bulk2Space. a** Experimental design of the bulk transcriptome data. The tissue was segmented into distinct regions, namely, normal glands (pink), inflammation (blue), and several cancer regions (red, green, purple, and yellow). Three tissue regions (1.1 normal, 2.3 inflammation, and 3.3 cancer region) were isolated to conduct spatially resolved transcriptomics. **b** The t-SNE layout of the generated single-cell profiles by Bulk2Space. Each cell type was represented by a unique color. **c** The proportions of generated single cells that were mapped to normal, inflammation, and cancer region by Bulk2Space, respectively. The color of each cell type was consistent with (**b**). **d** The proportion of cells assigned to normal (pink), inflammation (blue), and cancer (green) regions for each cell type. **e** The predicted proportion of distinct cell types for each tissue region by Bulk2Space. **f** The proportions of cancer-associated fibroblasts (CAFs) in normal (left), inflammation (middle), and cancer regions (right). **g** The proportions of immune cells in normal, inflammation, and cancer regions. **h** The proportions of cancer epithelial cells in normal, inflammation, and cancer regions. Source data are provided as a Source Data file.

local inflammation, a fundamental innate immune response to perturbed tissue homeostasis, thus leading to tissue cancerization[53]. Bulk2Space results were consistent with this phenomenon.

## Discussion

Understanding the transcriptional heterogeneity within the tissue from the perspective of single-cell spatial resolution and full gene coverage is an essential direction for the development in the field of biological sciences[1,56]. The spatial locations of cells may determine their identity and how they interplay with each other in the microenvironment[57,58]. However, there are many technical challenges, and thus, such a technology has not been fully realized yet. Traditional bulk transcriptome data can provide gene expression patterns and reflect the overall status of tissue, which is widely used in studies on the occurrence and development of diseases. Nevertheless, disease-inherent cell composition and spatial distribution are difficult to analyze. Although several attempts can infer the cell-type proportions from traditional RNA-seq data, understanding bulk tissue at single-cell spatial resolution remains a pressing need in the field. Cell identity can

be characterized by the clustering space of single-cell gene expression profiles[59], and expression features can be stably retained across different conditions, technologies, and species[29]. Based on this, we used a deep learning model, termed β-VAE, to generate single-cell profiles with biological significance within the clustering space of each cell type and map them accurately to tissue coordinates, thus deconvolving the bulk transcriptomics into spatially resolved single-cell transcriptomics data.

As thoroughly considered the merits and demerits of the widespread use of two spatially resolved transcriptomics technologies, we employed different spatial mapping strategies for each method. For image-based in situ hybridization, Bulk2Space could predict the spatial expression of mRNA species apart from the target genes. Moreover, Bulk2Space can help annotate cells whose types are difficult to further distinguish by targeted methods. For spatial barcoding-based RNA-seq, each sequenced tissue region contains transcripts from different cells and can thus be regarded as a mixture of cells, not allowing single-cell resolution. Bulk2Space can assign single cells to optimal spots, and these mixtures are split into collections of individual cells, thus

providing a spatially resolved single-cell deconvolution strategy for these spatial data. Our results showed that Bulk2Space had great biological and clinical application prospects, including linking tissue molecular characteristics with histological phenotypes, revealing the spatial specific variation of cells, discovering the spatial expression pattern of novel genes, achieving more refined cell clustering, and predicting the molecular mechanism underlying the progression of the disease.

## Methods

### Datasets

All scRNA-seq, spatially resolved transcriptomics, and bulk RNA-seq datasets used in this study were collected from high-quality publications, Gene Expression Omnibus (GEO), and The Cancer Genome Atlas (TCGA), wherein unannotated, ambiguous, or low quality-cells were excluded. The detailed description of each dataset was summarized in Supplementary Data 1.

### Computing environment

The Bulk2Space was developed on two workstations, as listed below.

**Workstation 1**. Dell Precision Tower 7820 Workstation, CPU (Intel Xeon Gold 5118, 2.3 GHz × 2), RAM (64 GB, 16 GB × 4, DDR4, 2933 MHz), Hard Drive (SSD, SATA Class 20, 512 GB; HDD, 7200 rpm, SATA), Graphics Card (NVIDIA, Quadro P4000, 8 GB), Operating System (Ubuntu 16.04), Running Environment (CUDA 11.6, Torch 1.12.1, Python 3.8.5, deep-forest 0.1.5, easydict 1.9, numpy 1.19.2, pandas 1.1.3, scanpy 1.8.1, scikit-learn 1.0.1, scipy 1.5.2, tqdm 4.50.2, Unidecode 1.3.0).

**Workstation 2**. Dell Precision Tower 7920 Workstation, CPU (Intel Xeon Gold 6230, 2.1 GHz × 2), RAM (192 GB, 16 GB × 12, DDR4, 2933 MHz), Hard Drive (SSD, SATA Class 20, 512 GB; HDD, 7200 rpm, SATA), Graphics Card (NVIDIA, RTX2080Ti VIDEO CARD V2 × 2, 22 GB), Operating System (Ubuntu 18.04), Running Environment (CUDA 11.0, Torch 1.7.1, Python 3.8.5, deep-forest 0.1.5, easydict 1.9, numpy 1.19.2, pandas 1.1.3, scanpy 1.8.1, scikit-learn 1.0.1, scipy 1.5.2, tqdm 4.50.2, Unidecode 1.3.0).

### Data processing

All data were preprocessed using R (version 4.1.1). For mouse brain scRNA-seq data by Moffitt[43] et al., we filtered 841 ambiguous cells and 88 unstable cells. For human prostate cancer scRNA-seq data by Wu[54] et al., we filtered 183 unassigned cells. All cells of other scRNA-seq as well as spatially resolved transcriptomics datasets were retained. For mouse hypothalamus MERFISH data, we filtered 5 "blank" barcodes as well as *the Fos* gene, whose expression value in all cells was 'NA'. For human prostate cancer RNA-seq data, we used biomaRt (version 2.48.3) to transform ensemble id to gene name. For all scRNA-seq and spatially resolved transcriptomics datasets, the raw counts were normalized using the global-scaling normalization method "LogNormalize" by Seurat (version 4.0.4)[60]. For the MERFISH dataset, the normalized data was determined as the raw count per cell divided by the cell volume and scaled by 1000.

### Step 1: Deconvolution of bulk transcriptome data

**Cell type proportions prediction.** For a given bulk RNA-seq dataset, we aimed to calculate its cell-type proportions firstly, and then generate single-cell gene expression profiles based on the calculated proportions. Since cell type and gene expression of all single cells can be accessed from the scRNA-seq data reference, we collected cells with the same cell type and then average their gene expression. This average vector $c_i \in \mathbb{R}^N$ is defined as the gene expression of cell type

$i$ ($i \in \{1,2,\cdots,C\}$, where $C$ denotes the total number of cell types). Given the gene expression vector $\mathbf{x} \in \mathbb{R}^N$ of a bulk RNA-seq dataset, we aimed to predict the cell type proportion:

$$\sum_{i=1}^{C} p_i \mathbf{c}_i = \mathbf{x} \tag{1}$$

Here, $p_i$ is the proportion of cell type $i$.

We applied least square estimation (LSE) to estimate $p_i$ by minimizing the squared discrepancies between observed data.

$$\min_{\mathbf{p}} ||\mathbf{x} - \mathbf{C}\mathbf{p}||_2 \tag{2}$$

where $\mathbf{C} \in \mathbb{R}^{N \times C}$ denotes the cell type gene expression profile, and each column represents a different cell type. $\mathbf{p} = [p_1, p_2, \cdots, p_C]^T$ is the proportion vector to be estimated.

**Single-cell simulation by β-VAE.** Let $\mathcal{D} = \{X, V, W\}$ be a set consisting of gene expression vectors $x \in \mathbb{R}^N$ and two sets of ground truth data generative factors: conditionally independent factors $v \in \mathbb{R}^K$, where $\log p(v|x) = \sum_k \log p(v_k|x)$; and conditionally dependent factors $w \in \mathbb{R}^H$. We assume that $x$ are generated by a true world simulator $S$ using the corresponding ground truth data generative factors: $p_\theta(x|v,w) = S(v,w)$, where $\theta$ is the generative model parameter.

We wanted to develop an unsupervised deep generative model that, using samples from $X$ only, can learn the joint distribution of the data $x$ and a set of generative latent factors $z$ ($z \in \mathbb{R}^M$, where $M \geq K$) that can be used to generate the observed data $x$; i.e., $p_\theta(x|z) \approx p(x|v,w) = S(v,w)$. However, since the integral of the marginal likelihood $p_\theta(x) = \int p_\theta(\mathbf{z}) p_\theta(\mathbf{x}|\mathbf{z}) \, d\mathbf{z}$ is intractable (so we cannot evaluate or differentiate the marginal likelihood), the true posterior density $p_\theta(\mathbf{z}|x) = p_\theta(\mathbf{x}|\mathbf{z}) p_\theta(\mathbf{z})/p_\theta(\mathbf{x})$ is intractable.

To solve this problem, for a given observation $x$, we described the inferred posterior configurations of the latent factors $\mathbf{z}$ through a probability distribution $q_\phi(\mathbf{z}|\mathbf{x})$: an approximation to the intractable true posterior $p_\theta(x|z)$. We aimed to ensure that the inferred latent factors $q_\phi(\mathbf{z}|\mathbf{x})$ capture the generative factors $v$ in a disentangled manner. A disentangled representation can be defined as one where single latent units are sensitive to changes in single generative factors, while being relatively invariant to changes in other factors. In a disentangled representation, knowledge about one factor can generalize to novel configurations of other factors. The conditionally dependent data generative factors $w$ can remain entangled in a separate subset of $\mathbf{z}$ that is not used for representing $v$.

An intuitive approach was to minimize the KL divergence between the approximate and the true posterior:

$$
\begin{aligned}
\mathcal{D}_{KL}\Big(q_\phi(\mathbf{z}|\mathbf{x})||p_\theta(\mathbf{z}|x)\Big) &= -\sum_{\mathbf{z}} q_\phi(\mathbf{z}|\mathbf{x}) \log\left(\frac{p_\theta(\mathbf{z}|x)}{q_\phi(\mathbf{z}|\mathbf{x})}\right) \\
&= -\sum_{\mathbf{z}} q_\phi(\mathbf{z}|\mathbf{x}) \log\left(\frac{\frac{p_\theta(x,\mathbf{z})}{p_\theta(x)}}{q_\phi(\mathbf{z}|\mathbf{x})}\right) \\
&= -\sum_{\mathbf{z}} q_\phi(\mathbf{z}|\mathbf{x})\left[\log\left(\frac{p_\theta(\mathbf{x}|\mathbf{z})}{q_\phi(\mathbf{z}|\mathbf{x})}\right) - \log(p_\theta(x))\right] \\
&= -\sum_{\mathbf{z}} q_\phi(\mathbf{z}|\mathbf{x}) \log\left(\frac{p_\theta(\mathbf{x}|\mathbf{z})}{q_\phi(\mathbf{z}|\mathbf{x})}\right) + \log(p_\theta(x)) \\
&= -\mathcal{L}(\theta, \phi; \mathbf{x}) + \log(p_\theta(x))
\end{aligned}
$$
$$\tag{3}$$

Here, $\mathcal{L}(\theta, \phi; \mathbf{x})$ is called the variational lower bound, and can be written as:

$$
\begin{aligned}
\mathcal{L}(\theta, \phi; \mathbf{x}) &= \sum_{\mathbf{z}} q_\phi(\mathbf{z}|\mathbf{x}) \log\left(\frac{p_\theta(\mathbf{x}|\mathbf{z})}{q_\phi(\mathbf{z}|\mathbf{x})}\right) \\
&= \sum_{\mathbf{z}} q_\phi(\mathbf{z}|\mathbf{x}) \log\left(\frac{p_\theta(\mathbf{x}|\mathbf{z})p_\theta(\mathbf{z})}{q_\phi(\mathbf{z}|\mathbf{x})}\right) \\
&= \sum_{\mathbf{z}} q_\phi(\mathbf{z}|\mathbf{x})\left[\log(p_\theta(\mathbf{x}|\mathbf{z})) + \log\left(\frac{p_\theta(\mathbf{z})}{q_\phi(\mathbf{z}|\mathbf{x})}\right)\right] \\
&= \mathbb{E}_{q_\phi(\mathbf{z}|\mathbf{x})}[\log(p_\theta(\mathbf{x}|\mathbf{z}))] - \mathcal{D}_{KL}\left(q_\phi(\mathbf{z}|\mathbf{x})||p_\theta(\mathbf{z})\right)
\end{aligned}
\tag{4}
$$

To encourage this disentangling property in the inferred $q_\phi(\mathbf{z}|\mathbf{x})$, we introduced a constraint over it by trying to match it to a prior $p(\mathbf{z})$ that can both control the capacity of the latent information bottleneck, and embodies the desiderata of statistical independence mentioned above. We set the prior to be an isotropic unit Gaussian $p(z) \sim \mathcal{N}(0, I)$, then the constrained optimization problem can be written as:

$$
\max_{\phi, \theta} \mathbb{E}_{q_\phi(\mathbf{z}|\mathbf{x})}[\log(p_\theta(\mathbf{x}|\mathbf{z}))] \quad subject\ to \quad \mathcal{D}_{KL}\left(q_\phi(\mathbf{z}|\mathbf{x})||p_\theta(\mathbf{z})\right) < \epsilon \tag{5}
$$

where $\epsilon$ specifies the strength of the applied constraint. Re-writing the above equation as a Lagrangian under the KKT conditions, we obtain:

$$
\mathcal{F}(\theta, \phi, \beta; \mathbf{x}, \mathbf{z}) = \mathbb{E}_{q_\phi(\mathbf{z}|\mathbf{x})}[\log(p_\theta(\mathbf{x}|\mathbf{z}))] - \beta\left(\mathcal{D}_{KL}\left(q_\phi(\mathbf{z}|\mathbf{x})||p_\theta(\mathbf{z})\right) - \epsilon\right) \tag{6}
$$

where the KKT multiplier $\beta$ is the regularization coefficient that constrains the capacity of the latent information channel $\mathbf{z}$ and puts implicit independence pressure on the learnt posterior due to the isotropic nature of the Gaussian prior $p_\theta(\mathbf{z})$. Since $\beta, \epsilon \geq 0$, according to the complementary slackness KKT condition, the equation can be re-written as:

$$
\begin{aligned}
\mathcal{F}(\theta, \phi, \beta; \mathbf{x}, \mathbf{z}) \geq \mathcal{L}(\theta, \phi; \mathbf{x}, \mathbf{z}, \beta) &= \mathbb{E}_{q_\phi(\mathbf{z}|\mathbf{x})}[\log(p_\theta(\mathbf{x}|\mathbf{z}))] \\
&\quad - \beta\left(\mathcal{D}_{KL}\left(q_\phi(\mathbf{z}|\mathbf{x})||p_\theta(\mathbf{z})\right)\right)
\end{aligned}
\tag{7}
$$

which is the β-VAE formulation with the addition of the $\beta$ coefficient.

Here, different $\beta$ will change the degree of applied learning pressure during training, thereby encouraging different learned representations. We postulated that to learn disentangled representations of the conditionally independent data generative factors $\boldsymbol{v}$, it is important to set $\beta > 1$ to impose a stronger constraint on latent bottleneck than the original VAE. These constraints limit the capacity of $\mathbf{z}$, coupled with the pressure to maximize the log likelihood of the training data $\mathbf{x}$ under the model, encouraging the model to learn the most efficient representation of the data. The additional pressure from high $\beta$ values may create a trade-off between reconstruction fidelity and the quality of disentanglement within the learned latent representations. When the appropriate balance is found between information preservation (reconstruction cost as regularization) and latent channel capacity restriction ($\beta > 1$), disentangled representations emerge.

**Model configuration.** In step1, we simulate single-cell by β-VAE. Both the encoder and decoder apply a four-layer perceptron, where each layer is followed by a RELU activation except the last layer of the encoder. For the encoder, the number of neurons in each layer is 2048, 1024, 512 and 512 respectively. While for the decoder, the number of neutrons in each layer is 512, 1024, 2048 and $k$, respectively, where $k$ represents the number of genes in the dataset. The dimensionality of the latent space learned by the beta-VAE is 256. The relative weighting

of the reconstruction loss and regularization loss is 1:4. We apply the Adam with weight decay (AdamW) optimizer with an initial learning rate as 1e−4, the decoupled weight decay as 5e−4 and Adam's $\beta$ parameters as 0.9 and 0.999. Moreover, the default running epoch is fixed to 3000, but we use early stopping during the training phase, with which we stop training when the training loss no longer reduces for 50 epochs.

Here we introduce the implementation details of our benchmark approaches. For GAN and cGAN, the generator consists of two fully connected layers, followed by LeakyRELU and RELU activations, respectively. The discriminator is also made up of two fully connected layers, where the first layer is followed by a LeakyRELU activation. We use the Adam optimizer with the initial learning rate of 1e-4 and the betas parameters of 0.5 and 0.999. We train util the loss in the generative and discriminative phases is no longer reduced for 50 epochs, and use the model obtained at this time as the final model for prediction.

**Step2: Mapping generated single cells to spatial locations**
**Spatial barcoding-based RNA-seq.** After obtaining the generated data, we needed to map each cell and the spot it belongs to. To improve the accuracy and reduce the complexity of this process, we divided it into two steps. First, we calculated the cell type proportion of each spot, and then predicted which cells are contained in each spot based on this proportion.

1. Calculate cell type proportions. The procedure is identical to the Step 1 deconvolution, we repeated this process to obtain the cell type proportion of each spot in the tissue.
2. Deep forest for spot recommendation. With cell type proportion, we next predicted which cells are contained in each spot. We defined this task as a binary classification problem. We designed a classifier, whose input is the gene expression vectors of both single cell $\mathbf{s}$ and spot $\mathbf{x}$. When the cell belongs to the spot, the output of the classifier is 1, otherwise the output is 0. Specifically, the input can be represented as $\mathbf{v} = [\mathbf{s}; \mathbf{x}; \mathbf{s} - \mathbf{x}] \in \mathbb{R}^{N \times 3}$, where semicolon means concatenate operation. $\mathbf{s} - \mathbf{x} \in \mathbb{R}^N$ can be regarded as an auxiliary signal, since when there is a negative number occurring in this vector, the single cell must not belong to the spot.

We applied multi-Grained Cascade Forest (gcForest), a decision tree ensemble with a cascade structure and further enhanced by multi-grained scanning, to build our classifier. Each level of cascade receives feature information processed by its preceding level and outputs its processing result to the next level. Each level is an ensemble of decision tree forests, i.e., an ensemble of ensembles. We also included completely-random tree forests) to encourage diversity. Given an instance, each forest counts the percentage of different classes of training samples at the leaf node where the concerned instance falls, and then average across all trees in the same forest to produce an estimate of class distribution. The estimated class distribution forms a class vector, which was generated by $k$-fold cross validation and then concatenated with the original feature vector to be input to the next level of the cascade.

To enhance cascade forest, we introduced a procedure of multi-grained scanning. We used sliding windows to scan the raw features. Specially, we used a window with a fixed length of $k$ to slide the input vector into $k$-dimensional feature vectors. We then feeded these vectors to $n$ forests and finally produce $n$ two-dimensional vectors. By using multiple sizes $k$ of sliding windows, differently grained feature vectors were generated.

The we summarized the overall procedure of gcForest. For the original raw features, three window sizes $\{\lfloor N/16 \rfloor, \lfloor N/8 \rfloor, \lfloor N/4 \rfloor\}$ were used for multi-grained scanning. The generated data were used to train a completely-random tree forest containing 100 trees. We concatenated the output of these two forests as the transformed feature vectors, which also acted as the input of cascade forest. The final

model was a cascade of cascades, where each cascade consisted of multiple levels each corresponding to a grain of scanning. Each level consisted of 4 completely-random tree forests, each containing 100 trees. In other words, the transformed feature vectors were augmented with the class vector generated by the previous grade, and then were used to train the current grade of cascade forests.

Given a test instance, it would obtain its corresponding transformed feature vector through the multi-grained scanning procedure, and then go through the cascade till the last level. The two-dimensional class vectors at the last level were aggregated, and the class with the largest aggregate value was selected to obtain the final prediction result.

This decision tree ensemble approach not only had much fewer hyper-parameters than deep neural networks, but also retained the interpretability of tree models. Its model complexity could be automatically determined in a data-dependent manner, which made gcForest work well even on small-scale data.

**Image-based in situ hybridization.** We developed a simple method for predicting the spatial distributions of genes not measured in spatial transcriptomic data which produced by in situ RNA imaging-based technologies. To do this, Bulk2Space used the spatial transcriptomic data as the reference, and generated scRNA-seq data were mapped to the tissue space corresponding to this reference. To eliminate the differences between scRNA-seq data and spatial transcriptomic data caused by different experiment types, an empirical Bayes framework was used to remove the batch effect, so that both sets of data are at the same scale level. Subsequently, the cross-dataset k-nearest neighbor graph of spatial transcriptomic data in scRNA-seq data was computed in the aligned space, and then the predicted whole gene expression profile of each cell of spatial transcriptomic data could be calculated to the mean of its k nearest neighbors in scRNA-seq data.

**Model configuration.** Deep forest is applied for spot recommendation. Specifically, the number of samples used to construct feature discrete bins is set to 200000. If the size of training set is smaller than it, then all training samples will be used. The type of binner used to bin feature values into integer-valued bins is "percentile", which means each bin will have approximately the same number of distinct feature values. We set the maximum number of cascade layers in the deep forest as 20, and apply 2 estimator in each cascade layer. Gini impurity is used to measure the quality of a split. We have no constraints on the maximum depth of each tree. The training process terminates when the validation performance on the training set does not improve compared against the best validation performance achieved so far to 2 tolerant rounds. And the counting on tolerant rounds is triggered if the performance of a fitted cascade layer does not improve by 1e-5 compared against the best validation performance achieved so far.

We also introduce the implementation details of our benchmark approaches. (1) For Logistic Regression (LR), we implement L2 regularization as the additional penalty term to solve the problem of overfitting. We use L-BFGS algorithm as the solver, which uses the Hessian matrix to iteratively optimize the loss function. We fit the model according to the given training data and return the probability estimates on testing set. (2) For Decision Tree (DT), we use the Gini impurity to measure the quality of a split and choose the best split at each node. We don't set any maximum depth of the tree, so nodes are expanded until all leaves are pure or until all leaves contain less than 2 samples. We build the decision tree classifier from the training set, and then predict class probabilities of the input samples. (3) For Gradient Boosting Decision Tree(GBDT), we set the learning rate as 0.1 and the number of boosting stages to perform as 100. The loss function to be optimized is log loss function, and we choose Friedman MSE to measure the quality of a split. The maximum depth of the individual regression estimators is set to

3. We fit the gradient boosting model on the training set and use the trained model to make predictions on testing set. (4) For Multilayer Perceptron(MLP), we apply a two-layer perceptron, where each layer is followed by a batch normalization, a RELU activation, and a dropout layer with the probability of element zeroing as 0.1. We use the Adam optimizer with the initial learning rate of 1e-4 and the betas parameters of 0.9 and 0.999. We train until the loss is no longer reduced for 30 epochs, and use the model obtained at this time as the final model for prediction. (5) For DeepGBM, we set the number of tree groups to 100. The dimension of leaf embedding for a tree group is set to 20. The structure of the distilled NN model is a fully connected network with "100-100-100-50" hidden layers. We adopt the feature selection in each tree group, where we first sort the features according to the information gain, and the top 128 of them are selected as the inputs of distilled NN model. We use the Adam with weight decay (AdamW) optimizer with an initial learning rate as 2e-3, the decoupled weight decay as 1e-6, and Adam's betas parameters as 0.9 and 0.999. We train until the model reaches the highest ROC-AUC score, and use it for prediction.

## Performance evaluation of Bulk2Space (simulated datasets)
### Benchmarking the deconvolution step of Bulk2Space on simulated datasets.

1. Simulated bulk data and the single-cell reference were generated from same datasets (paired simulations).

**Datasets.** Ten scRNA-seq datasets were applied to benchmark our method, including human and mouse primary tissues. Five human scRNA-seq datasets including peripheral blood[46] ("GSE92495"), brain[61] ("GSE103723"), kidney[62] ("GSE121862"), liver[63] ("GSE124395"), and lung[64] ("GSE130148"). Five mice scRNA-seq datasets including brain[65] ("GSE60361"), kidney[66] ("GSE119531"), lung[67] ("GSE127465"), pancreas[68] ("GSE84133"), and testis[69] ("GSE112393").

**Data simulations.** The simulated data were derived from the above 10 scRNA-seq datasets. We randomly divided the scRNA-seq data into two parts, one was treated as the single-cell reference (data_2) and the other (data_1) for the construction of bulk transcriptome data via aggregating all the single-cell gene expression profiles. Considering the cell composition of bulk RNA-seq data varies greatly in the natural situation, for each data_1, we further changed the cell-type proportion and synthesized 3 corresponding bulk transcriptome data with different cell compositions. In total, 30 paired simulated data were synthesized in this study. The detailed description of the experimental design was summarized in Supplementary Data 2.

**Compared methods.** **a**, Generative adversarial networks (GAN), **b**, conditional generative adversarial networks (CGAN). The generator consists of two fully connected layers, followed by LeakyRELU and RELU activations respectively. The discriminator is also made up of two fully connected layers, where the first layer is followed by a LeakyRELU activation. We use the Adam optimizer with the initial learning rate of 1e-4 and the betas parameters of 0.5 and 0.999. We train until the loss in the generative and discriminative phases is no longer reduced for 50 epochs, and use the model obtained at this time as the final model for prediction.

**Benchmark metrics.** Pearson correlation coefficient (PCC), Spearman's rank correlation coefficient (SRCC), and root mean squared error (RMSE) were used to assess the similarity of the gene expression profile between scRNA-seq data reference and synthetic bulk transcriptome data.

2. Simulated bulk data and the single-cell reference were generated from different datasets (unpaired simulations).

**Datasets.** Eight human pancreas scRNA-seq datasets from different resources were also applied to benchmark our method, including 1 CelSeq[70] ("GSE81076"), 1 CelSeq2[71] ("GSE85241"), 1 Fluidigm C1[72] ("GSE86469"), 4 inDrops[68] ("GSE84133"), and 1 SMART-Seq2[73] (E-MTAB-5061).

**Data simulations.** The simulated data were derived from the above 8 scRNA-seq datasets. We randomly selected one as the single-cell reference and another dataset from different resources as bulk transcriptome data. The detailed description of the experimental design was summarized in Supplementary Data 3.

**Compared methods. a**, Generative adversarial networks (GAN). **b**, conditional generative adversarial networks (CGAN). The model details were described above. **c**, bMIND[41]. We followed the guidelines on the bMIND GitHub repository: https://github.com/randel/MIND. We set the cell type proportion parameter "frac" to the output value of Bulk2Space to ensure the generated gene expression profile comparability.

**Benchmark metrics.** The benchmark metrics were described above.

## Evaluating the robustness of the deconvolution step of Bulk2Space

we evaluated the robustness of Bulk2space by introducing two kinds of noise: cell type noise and gene expression noise. For cell type noise, we randomly altered the type of cells with the ratio of 0.01, 0.02, 0.04, 0.08, 0.1, 0.2, 0.3, 0.4, 0.5, 0.7, and 0.9. For gene expression noise, we firstly constructed a noise expression profile $C$,

$$C \sim \mathrm{U}(-x, x) \tag{8}$$

Here, $x$ is the maximum gene expression value of scRNA-seq data reference.

We next constructed synthetic expression matrix by combining noise expression profile with scRNA-seq data reference, and the ratio of gene expression-altered cells remains 0.01, 0.02, 0.04, 0.08, 0.1, 0.2, 0.3, 0.4, 0.5, 0.7, and 0.9. PCC, SRCC and RMSE between the synthetic expression matrix and constructed bulk RNA-seq data were used to evaluate performance.

## Benchmarking the spatial mapping step of Bulk2Space on simulated datasets

1. Spatial barcoding-based data

**Datasets.** Eighteen datasets metioned above were applied to benchmark our method. 10 datasets were applied to construct single-cell data and spatial reference from same datasets, and other 8 datasets were applied to construct single-cell data and spatial reference from different datasets.

**Data simulations.** Starting from scRNA-seq data reference, we randomly chose 10 cells from it and aggregated their transcriptomic profiles as a spot of pseudo spatial transcriptomic data. The spot with over 25000 UMI counts would be sampled down to 20000 UMI counts to better meet the true situation. We also constructed pseudo spatial transcriptomic data with 100, 200, 500, 1000, and 5000 spot numbers to simulate true spatial transcriptomics data produced by different spatial barcoding technologies. Besides, the simulation that the scRNA-seq and spatial datasets are from different sources was also considered. The detailed description of the experimental design was summarized in Supplementary Data 4 (paired simulation) and Supplementary Data 5 (unpaired simulation).

**Compared methods. a**, Logistic Regression (LR). We implement L2 regularization as the additional penalty term to solve the problem of overfitting. We use L-BFGS algorithm as the solver, which uses the Hessian matrix to iteratively optimize the loss function. We fit the model according to the given training data and return the probability estimates on testing set. **b**, Decision Tree (DT). We use the Gini impurity to measure the quality of a split and choose the best split at each node. We don't set any maximum depth of the tree, so nodes are expanded until all leaves are pure or until all leaves contain less than 2 samples. We build the decision tree classifier from the training set, and then predict class probabilities of the input samples. **c**, Gradient Boosting Decision Tree (GBDT). We set the learning rate as 0.1 and the number of boosting stages to perform as 100. The loss function to be optimized is log loss function, and we choose Friedman MSE to measure the quality of a split. The maximum depth of the individual regression estimators is set to 3. We fit the gradient boosting model on the training set and use the trained model to make prediction on testing set. **d**, Multilayer Perceptron (MLP). We apply a two-layer perceptron, where each layer is followed by a batch normalization, a RELU activation, and a dropout layer with the probability of element zeroing as 0.1. We use the Adam optimizer with the initial learning rate of 1e-4 and the betas parameters of 0.9 and 0.999. We train until the loss is no longer reduced for 30 epochs, and use the model obtained at this time as the final model for prediction. **e**, DeepGBM. We set the number of tree groups to 100. The dimension of leaf embeddin for a tree group is set to 20. The structure of the distilled NN model is a fully connected networks with "100-100-100-50" hidden layers. We adopt the feature selection in each tree group, where we first sort the features according to the information gain, and the top 128 of them are selected as the inputs of distilled NN model. We use the Adam with weight decay (AdamW) optimizer with an initial learning rate as 2e-3, the decoupled weight decay as 1e-6, and Adam's betas parameters as 0.9 and 0.999. We train until the model reach the highest ROC-AUC score, and use it for prediction. **f**, RCTD[34]. We followed the guidelines on the RCTD GitHub repository: https://github.com/dmcable/spacexr. We set the parameter doublet_mode = 'full'. **g**, spatialDWLS[35]. We followed the guidelines on the spatialDWLS GitHub repository: https://github.com/rdong08/spatialDWLS_dataset/tree/main/codes. We set the parameter n_cell = 20. **h**, stereoscope[36]. We followed the guidelines on the stereoscope GitHub repository: https://github.com/almaan/stereoscope. We set the parameter sc epochs = 10000, st epochs = 10000. **i**, SPOTlight[37]. We followed the guidelines on the SPOTlight GitHub repository: https://github.com/MarcElosua/SPOTlight. We set the parameter n_cells = 75.

**Benchmark metrics.** Pearson correlation coefficient (PCC), Spearman's rank correlation coefficient (SRCC), and root mean squared error (RMSE) were used to assess the similarity of the gene expression profile per spot between the predict result and ground truth. Due to RCTD, spatialDWLS, stereoscope, and SPOTlight could not obtain gene expression profiles for each spot, we calculated gene signature matrix of each cell type based on single-cell reference, and the product of the predicted proportions of each cell type and the gene signature matrix of each cell type was regarded as the gene expression profile of each spot.

2. Image-based in situ hybridization data

**Datasets.** A mouse hypothalamus scRNA-seq data ("GSE113576") and MERFISH data (Bregma +0.26) (image-based reference) from the same laboratory[43] were applied to test the performance of the Bulk2Space algorithm for the second spatial mapping strategy. We also applied another mouse hypothalamus scRNA-seq data[44] ("GSE87544") from different source as single-cell reference to confirm the robustness of Bulk2Space.

**Benchmark metrics.** Pearson correlation coefficient (PCC) was used to assess the similarity of the gene expression profile between the predict result and ground truth. We took two strategies, five-fold cross validation and leave-one-out cross validation, to validate performance of Bulk2Space. In five-fold cross validation, we random split 150 target genes identified in MERFISH into five folds using R package caret (version 6.0−92), a split of 80% for reference and 20% for validation. In leave-one-out cross validation, 149 were used as reference genes and and one was used for validation.

## Performance evaluation of Bulk2Space using biological datasets

**Deconvolution performance evaluation using paired bulk and single-cell datasets.** Three paired mouse liver bulk and single-cell datasets[74] ("GSE119340") were applied to evaluate the performance of the deconvolution step of Bulk2Space.

**Reconstruction of mouse hippocampus subregions at single-cell resolution.** The mouse hippocampus Slide-seq v2[12] and scRNA-seq data[45] were downloaded from the Seurat website: https://satijalab.org/seurat/articles/spatial_vignette.html were applied to evaluate the performance of the spatial mapping step of Bulk2Space on real single-cell data and real spatial data. We down sampled Slide-seq v2 data to 5000 spots to speed the spatial mapping step. For Bulk2Space, we set the parameter $k = 2$ and top_marker_num = 100, all other parameter followed the default values. For Seurat, we followed the standard analysis workflow on the Seurat website and set the parameter resolution = 0.3 to perform unsupervised clustering.

**Robustness evaluation of Bulk2Space using repeated data.** We conducted 100-time repetitions on the deconvolution step of Bulk2-Space to evaluate the robustness of β-VAE using the human peripheral blood[46] ("GSE92495") scRNA-seq data. The construction of the single-cell reference and bulk transcriptome data was the same as mentioned above. We further evaluated the spatial mapping results using the pancreatic ductal adenocarcinoma (PDAC) dataset[47] and the human melanoma dataset[48,49] with 3-time repetitions. For Bulk2Space, all parameters followed the default values.

**Deconvolution of bulk RNA-seq data using annotation-free single-cell reference by Bulk2Space.** We further evaluated the performance of Bulk2Space without providing cell type information. We randomly divided the human peripheral blood[46] ("GSE92495") scRNA-seq data into two parts, one was treated as the single-cell reference and the other for the construction of bulk transcriptome data via aggregating all the single-cell gene expression profiles. For the single-cell reference, we followed the scRNA-seq data analysis workflow on the Seurat website: https://satijalab.org/seurat/articles/pbmc3k_tutorial.html to get 5 "clustering spaces" using "FindClusters" function with the parameter resolution = 0.2. For Bulk2Space, all parameters followed the default values.

**Revealing spatial, molecular, and functional heterogeneity of B cells in melanoma (consecutive slices).** The scRNA-seq data ("GSE72056") by Tirosh[48] et al. was used as the single-cell reference, and two consecutive slices of the ST data by Thrane[49] et al. were used as spatial reference (slice 1) and synthesized bulk data (slice 2) (via aggregating all the spots gene expression profiles), respectively. For Bulk2Space, we set the parameter epoch_num = 3500, $k = 10$, and top_marker_num = 500, all other parameters followed the default values.

We firstly evaluated the expression correlation of marker genes for five major cell types between generated single-cell expression profiles from synthesized bulk data (slice 2) and ground truth. We applied "FindAllMarkers" function of Seurat (version 4.0.4)[60] to calculate the marker genes of each cell type with the parameter

logfc.threshold = 0.5. We also evaluated the expression correlation of marker genes for five major cell types between spatial mapping result of Bulk2Space and spatial reference (slice 1).

For B cells spatial heterogeneity analysis, we determined two spatial areas (lymphoid area and tumor area) based on histological annotation firstly. Then, differentially expressed genes analysis between B cells from different areas was applied by the "FindAllMarkers" function with the parameter logfc.threshold = 0.25. For these differentially expressed genes, we further performed the pathway enrichment analysis using the Metascape (https://metascape.org) to investigate the biological functions.

**Linking histomorphology and transcriptomics in PDAC (discrete slices).** Bulk2Space was performed for another PDAC dataset[47]. We selected PDAC-B scRNA-seq data as the single-cell reference, and two inconsecutive slices of the ST data were used as spatial reference (PDAC-B ST1, slice 1) and synthesized bulk data (PDAC-B ST2, slice 2) (via aggregating all the spots gene expression profiles), respectively. For Bulk2Space, we set the parameter epoch_num = 3500, $k = 10$, and top_marker_num = 500, all other parameters followed the default values. The evaluation procedure of the marker gene expression correlation between Bulk2Space results and slices was same as it in the human melanoma dataset analysis.

## Application of Bulk2Space

**Bulk2Space integrates spatial gene expression and histomorphology in PDAC.** The pancreatic adjacent tissues and pancreatic cancer tissues bulk data by Wu[50] et al. were downloaded from "GSE171485". We firstly applied the PDAC-A scRNA-seq data by Moncada[47] et al. as the single-cell reference to deconvolute pancreatic adjacent tissues bulk data and pancreatic cancer tissues bulk data respectively. We then applied the PDAC-A ST1 by Moncada[47] et al. as the spatial reference and performed spatial mapping step for the generated single-cell gene profiles from pancreatic cancer tissues bulk data. For Bulk2Space, we set the parameter epoch_num = 3500, $k = 10$, and top_marker_num = 200, all other parameters followed the default values.

**Reconstruction of mouse isocortex layers at single-cell resolution using Spatial-seq data.** The mouse isocortex bulk data was produced by Spatial-seq (see Methods). We applied a mouse primary visual cortex scRNA-seq data[52] and two tissue sections (anterior section 1 and posterior section 1) sequenced by 10X Visium as the single-cell reference and spatial references, respectively. For Bulk2Space, we set the parameter epoch_num = 3500 and top_marker_num = 300, all other parameters followed the default values.

**Reconstruction of mouse hypothalamus structure at single-cell resolution using Spatial-seq data.** The mouse hypothalamus bulk data was produced by Spatial-seq (see Methods). We applied a mouse hypothalamus scRNA-seq data ("GSE113576") and MERFISH data (Bregma −0.04) by Moffitt[43] et al. as the single-cell reference and spatial reference, respectively. For Bulk2Space, we set the parameter epoch_num = 3500 and top_marker_num = 500, all other parameters followed the default values. For those ambiguous cells in MERFISH data, we re-annotated their cell type based on the whole gene expression profiles predicted by Bulk2Space.

**Uncovering spatial gene expression dynamics in inflammation-induced prostate cancer.** The human prostate cancer bulk data was downloaded from TCGA (https://portal.gdc.cancer.gov). We selected the human prostate cancer scRNA-seq data by Wu[54] et al. as the single-cell reference, and three ST data from different tissue section location as spatial references (p1.1 Normal glands, p2.3 Inflammation, p3.3 Cancer Gs 3+4). For Bulk2Space, we set the parameter epoch_num = 3500, $k = 10$, and top_marker_num = 500, all other parameters followed

the default values. We then compared the distribution of various cell types in these three different tissue regions.

## Spatial-seq protocol

**Multiplexed spatial barcoding.** A virtual 48×48×48 three-dimensional well array (barcoding array) was constructed in the X, Y, and Z directions, with 48×48 wells in each layer. 384-well plates (24×16 wells) were selected as the basic units (2×3 plates) of each layer. Each well in the barcoding array can be labeled as barcode ($\{X_i,Y_j,Z_k\},i,j,k \in N,[1,48]$). Magnetic beads coated with carboxyl group (Cat. # 40200, purchased from BEAVER Biomedical Engineering Co., Ltd.) were distributed into each well for barcode synthesis. There were three rounds of barcode extension reactions. In the first round, 5' amino-modified barcoded oligonucleotides (barcode $X_i$) were conjugated to the beads. In the second round of extension reaction, barcoded oligonucleotides (barcode $Y_j$) which had a sequence of bases at the 5' end complementary to the 3' end of barcode $X_i$ were linked to the end of barcode $X_i$ by PCR. Similarly, in the third round of reaction, barcode sequences containing a unique molecular identifier (UMI) and a polyT tail (barcode $Z_k$) were extended to barcode $Y_j$ by PCR. All oligonucleotides were purchased from Sangon Biotech (Shanghai) Co., Ltd. This method only needs 144 (48×3) different barcodes to theoretically generate magnetic beads with $48^3$ distinct kinds of magnetic beads with known sequences. More importantly, the entire synthesis process was serialized and multiplexed, which can be accomplished with the help of arrayed pipettes or liquid workstations, rather than synthesizing the magnetic beads well by well. The spatially barcoded beads can be used in many areas, such as multiplexed labeling of tissue samples or even single cells.

**Collection of mouse brain regions.** Three wild-type adult C57BL/6 J mice (SPF, male, 20–25 g) aged 8–10 weeks were used for Spatial-seq experiments. The environmental conditions in the mouse facility were: 12 h light and 12 h dark cycle (light on from 8:00 a.m. to 8:00 p.m.), light intensity range of 15–20 lux, temperature range of 22–26 °C, humidity range of 40–70%, and free access to food and water. The use and care of the mice were in accordance with the guidelines of the Animal Advisory Committee of Zhejiang University and the US National Institutes of Health Guidelines for the Care and Use of Laboratory Animals. All procedures were approved by the Animal Advisory Committee of Zhejiang University. The mouse brain was sliced into 14-µm sections from the coronal and sagittal directions. Each tissue slice was registered to a reference brain template provided by the Allen Brain Atlas (https://portal.brain-map.org/). After spatial registration, the anatomical regions of the mouse brain were delineated and annotated. The outlines, dissection sequences, and collectors of the brain regions were specified from the annotated data and then imported to an LCM instrument (Laser Microdissection System equipped with a DM68 microscope, a Leica LMD6 laser cutter, and a single-cell capture collector LMT350, Leica Microsystems, Germany). LCM is a microscope-guided powerful cutting system incorporating UV light for contact- and contamination-free isolation of areas of interest from tissue sections[15]. Brain regions were dissected from the tissue by LCM according to the imported files and fell into the collector loaded with barcoded beads in advance. Spatial-seq has the potential to achieve spatially resolved whole transcriptome sequencing of a large number of single cells.

**Multiplex RNA-seq.** The tissue was lysed in the barcoded well, allowing mRNA to be captured by polyT tail on the surface of the magnetic beads. The lysis buffer was prepared with Tris-HCl (120 µL, pH 7.5, Cat. # T1140, purchased from Beijing Solarbio Science & Technology Co., Ltd.), LiCl (80 µL, Cat. # AM9480, purchased from Invitrogen), 10% SDS solution (120 µL, RNase-free, Cat. # AM9823, purchased from Invitrogen), EDTA (16 µL, Cat. # ST066, Shanghai beyotime Biological Co.,Ltd.), 0.5 M DTT solution (16 µL, DNase, RNase & Protease free, Cat.

\# ST041, Shanghai beyotime Biological Co.,Ltd.), and water (852 µL, nuclease-free, Cat. # AM9930, Ambion). Each well was loaded with lysis buffer. The tissue was lysed on ice for 12 min. The beads were then washed and transferred for reverse transcription (RT), The captured mRNA was reverse transcribed using PrimeScript II Reverse Transcriptase (Cat. # 2690A, Takara).to construct cDNA libraries following the guidance of Illumina Nextera XT DNA Library Preparation Kits. A paired-end sequencing was conducted to decode the spatial barcode in the 3' end and detect RNA species in the 5' end of the cDNA.

**Sequence alignment.** FastQC was utilized for quality control of the RNA-seq data (http://www.bioinformatics.babraham.ac.uk/projects/fastqc/). The derived RNA-seq data were fragmented into lots of files based on the spatial barcodes. The digital gene expression matrix for each barcode was retrieved follow the Drop-seq sequence alignment cookbook.

**Data normalization.** FPKM, RPKM, RPM, and TPM were calculated from the gene expression matrix according to the following formulas.

$$FPKM = \frac{ExonMappedFragments \times 10^9}{TotalMappedFragments \times ExonLength} \tag{9}$$

$$RPKM = \frac{(ExonMappedRead/TotalMappedReads \times 10^6) \times 10^3}{ExonLength} \tag{10}$$

$$RPM = \frac{ExonMappedReads \times 10^6}{TotalMappedReads} \tag{11}$$

$$TPM = \frac{N_i/L_i \times 10^6}{sum(N_1/L_1 + N_2/L_2 + \cdots + N_n/L_n)} \tag{12}$$

The data were normalized using the global-scaling normalization method "LogNormalize" in Seurat. A Bayesian network-based method termed ComBat was used for batch effect removal of data derived from different cDNA libraries. The expression matrix of each sample was normalized using the "Deconvolution Normalization" algorithm to align the median gene expression.

**Ethical statement.** All experiments were approved by and conducted in accordance with the ethical guidelines of the Zhejiang University Animal Experimentation Committee (Protocol number, 14875).

## Statistics and reproducibility

In this study, 152 simulated datasets (the data points of paired simulations for deconvolution, unpaired simulations for deconvolution, paired simulations for spatial mapping, and unpaired simulations for spatial mapping are 30, 12, 50, and 60, respectively), 13 biological datasets, and 2 experimental datasets were used to evaluate the Bulk2Space algorithm. Unannotated, ambiguous, or low quality-cells were excluded from the analysis. Pearson correlation coefficient, Spearman's rank correlation coefficient, and root mean squared error were used to compare the performance between different methods. For paired data simulations, single cells were randomly selected to synthesize the reference and the test set. For unpaired data simulations, single-cell profiles were randomly selected to synthesize the reference and the test set. The Investigators were not blinded to allocation during experiments and outcome assessment.

## Reporting summary

Further information on research design is available in the Nature Research Reporting Summary linked to this article.

## Data availability

The original data used in this paper can be accessed through the following links: (1) single-cell RNA-seq data of the human blood: GEO accession: "GSE92495"[46]; (2) single-cell RNA-seq data of the human brain: GEO accession: "GSE103723"[61]; (3) single-cell RNA-seq data of the human kidney: GEO accession: "GSE121862"[62]; (4) single-cell RNA-seq data of the human liver: GEO accession: "GSE124395"[63]; (5) single-cell RNA-seq data of the human lung: GEO accession: "GSE130148"[64]; (6) single-cell RNA-seq data of the mouse brain: GEO accession: "GSE60361"[65]; (7) single-cell RNA-seq data of the mouse kidney: GEO accession: "GSE119531"[66]; (8) single-cell RNA-seq data of the mouse lung: GEO accession: "GSE127465"[67]; (9) single-cell RNA-seq data of the mouse pancreas: GEO accession: "GSE84133"[68]; (10) single-cell RNA-seq data of the mouse testis: GEO accession: "GSE112393"[69]; (11) single-cell RNA-seq data of the human pancreas with 975 cells: GEO accession: "GSE81076"[70]; (12) single-cell RNA-seq data of the human pancreas with 2133 cells: GEO accession: "GSE85241"[71]; (13) single-cell RNA-seq data of the human pancreas with 597 cells: GEO accession: "GSE86469"[72]; (14) four sets of single-cell RNA-seq data of the human pancreas with 1635, 1562, 3330, 1230 cells, respectively: GEO accession: "GSE84133"[68]; (15) single-cell RNA-seq data of the human pancreas with 2288 cells[73] (https://www.ebi.ac.uk/arrayexpress/experiments/E-MTAB-5061/); (16) single-cell RNA-seq data of the mouse hypothalamus using 10X Genomics: GEO accession: "GSE113576"[43]; (17) single-cell RNA-seq data of the mouse hypothalamus using Drop-seq: "GSE87544"[44]; (18) three sets of single-cell RNA-seq data of the mice liver: GEO accession: "GSE119340"[42]; (19) single-cell RNA-seq data of the mouse hippocampus region[45] (https://www.dropbox.com/s/cs6pii5my4p3ke3/mouse_hippocampus_reference.rds?dl=0); (20) two sets of single-cell RNA-seq data of the human pancreatic ductal adenocarcinoma (PDAC): GEO accession: "GSE111672"[47]; (21) single-cell RNA-seq data of the human melanoma: GEO accession: "GSE72056"; (22) single-cell RNA-seq data of the mouse cortex region[52] (https://www.dropbox.com/s/dl/cuowvm4vrf65pvq/allen_cortex.rds); (23) single-cell RNA-seq data of the human prostate cancer[54] (https://singlecell.broadinstitute.org/single_cell/study/SCP1415); (24) MERFISH data of the mouse hypothalamic preoptic region at bregma 0.26[43] (https://datadryad.org/stash/dataset/doi:10.5061/dryad.8t8s248); (25) slide-seq v2 data of the mouse hippocampus region[12] (https://singlecell.broadinstitute.org/single_cell/study/SCP815/highly-sensitive-spatial-transcriptomics-at-near-cellular-resolution-with-slide-seqv2); (26) three sets of spatially resolved transcriptomics data of the human PDAC using "Spatial Transcriptomics": GEO accession: "GSE111672"[47]; (27) two sets of spatially resolved transcriptomics data of the human melanoma using "Spatial Transcriptomics"[49] (https://www.spatialresearch.org/resources-published-datasets/doi-10-1158-0008-5472-can-18-0747/); (28) two sets of 10X Visium data of the mouse anterior cortex and posterior cortex regions (https://www.10xgenomics.com/cn/resources/datasets/mouse-brain-serial-section-1-sagittal-anterior-1-standard-1-1-0); (29) two sets of spatially resolved transcriptomics data of the human prostate cancer using "Spatial Transcriptomics"[55] (https://www.spatialresearch.org/resources-published-datasets/10-1038-s41467-018-04724-5/); (30) three sets of bulk RNA-seq data of the mice liver: GEO accession: "GSE119340"[42]; (31) two sets of bulk RNA-seq data of the human PDAC: GEO accession: "GSE171485"[50]; (32) bulk RNA-seq data of the human prostate cancer (https://portal.gdc.cancer.gov). The bulk RNA-seq data of the mouse cortex and mouse hypothalamus regions reported in this manuscript using our in-house developed Spatial-seq have been deposited to the Gene Expression Omnibus under accession number "GSE192999". All other relevant data supporting the key findings of this study are available within the article and its Supplementary Information files or from the corresponding author upon reasonable request. Source data are provided with this paper.

## Code availability

Bulk2Space is available as a python package and the source code is deposited on GitHub (https://github.com/ZJUFanLab/bulk2space)[75].

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

## Acknowledgements

This work is supported by the National Natural Science Foundation of China (81973701, X.F.; 91846204, H.C.; and U19B2027, H.C.), the Natural Science Foundation of Zhejiang Province (LZ20H290002, X.F.), the Innovation Team and Talents Cultivation Program of the National Administration of Traditional Chinese Medicine (ZYYCXTD-D-202002, X.F.; ZYYCXTD-D-202207, Y.G.), the Westlake Laboratory (Westlake Laboratory of Life Sciences and Biomedicine), and the Alibaba Cloud (Alibaba Cloud Computing Co. Ltd.).

## Author contributions

X.F., H.C., Y.G., and J.L. conceived the study. J.L. and J.Q. collected datasets involved in this article, benchmarked all methods, and participated in the development of the Bulk2Space algorithm. Y.F., Z.C., and X.Z. implemented the code and interface to the Bulk2Space algorithm. N.Z., X.S., H.Y., P.Y., J.C., L.S., and J.Z. provided a lot of advice on algorithm implementation and biological applications. J.L., Y.H. (Yang Hu), Y.H.(Yining Hu), L.Y., and X.L. conducted the multiplexed RNA-seq experiment using Spatial-seq. D.W. provided important advice on brain tissue isolation, spatial registration, and annotation. All authors wrote the manuscript, and read and approved the final manuscript.

## Competing interests

The authors declare no competing interests.
