## [Peer Review File · Nature Communications]

Reviewers' comments:

Reviewer #1 (Remarks to the Author):

Liao et al. present a new tool "Bulk2Space" that can simultaneously reveal the spatial and cellular heterogeneity of bulk RNA-seq data. Their method is novel, and the authors make some attempts on simulation validations. The real data analysis looks promising. The manuscript is clear-written, and the research question is interesting. However, I have a few concerns with the simulations and assumptions about the method. I have provided more details in my comments below.

1. The method, Bulk2Space is a 2-step process of resolving cell type proportions and spatial locations for bulk tissues. The first step is deconvolution of bulk transcriptome data and generation of single cell data, and the authors claim that they can resolve/regenerate bulk data on cell level.

a. The generated single cell data is randomly simulated from beta-VAE (page6, line 123) with the constraint of cell type abundance. Because of that, we should observe different generated data if we repeat first step of Bulk2Space even with the same single-cell reference and cell type abundance constraint. Have you conducted repetitions on step1? Do all repetitions provide similar spatial results? How does the randomness from beta-VAE affect the spatial results?

b. There exist methods that estimates cell-type-specific expressions on subject/sample level and have good results, such as bMIND (bMIND: Wang, Jiebiao, Kathryn Roeder, and Bernie Devlin. "Bayesian estimation of cell type-specific gene expression with prior derived from single-cell data." *Genome Research* (2021) 31: 1807-1818.) Have you tried to compare the cell-type aggregated generated single cell data with the estimation results from bMIND?

c. This is about the robustness in simulation setting of step1. Bulk data are synthesized from the same single cell resource as for single-cell reference. (1) Do you have any evidence of robustness if we use two different resources of single cell data for the same type of tissue? In practice, the bulk tissue and the single-cell reference are from two different resources. At least, we can also try to change the drop-out rate for single-cell reference data. (2) How much does this procedure rely on single-cell reference, such as sequencing depth?

2. The second step of Bulk2Space is to map generated single cells to spatial locations. The authors discussed two spatial resolved transcriptomic references: spatial barcoding transcriptomics and image-based in situ transcriptomics and assigned generated single cells to spatial coordinates.

a. For spatial barcoding transcriptomics reference, I'm glad that the author can use deep-Forest to assign cell compositions of spatial data with high accuracy. It is not easy problem even using real single-cell reference for spatial data cross platform, for example RCTD (Cable, D.M., Murray, E., Zou, L.S. et al. Robust decomposition of cell type mixtures in spatial transcriptomics. *Nat Biotechnol* (2021).

<https://doi.org/10.1038/s41587-021-00830-w>). Have you considered to validate your accuracy with real single-cell data and real spatial data?

Reviewer #2 (Remarks to the Author):

I. Summary

Liao and colleagues present Bulk2Space, a Python-based deep learning framework that attempts to map bulk RNA-seq data to spatially-resolved transcriptional information with single-cell resolution. The authors use the generative properties of variational autoencoders (VAEs) to synthetically generate single-cell RNAseq data from bulk RNAseq data, based on the assumptions that bulk RNAseq can be approximated as a weighted linear combination of single-cell measurements and that it maps to a latent clustering space of cells, modeling by the VAE. Then, taking the generated output of the VAE, the authors define a method for mapping the generated single cell data to spatial coordinates, relying on some spatial reference map, provided either by a barcode-based spatial transcriptomics or probe-based in situ hybridization (e.g., FISH). They demonstrate their method on several applications, where they map estimated or existing bulk RNAseq data to spatial information based on some type of spatial reference. The code for this tool is open-sourced as a Python package.

While the methodology and architecture proposed is sound, I have major concerns related to the training and evaluation of Bulk2Space. These fundamental machine learning concerns, outlined below, obstruct complete assessment of the soundness and quality of their empirical results.

II. Major Comments

A. Training data and methodology for Bulk2Space

The manuscript, Methods, and results are devoid of any information about how the Bulk2Space machine learning models were trained and validated. These are critical details to understanding the functionality of the method and to evaluating its performance. Without this information, it is not possible to properly assess their empirical results. The authors must provide information on the training data (e.g., dataset identity, size, distribution); validation procedure (e.g., early stopping, etc); and general training methodology (e.g., pre-training with fine-tuning, re-training on each new dataset or task). This information must be presented succinctly within the Main Text; described in detail in the Methods and Supplement; and supported with concrete results (e.g., evolution of loss on training and validation sets)

in the Main Figures and/or Supplementary Figures. Below I enumerate some specific questions and concerns which must be addressed:

1. What data was used to train these models? What is the breakdown of the data between training and validation?
2. Was the same trained model used for all analyses and results presented in the paper?
3. Was the model pre-trained on a large, diverse collection of data, and then fine-tuned for each task? Or not fine-tuned? Alternatively, must Bulk2Space be re-trained from scratch on new data for each new application/task?
4. What is the nature of the latent space learned by the beta-VAE? Could the authors provide perturbation analyses to probe the latent space? Similarly, what is the dimensionality of the latent space learned by the beta-VAE?
5. Is there any data pre-processing prior to training the beta-VAE?
6. What is the relative weighting of the reconstruction loss and regularization loss in training the beta-VAE?
7. What hardware was used to train the models? For how many epochs?
8. Was any hyperparameter optimization employed?

These are details critical to machine learning work that are altogether missing from this manuscript. Please address.

B. Model architectures and details

Similarly, the manuscript is lacking information on the specifics of the beta-VAE model architecture and its parameters. For example:

1. What is the model architecture? CNN-based? Multi-layer perceptron/fully-connected?
2. How many layers, neurons are used in the encoder/decoder networks?
3. What is the dimensionality of the latent space?

And so on. I was unable to find details on this information in the manuscript/Sl text. We can make some inferences about this from looking at the code; however, it is not the expectation to have to look at code to derive this information. Please provide this information.

C. Model evaluation and data ambiguities

Continuing in this line, there remain ambiguities about the model evaluation procedure and the data for model evaluation. Some specific concerns:

1. The authors reference 30 single-cell transcriptomic datasets early in the manuscript. However, in the Methods, there is mention in Line 442: “Ten scRNA-seq were applied...”. Please clarify whether 30 single-cell datasets were used, or 10, and in either case, what the nature and references for these datasets are.
2. The descriptions of the dataset sourcing are generally insufficient. Namely, there are missing references – what studies did these datasets come from? Furthermore, for the spatial data, the authors state in Line 451-452, “the human PDAC, human melanoma... were used for reference and validation”. How were these data generated, i.e., what spatial transcriptomic technique? If pulled from literature, what are the references and accession numbers for these datasets?
3. Line 183-184: “A comprehensive comparison of six machine learning or deep learning approaches was then conducted using 40 datasets” – what are these 40 datasets? What is their size? What are the benchmark approaches tested? What are the details of those architectures; were all hyperparameter optimized and trained on the same datasets? What was the procedure for evaluation? Further analyses and discussion are required.

D. Model generalizability and “test set” performance

There are additionally concerns about the Bulk2Space model’s performance on a truly independent “test” set and its generalizability to other specimens/datasets. Namely, I have concerns about the “Slice 1” and “Slice 2” analyses presented:

1. The analyses in Figures 3 and 4 describe use of both “Slice 1” and “Slice 2”, which seem to be consecutive sections from the same tissue, wherein inference evaluations on Slice 1 are done based on deconvolution of bulk data generated from Slice 2. If these two slices are derived from the same specimen, and especially as consecutive sections, they will inherently be highly correlated, and therefore will bias the “test set” performance. What was the spatial distance between the two slices, i.e., were they direct consecutive sections? Could the authors provide H&E of the two slices? More importantly, what about performance and generalization to an independent “test” specimen? This is as present a major consideration and limitation of the work.

2. The authors should evaluate their method on multiple independent “test” tissues/specimens after training is complete (see comments above regarding model training). These evaluations are critical to assessing the performance and generalizability of the method.

3. In addition, the authors perform an analysis wherein a single-cell dataset is split into a “reference” subset and a “bulk data synthesis” subset. They then aggregate the “bulk synthesis” subset to generate pseudo-bulk transcriptome data, and then evaluate correlation between the generated and true single-cell data on a held-out set. With this setup, questions about generalizability remain. What about training on data from studies that have independent, paired bulk and single-cell data? Evaluation on these datasets will help verify the robustness of the method.

4. Similarly, what about analyses when the scRNAseq and spatial datasets are decoupled, i.e., not from the same tissue specimen or source? This evaluation is also requested to verify robustness and generalizability.

E. Other major comments

1. The authors describe the Spatial-seq method to generate new spatial transcriptomics data and evaluate Bulk2Space’s performance using Spatial-seq data as a reference. When deployed for the same application or on the same tissue type, how well does the method generalize to other spatial transcriptomic methods for generating the spatial reference? Greater discussion on modularity and generalizability in this respect is warranted.

2. When describing the generated single-cell reconstructions, the authors often use the language “deconvolved single-cell profiles” or “after deconvolution, input bulk data is transformed into single-cell transcriptomics”. This language is a bit misleading, as it gives the impression that these outputs are reflective of real data, when in truth the output data is a generative reconstruction. There are certain instances when the authors correctly state “generated single cells”; however, this is not consistent. Please correct.

3. It is not clear that the requirement for a spatial reference will ultimately be needed. In training, it seems that the spatial reference supervises a mapping from reconstructed single-cell output (after bulk deconvolution) to spatial coordinates. It seems then that after training, it would be possible to go directly from bulk RNA-seq to spatial coordinates without a reference but with fully trained models. Could the authors please clarify this point? Analyses without spatial reference would significantly improve the broad utility of the method.

4. Ideally in deployment, one would want to use Bulk2Space in a new setting where they may not have information on the cell types present. Namely, one could envision performing the analysis and spatial mapping and then annotating the cell types. The package should support this ability and support the option to not provide marker genes as input.

III. Minor Comments

1. Grammatical and language errors can be found throughout the text. Please correct these in revision. Two examples are included here. First, line 58-59: “The in silico method has great potential to predict spatial heterogeneity from bulk RNA-seq data at single-cell resolution by integrating cutting-edge technologies”, should be corrected to something like “In silico methods have the potential to...”. Second, line 62: “The emerging of an approach that can efficiently convert...”. Please correct grammatical issues. This will help improve the readability of the paper.
2. Figure 1b: In the “Single-cell reference” sub-region, there are two “C4” annotations, when it seems that one should be labeled C2. Please correct. Additionally, if the organization numbers (1)-(6) are not referenced in the figure caption, they should be eliminated; otherwise, they may be referenced in the figure caption.
3. Figure 2: please clarify what tissue type(s) are used for the analyses in subpanels (d)-(i).
4. Line 237: The use of “good correlation” is inexact and nonspecific. What is a good correlation? Please provide the value of the correlation coefficient and an associated metric of statistical significance, both here and elsewhere throughout the text.
5. Line 368-372: “The Bulk2Space results indicated that...CAF_s accumulated in normal glands at the early stage...promoting the occurrence of local inflammation, a fundamental innate immune response to perturbed tissue homeostasis, thus leading to tissue cancerization.” This is an overstated claim and an over-inflated interpretation of the results. The authors have performed a correlative analysis, but have not proven anything about the causal source or mechanisms of tissue cancerization in this context. Please temper the claims.
6. Line 411: “Bulk2Space can assign single cells to optimal spots, and these mixtures are split into collections of individual cells, thus achieving single-cell resolution.” The claim to have achieved single-cell resolution is overstated. Bulk2Space describes a generative process that involves the generation of synthetic data (in converting bulk RNA-seq to estimate single-cell profiles); based on this the resulting assignment to spatial coordinates is simply a prediction – the statement of achieving single-cell resolution implies some sort of measurement, which is not the case here. Please correct.
7. Line 412-415: “...uncovering the molecular mechanism underlying the progression of the disease”. Like the comment regard Line 368-372, this is an overstated claim. The authors have not proven anything about the causal source or mechanisms of tumor progression in this context; please temper the claims.
8. Code: The code needs to be cleaned and better commented. To promote use by the community, there should be more extensive documentation and explanation such that people will know how to use it. In particular, the `bulk2space.py` file is a single script – there are no other functions provided, and it is not well commented – this is not modular and usable coding practice. Please improve the clarity, readability, and usability of the codebase in order to promote its use by the community.

Reviewer #3 (Remarks to the Author):

The authors present a deep-learning method bulk2space which aims at deconvoluting bulk RNA-sequencing data in cellular and spatial resolution. This research addresses an area of unmet need, however more evidence and clarification are required to support the proposed method.

Major comments

- As a central part of the work, the description of the deconvolution procedure (Fig 1b, [117-125]) is rather succinct and it is current from difficult to follow. It would benefit for more explanation.
- How does bulk2space methods of deconvoluting spatial barcoding methods compare to some already published methods such as Stereoscope, SpatialDWLS or Spotlight?
- To confirm the robustness of bulk2space, it would be ideal to see how for the second spatial mapping strategy (MERFISH) bulk2space performs when spatial data deconvolution is carried out when using scRNA-Seq data from a similar tissue from a separate experiment.
- In validation of MERFISH data with reference genes, a split of 80% for reference and 20% for validation would help further validate performance of the algorithm.
- Performance of Bulk2Space should be further verified using bulk transcriptomics data not generated from the reference scRNA-Seq data but from a separate bulk transcriptomics experiment, as this will more closely represent how users will employ the algorithm.
- Availability of the algorithm: could the authors provide a detailed protocol for installation including necessary dependencies. We have manage to run bulk2Space, not without difficulties:
 - ☐ We tried to set up conda environment on two separate systems with GPUs to run demo data, setting up the conda environment failed on both systems. Further information on the system the conda environment was tested on would be beneficial, to see if it could be reproduced by users. Additionally, if programme is able to be run without GPUs, that option should be made more explicit to broaden the scope of potential users and make tool more accessible.
 - ☐ More information should be provided on the Github page on the programmes parameters.

☒ Additional specifications required on the computing power required to run the programme and GPUs/CUDA system it is compatible with. Bulk2Space currently only works on certain Linux servers with specific CUDA version support which is not mentioned in the paper or the Github. This should be made more explicit. More relaxed constraints would be ideal as not all users will have access to a similar environment.

☒ Test run using provided demo based on spatial-barcoding data ran successfully on one of the systems tested. Additional documentation should be provided on Github page with regards to the significance of the output files to give the user a clearer understanding of what they correspond to.

Minor comments

- The value of Bulk2Space in being able to deconvolve multiple RNA-Seq datasets when a single scRNA-Seq and spatial transcriptomics reference are provided needs to be stated more explicitly in the abstract and throughout the main body of the paper. The submitted abstract reads as Bulk2Space would be a deep learning framework that allows spatial deconvolution of bulk data without the necessary references required.
- For instance, in line 85: ‘prevailing spatially resolved transcriptomics methods can provide reference coordinates for single cells’ as an explanation for their method yet in their introduction the authors state that they will incorporate data from spatial methods that have not achieved single-cell resolution. Keeping consistent across spatial methods will make the method easier to follow for readers.
- Check grammar on paragraphs 58-64, eg “the emerging of an approach”
- Sentence in line 68 could be made clearer.
- Fig 2b color for generated and ground truth data is hard to distinguish
- In figure 2F it is difficult to distinguish from this visualisation the exact similarity and differences between the ground truth and bulk2space generated data.
- Figure 3D histological annotation needs to be clearly – difficult to see contrasting colours on top of H&E

- In Figure 3c and 3d the colours chosen for cancer cells and erythrocytes need to be more contrasting as it is currently challenging to distinguish between them in the figures.
- In figure 3G the highest correlation between expression of markers is ~ 0.5 , an explanation for this result should be offered.
- Explanation of how histological annotations were performed is needed.
- In Figures 4B and 4E the expression correlation of marker genes between bulk2space and the slices is not particularly strong, could an explanation be offered for this?
- Figure 4G legends need quantitative value instead of just 'high'.
- Figure 6B and 6E B cells, monocytes and NK cells need to be in more contrasting colours to be able to more easily distinguish between them.
- It would be ideal if all t-SNE plots in paper should be updated to UMAPs.

A point-by-point response to reviewers' comments

First and foremost

We have studied the reviewers' comments carefully and fully agree with their constructive suggestions. According to their comments, we have made thorough revisions which were marked in blue in our resubmitted manuscript around the further evaluation and validation of the performance, comparison with other published approaches, and providing additional methodological details. We have tried our best to answer the concerns raised by reviewers point by point. Compared with the original version, the revised manuscript and supplementary information contain more than 10000 more words, 16 more figures, 42 more references, and 5 more supplementary data. We hope this edition will satisfy the reviewers and address all the concerns to win their approval for the publication of our paper.

Reviewer #1 (Remarks to the Author):

1. The first step, Deconvolution.

a. The generated single cell data is randomly simulated from beta-VAE (page6, line 123) with the constraint of cell type abundance. Because of that, we should observe different generated data if we repeat first step of Bulk2Space even with the same single-cell reference and cell type abundance constraint. Have you conducted repetitions on step1? Do all repetitions provide similar spatial results? How does the randomness from beta-VAE affect the spatial results?

Response: Thanks for your suggestion, we agree that repeating the simulation procedure could help evaluate the performance and robustness of Bulk2Space. We conducted 100 repetitions on the deconvolution step of Bulk2Space to evaluate the robustness of β -VAE using the human peripheral blood scRNA-seq data. As shown in Figure 1, the single-cell generation remained highly robust across 100 repetitions of β -VAE, with the pairwise correlations of generated single-cell profiles between replicates above 0.998 (Figure 1a). For each replicate, the averaged correlation of the generated single-cell data with other replicates remained consistent. With the robust performance of the single-cell generation in the deconvolution step, we further investigated whether the randomness of the single-cell generation would affect the spatial mapping results. We performed Bulk2Space to map single cells generated from three replicates of bulk PDAC and melanoma bulk data using the same references. As

shown in Figure 2 and Figure 3, the spatial distribution and composition of cell types in the three repeated experiments were consistent, suggesting a robust performance of Bulk2Space.

In conclusion, the repetition test suggested that the randomness of the single-cell generation function in Bulk2Space would affect the deconvolution and spatial mapping results to a small extent. However, although the single-cell data generated each time were slightly different, the overall prediction results showed robust performance of Bulk2Space in the spatial distribution of cell types, the cell-type composition and proportion in spots, and the spatial patterns of gene expression. The results have been included in the supplementary materials.

Figure 1. Spatial deconvolution of pancreatic ductal adenocarcinoma (PDAC) data by

Bulk2Space using 3 repetitions. a, Spatial deconvolution of the three replicates of bulk PDAC data. Each spot represented the composition and proportion of cell types. **b**, Spatial expression patterns of cell-type-specific marker genes for cancer cells, endothelial cells, and ductal cells.

Figure 2. Spatial deconvolution of pancreatic ductal adenocarcinoma (PDAC) data by Bulk2Space using 3 repetitions. a, Spatial deconvolution of the three replicates of bulk PDAC data. Each spot represented the composition and proportion of cell types. **b**, Spatial expression patterns of cell-type-specific marker genes for cancer cells, endothelial cells, and ductal cells.

Figure 3. Spatial deconvolution of melanoma data by Bulk2Space using 3 repetitions. **a**, Spatial deconvolution of the three replicates of bulk melanoma data. Each spot represented the composition and proportion of cell types. **b**, Spatial expression patterns of cell-type-specific marker genes for B cells, cancer-associated fibroblasts (CAFs), and malignant cells.

b. There exist methods that estimates cell-type-specific expressions on subject/sample level and have good results, such as bMIND (bMIND: Wang, Jiebiao, Kathryn Roeder, and Bernie Devlin. "Bayesian estimation of cell type-specific gene expression with prior derived from single-cell data." *Genome Research* (2021) 31: 1807-1818.) Have you tried to compare the cell-type aggregated generated single cell data with the estimation results from bMIND?

Response: Thanks for your valuable suggestion. We have compared the performance of Bulk2Space with bMIND in our revised version. Because other deconvolution

methods, such as CPM, CIBERSORT, and ImmuCC, can only predict cell-type proportions instead of gene expression of generated data, we compared Bulk2Space with GAN, CGAN, and bMIND. As illustrated in Figure 4a, the gene expression correlation between the generated single-cell data and input bulk data was calculated to evaluate and compare the four candidate algorithms. Bulk2Space outperformed the other three methods by the *Pearson* correlation of gene expression and the gene expression variation (root mean squared error, RMSE). We found that bMIND showed a higher *Spearman* correlation of gene expression than Bulk2Space, GAN, and CGAN. Then we compared the correlations of marker gene expression for different cell types between four generations and the ground truth (Figure 4b). The results suggested that single cells generated by Bulk2Space and CGAN had higher correlations than GAN and bMIND in gene expression between generated data and the ground truth. The results have been included in the supplementary materials.

c. This is about the robustness in simulation setting of step1. Bulk data are synthesized from the same single cell resource as for single-cell reference. (1) Do you have any evidence of robustness if we use two different resources of single cell data for the same type of tissue? In practice, the bulk tissue and the single-cell reference are from two different resources. At least, we can also try to change the drop-out rate for single-cell reference data. (2) How much does this procedure rely on single-cell reference, such as sequencing depth?

Response: Thanks for your comments. We have systematically evaluated the robustness of the deconvolution of Bulk2Space using 8 separated datasets derived from the human pancreas and sequenced by different platforms. Since the unpaired simulation data were derived from different platforms. We discuss whether the quality, for instance, sequencing depth, of the reference data could affect the deconvolution of Bulk2Space. Compared with paired simulation, the overall correlation of gene expression between generated data and ground truth decreased because of batch effects. However, when using unpaired simulations, as shown in Figure 4, the comparison between 12 simulations remained robust, suggesting that Bulk2Space was not so dependent on the single-cell reference, although the quality of the single-cell reference directly affected the characterization of the clustering space of different cell types. The results have been included in the supplementary materials.

Figure 4. Deconvolution of synthetic bulk data by Bulk2Space using unpaired simulations and comparison with other methods. a, Comparison of the performance of deconvolution between four different methods using unpaired simulation data. Left, *Pearson* correlation of gene expression between generated and input bulk data. Middle, *Spearman* correlation of gene expression between generated and input bulk data. Right, gene expression variation between generated and input bulk data using RMSE. **b**, Comparison of correlations of the marker gene expression between Bulk2Space (β -VAE), CGAN, GAN, and bMIND. The heatmap showed the correlation of different cell types in marker gene expression between generative methods and the ground truth. Right, the scale bar.

2. The second step, spatial mapping.

a. For spatial barcoding transcriptomics reference, I'm glad that the author can use deep-Forest to assign cell compositions of spatial data with high accuracy. It is not easy problem even using real single-cell reference for spatial data cross platform, for example RCTD (Cable, D.M., Murray, E., Zou, L.S. et al. Robust decomposition of cell type mixtures in spatial transcriptomics. *Nat Biotechnol* (2021). <https://doi.org/10.1038/s41587-021-00830-w>). Have you considered to validate your accuracy with real single-cell data and real spatial data?

Response: Thanks for your comments. We appreciate Reviewer #1 for supporting our spatial mapping step using the deep-forest model. To validate the spatial mapping step of Bulk2Space in the biological situation, two biological datasets, a spatial and a single-cell reference, from separated experiments were used to allocate single cells onto the spatial coordinates of the mouse hippocampus. The spatial reference data were sequenced by Slide-seq v2 and the scRNA-seq data were downloaded from the Seurat website.

As shown in Figure 5, for comparison, we first performed the Seurat clustering algorithm on the spatial data (Figure 5a). Notably, Seurat only provides a classified the spots directly into 15 clusters based on the expression profiles without annotating their cell types. Then, we performed Bulk2Space's spatial mapping step to calculate the pairwise similarity of cells and spots between the scRNA-seq and the spatial reference data. Each spot was deconvolved into single cells with cell-type annotations (Figure 5b). The spatial distribution of single cells predicted by Bulk2Space was consistent with the real structural pattern of cell types in the mouse hippocampus region.

Subsequently, we highlighted the CA1, CA2, and CA3 subregions in the mouse hippocampus. Seurat annotated the three subregions with the same labels (Figure 5c). In contrast, Bulk2Space successfully reconstructed the refined structure of the three subregions by mapping the corresponding cell types to these regions (Figure 5d).

Moreover, we investigated the spatial expression patterns of marker genes for six spatial specific cell types, CA1 principal cells, CA2 principal cells, CA3 principal cells, dentate principal cells, oligodendrocytes, and astrocytes. As shown in Figure 5e, the expression patterns of the marker genes showed obvious spatial distribution characteristics, which were consistent with the spatial distributions of the

corresponding cell types. The results suggested that Bulk2Space could map cells to spatial locations precisely in the biological situation. The results have been included in the supplementary materials.

Figure 5. Reconstruction of the mouse hippocampus structure at single-cell resolution using biological datasets. **a**, Spatial clustering of the spots by Seurat. **b**, Spatial mapping of individual cells in the scRNA-seq data to spots in the spatial reference by Bulk2Space. **c**, Annotations of spots in CA1, CA2, and CA3 subregions by Seurat. **d**, Annotations of single cells that were mapped to spots in CA1, CA2, and CA3 subregions by Bulk2Space. **e**, Spatial expression distribution of three marker genes of CA1 principal cells, CA2 principal cells, CA3 principal cells, dentate principal cells, oligodendrocytes, and astrocytes.

Reviewer #2 (Remarks to the Author):

Major Comments

1. Training data and methodology for Bulk2Space

a. What data was used to train these models? What is the breakdown of the data between training and validation?

Response: Thanks. To benchmark the performance of Bulk2Space, we designed two types of data simulation procedures.

Collections of single-cell RNA-seq data for data simulation

First, the reference data and the synthetic input data were generated from the same dataset, which was termed paired simulation. Ten scRNA-seq datasets were collected for paired simulation, including human and mice primary tissues. Specifically, the 5 human scRNA-seq datasets consisted of the peripheral blood (GSE92495), brain (GSE103723), kidney (GSE121862), liver (GSE124395), and lung (GSE130148). And the 5 mice scRNA-seq datasets consisted of the brain (GSE60361), kidney (GSE119531), lung (GSE127465), pancreas (GSE84133), and testis (GSE112393) (Figure 6a). Second, the reference data and the synthetic input data were generated from the same tissue but different datasets, which was termed unpaired simulation. In this simulation, 8 human pancreas scRNA-seq datasets from different resources were collected for unpaired simulation, including a CelSeq (GSE81076), a CelSeq2 (GSE85241), a Fluidigm C1 (GSE86469), a SMART-Seq2 (E-MTAB-5061) and four inDrops (GSE84133) datasets (Figure 6b). The detailed description of the experimental design for data simulation was summarized in Supplementary Data 1.

Paired and unpaired simulations for evaluation of the deconvolution step

For the deconvolution step, each scRNA-seq data was randomly divided into two parts for paired simulations, with one as the single-cell reference and the other as the input bulk transcriptomics data via aggregating all its single-cell gene expression profiles (Figure 6c). Considering the cell composition of bulk RNA-seq data varies greatly in the biological circumstance, for each single-cell reference, we further changed the cell-type proportions and synthesized three corresponding bulk transcriptome data with different cell compositions. In total, 30 paired simulation data were synthesized in this study. Next, for unpaired simulations, the simulated data were derived from the above 8 scRNA-seq datasets of the human pancreas. One dataset was randomly selected as

the single-cell reference data and another dataset from a different resource of the 8 scRNA-seq datasets was selected to synthesize the bulk transcriptomics data (Figure 6d). In total, 12 unpaired simulation data were synthesized in this study.

Paired and unpaired simulations for evaluation of the spatial mapping step

For the spatial mapping step, scRNA-seq data were used to simulate spatially resolved transcriptomics. In detail, we randomly chose 10 cells from each scRNA-seq data and aggregated their gene expression profiles as a spot of pseudo spatial transcriptomics data. The spot with over 25000 UMI counts would be sampled down to 20000 UMI counts to better meet the biological situation. We also simulated pseudo spatial transcriptomics data with 100, 200, 500, and 1000 spot numbers to biologically reproduce data derived from different spatial barcoding technologies. Similar to the deconvolution step, both paired and unpaired simulations were constructed to benchmark the mapping step of Bulk2Space.

Figure 6. Data collection and simulation workflow. **a**, scRNA-seq data collection for paired simulations. 10 scRNA-seq data across different species and tissues were collected. **b**, scRNA-seq data collection for unpaired simulations. 8 scRNA-seq data of the human pancreas from different platforms were collected. **c**, Paired simulation procedure for the deconvolution benchmark of Bulk2Space. Each single-cell (SC)

dataset was divided into two parts, one was used as the single-cell reference and the other was aggregated to synthesize the input bulk data. **d**, Unpaired simulation procedure for the deconvolution benchmark of Bulk2Space. The reference data and the synthetic bulk data were generated from different datasets of the human pancreas. **e**, Paired simulation procedure for the spatial mapping benchmark of Bulk2Space. Each single-cell dataset was divided into two parts, one was used to simulate spatial reference via assigning single cells to pseudo spots with spatial coordinates, and the other was used as the test set. **f**, Unpaired simulation procedure for the spatial mapping benchmark of Bulk2Space. The spatial reference and the single-cell test set were generated from different datasets of the human pancreas.

b. Was the same trained model used for all analyses and results presented in the paper?

Response: No, we train new models for different application circumstances to obtain the best performance.

c. Was the model pre-trained on a large, diverse collection of data, and then fine-tuned for each task? Or not fine-tuned? Alternatively, must Bulk2Space be re-trained from scratch on new data for each new application/task?

Response: No, we didn't pre-train on a large, diverse collection of data systematically. We recommend training for each circumstance independently. However, we have trained Bulk2Space in many biological circumstances, eg. Different brain regions, melanoma, zebrafish, and PDAC. We have proved that Bulk2Space can robustly deconvolve different bulk data after training.

d. What is the nature of the latent space learned by the beta-VAE? Could the authors provide perturbation analyses to probe the latent space? Similarly, what is the dimensionality of the latent space learned by the beta-VAE?

Response: Thanks for your comments. β -VAE is a modification of the Variational Autoencoder with a special emphasis on discovering disentangled latent factors. For example, a model trained on photos of human faces might capture the gentle skin color, hair length, hair color, emotion, and many other relatively independent factors in separate dimensions. One benefit that often comes with disentangled

representation is good interpretability and straightforward generalization to various tasks. Intuitively, this latent space makes β -VAE an effective tool for generating and understanding variations in natural data. We conducted the perturbation analysis with β -VAE (Figure 7). Since the original latent vectors obey the Gaussian distribution, here we change the distribution to uniform distribution and Poisson distribution and show the visualizations. As illustrated in Figure 7, cells generated under the original setting (Gaussian distribution) are mapped close to the original cells, while under the uniform and Poisson distributions, the generated cells are more clustered, resulting in poor clustering results. This confirms that better generation results can be obtained by assuming that the latent vectors follow the Gaussian distribution. The dimensionality of the latent space learned by the β -VAE is 256. The results have been included in the supplementary materials.

Figure 7. Perturbation analysis of Bulk2Space. **a**, Single-cell generation using Gaussian distribution. Left, distribution of Bulk2Space generated single cells (blue) and the test set (grey). Right, clustering space of the generated single cells. **b**, Single-cell generation using Uniform distribution. **c**, Single-cell generation using Poisson distribution.

e. Is there any data pre-processing prior to training the beta-VAE?

Response: Thanks for your comments. Yes. We have described the normalization process for both bulk and single-cell RNA-seq data in detail in the Methods section of the manuscript.

f. What is the relative weighting of the reconstruction loss and regularization loss in training the beta-VAE?

Response: Thanks. The relative weighting of the reconstruction loss and regularization loss is 1:4, which means we set the parameter beta as 4.

g. What hardware was used to train the models? For how many epochs?

Response: Thanks. We have trained the models in multiple GPU stations such as the Dell T7920 station (2 CPU cores, 192 Gb RAM, and NVIDIA Geforce RTX 3090), Alibaba group AliCloud (GPU TESLA T4), Dell T7280 station (2 CPU cores, 128 Gb RAM, and NVIDIA Geforce RTX 2080Ti), and Dell T7280 station (2 CPU cores, 64 Gb RAM, and NVIDIA Geforce RTX 2080Ti). The default running epoch is fixed to 3000. We use early stopping during the training phase, with which we stop training when the training loss is no longer reduced for 50 epochs.

h. Was any hyperparameter optimization employed?

Response: Thanks. In β -VAE, we use the Adam with weight decay (AdamW) optimizer with an initial learning rate of $1e-4$, the decoupled weight decay of $5e-4$, and Adam's betas parameters of 0.9 and 0.999. Moreover, we use early stopping during the training phase, with which we stop training when the training loss is no longer reduced for 50 epochs.

2. Model architectures and details

a. What is the model architecture? CNN-based? Multi-layer perceptron/fully-connected?

Response: Thanks. Both the encoder and decoder apply a four-layer perceptron. Each layer in the model is followed by a RELU activation except the last layer of the encoder

b. How many layers, neurons are used in the encoder/decoder networks?

Response: Thanks. We apply four layers for both the encoder and decoder networks. For the encoder, the number of neurons in each layer is 2048, 1024, 512, and 512

respectively. For the decoder, the number of neurons in each layer is 512, 1024, 2048, and k (k represents the number of genes in the dataset), respectively.

c. What is the dimensionality of the latent space?

Response: Thanks. The dimensionality of the latent space learned by the beta-VAE is 256.

3. Model evaluation and data ambiguities

a. The authors reference 30 single-cell transcriptomic datasets early in the manuscript. However, in the Methods, there is mention in Line 442: “Ten scRNA-seq were applied...”. Please clarify whether 30 single-cell datasets were used, or 10, and in either case, what the nature and references for these datasets are.

Response: Thanks for your comments. We have described the collection of single-cell datasets in detail in the revised manuscript and supplementary information. As answered in Question 1a, ten scRNA-seq datasets were collected for paired simulation, including human and mice primary tissues. Specifically, the 5 human scRNA-seq datasets consisted of the peripheral blood, brain, kidney, liver, and lung. And the 5 mice scRNA-seq datasets consisted of the brain, kidney, lung, pancreas, and testis. The detailed description of the experimental design for data simulation was summarized in Supplementary Data 2.

b. The descriptions of the dataset sourcing are generally insufficient. Namely, there are missing references – what studies did these datasets come from? Furthermore, for the spatial data, the authors state in Line 451-452, “the human PDAC, human melanoma... were used for reference and validation”. How were these data generated, i.e., what spatial transcriptomic technique? If pulled from literature, what are the references and accession numbers for these datasets?

Response: Thanks for your comments. We have thoroughly revised the manuscript and stated the resource, data generation, techniques, and accession numbers for these datasets. Moreover, we summarized all datasets used in this study in Supplementary Data 1.

c. Line 183-184: “A comprehensive comparison of six machine learning or deep learning approaches was then conducted using 40 datasets” – what are these 40 datasets? What is their size? What are the benchmark approaches tested? What are the details of those architectures; were all hyperparameter optimized and trained on the same datasets? What was the procedure for evaluation? Further analyses and discussion are required.

Response: Thanks for your comments, we fully agree with you and have provided the detailed information in the revised version. As answered in Question1a, the 40 datasets used for the spatial mapping step were simulated from 10 single-cell RNA-seq datasets. The detailed description of the experimental design for data simulation was summarized in Supplementary Data 4.

Benchmark approaches and the details of their architecture, hyperparameter optimization, and evaluation procedure

GAN and cGAN: The generator consists of two fully connected layers, followed by LeakyRELU and RELU activations respectively. The discriminator is also made up of two fully connected layers, where the first layer is followed by a LeakyRELU activation. We use the Adam optimizer with the initial learning rate of $1e-4$ and the betas parameters of 0.5 and 0.999. We train until the loss in the generative and discriminative phases is no longer reduced for 50 epochs, and use the model obtained at this time as the final model for prediction.

Logistic Regression(LR): We implement L2 regularization as the additional penalty term to solve the problem of overfitting. We use the L-BFGS algorithm as the solver, which uses the Hessian matrix to iteratively optimize the loss function. We fit the model according to the given training data and return the probability estimates on the testing set.

Decision Tree(DT): We use the Gini impurity to measure the quality of a split and choose the best split at each node. We don't set any maximum depth of the tree, so nodes are expanded until all leaves are pure or until all leaves contain less than 2 samples. We build the decision tree classifier from the training set, and then predict the class probabilities of the input samples.

Gradient Boosting Decision Tree(GBDT): We set the learning rate as 0.1 and the number of boosting stages to perform as 100. The loss function to be optimized is the log loss function, and we choose Friedman MSE to measure the quality of a split. The

maximum depth of the individual regression estimators is set to 3. We fit the gradient boosting model on the training set and use the trained model to make predictions on the testing set.

Multilayer Perceptron(MLP): We apply a two-layer perceptron, where each layer is followed by batch normalization, a RELU activation, and a dropout layer with the probability of element zeroing as 0.1. We use the Adam optimizer with the initial learning rate of $1e-4$ and the betas parameters of 0.9 and 0.999. We train until the loss is no longer reduced for 30 epochs, and use the model obtained at this time as the final model for prediction.

DeepGBM: We set the number of tree groups to 100. The dimension of leaf embedding for a tree group is set to 20. The structure of the distilled NN model is a fully connected network with "100-100-100-50" hidden layers. We adopt the feature selection in each tree group, where we first sort the features according to the information gained, and the top 128 of them are selected as the inputs of distilled NN model. We use the Adam with weight decay (AdamW) optimizer with an initial learning rate of $2e-3$, the decoupled weight decay of $1e-6$, and Adam's betas parameters of 0.9 and 0.999. We train until the model reaches the highest ROC-AUC score, and use it for prediction.

4. Model generalizability and “test set” performance

a. The analyses in Figures 3 and 4 describe use of both “Slice 1” and “Slice 2”, which seem to be consecutive sections from the same tissue, wherein inference evaluations on Slice 1 are done based on deconvolution of bulk data generated from Slice 2. If these two slices are derived from the same specimen, and especially as consecutive sections, they will inherently be highly correlated, and therefore will bias the “test set” performance. What was the spatial distance between the two slices, i.e., were they direct consecutive sections? Could the authors provide H&E of the two slices? More importantly, what about performance and generalization to an independent “test” specimen? This is as present a major consideration and limitation of the work.

Response: Thanks for your comments. Let me explain, the slices in the melanoma application were consecutive slices. However, in the PDAC example, the two slices were derived from different tissue samples. In the revised version, we have included these two examples in the supplementary materials as validation cases for Bulk2Space and the H&E staining of the two slices were provided. Alternatively, three biological

bulk RNA-seq datasets of the PDAC and mouse isocortex from totally different experiments were collected for the spatial deconvolution application of Bulk2Space in the main text.

b. The authors should evaluate their method on multiple independent “test” tissues/specimens after training is complete (see comments above regarding model training). These evaluations are critical to assessing the performance and generalizability of the method.

Response: Thanks for your suggestion. As you can see in the revised manuscript and supplementary materials. We have provided more application cases based on independent tissues from separated experiments such as the reconstruction of the mouse hippocampus (Reviewer #1, Question 2a) and isocortex (Figure 8), spatial analysis of PDAC bulk data (Figure 9) in addition to the applications already in our original version of the manuscript.

Bulk2Space reconstructs the hierarchical structure of the mouse isocortex region sequenced by Spatial-seq

The bulk transcriptomics data of mouse isocortex were sequenced by our in-house developed multiplexed RNA-seq approach, termed Spatial-seq. The detailed information for Spatial-seq was described in the ‘Methods’ section of the manuscript and illustrated in supplementary information Fig. S18. As shown in Figure 8a, the mouse primary visual cortex regions in different coronal sections from anterior to posterior were collected and sequenced by SMART-seq2. The result scRNA-seq data were used as the single-cell reference. A sagittal section of the mouse brain was divided into two parts and sequenced using 10X Visium to obtain spatial transcriptomics data of the isocortex region. The spatial reference data were downloaded from 10X datasets (Supplementary Data 1).

Figure 8. Spatially resolved single-cell analysis of the mouse isocortex bulk data by Bulk2Space. **a**, The resources of the bulk transcriptome data, single-cell reference, and spatial reference. The bulk RNA-seq data of the mouse isocortex was sequenced using our in-house developed Spatial-seq which combines LCM and spatial barcoding strategies. The single-cell reference data is derived from scRNA-seq of a collection of consecutive mouse primary visual cortex sections from anterior to posterior. The spatial reference data is obtained from an open-access database using the 10X Visium

approach. **b**, The deconvolution results of the mouse isocortex bulk data by Bulk2Space. The UMAP layout showed the clustering space of the generated single cells. **c**, Spatial mapping of the single cells generated from the mouse isocortex bulk data by Bulk2Space. **d**, Pairwise expression correlation of cell-type-specific marker genes between single cells generated by Bulk2Space and single cells in the reference data. **e**, Spatial distribution of the cell-type proportion predicted by Bulk2Space. **f**, Spatial cell-type abundance of seven cell types with layered structure predicted by Bulk2Space. **g**, Spatial expression distribution of cell-type-specific marker genes predicted by Bulk2Space at single-cell resolution.

Bulk2Space integrates spatial gene expression and histomorphology in PDAC using biological datasets

As shown in Figure 9a, two bulk RNA-seq data of the pancreatic adjacent (PA) and the PDAC tissues, one scRNA-seq data, and one spatial transcriptomics data were derived from two individual experiments and three different technologies. The scRNA-seq data were used as the single-cell reference for the deconvolution of PA tissue and PDAC bulk data, and the spatial barcoding-based ST data were used as the spatial reference.

In conclusion, the performance evaluation of Bulk2Space using biological data instead of simulated data demonstrated that Bulk2Space can efficiently deconvolve bulk RNA-seq data into spatially resolved single-cell transcriptomics. The results were included in the main text of the revised manuscript.

Figure 9. Spatially resolved analysis of PDAC by Bulk2Space. **a**, Collection of PA and PDAC bulk transcriptome data, PDAC scRNA-seq data, and PDAC spatial transcriptomics data. Two bulk data were sequenced by bulk RNA-seq. The scRNA-seq data sequenced by inDrop-seq⁵⁰ is used as the single-cell reference. One slice of the sectioned PDAC tissue sequenced by ST¹⁴ was employed as the spatial reference. **b**, Cell-type proportions of single cells generated from PDAC and PA tissues by Bulk2Space. Purple box, the proportion of the cell type was higher in PA tissue than that in PDAC tissue. Black box, the proportion of the cell type was higher in PDAC tissue

than that in PA tissue. Teal, the PDAC tissue. Red, PA tissue. **c**, Clustering space of single cells generated from the PDAC (Top) and PA (Bottom) bulk data by Bulk2Space using UMAP layout. Different colors represented distinct cell types. **d**, Pairwise expression correlation of cell-type-specific marker genes between single cells generated by Bulk2Space and the single-cell reference for PDAC. **e**, Spatial mapping of single cells generated from PDAC bulk data. **f**, Top, Histological annotation for cancer region (red), duct epithelium (yellow), and normal pancreatic tissue (light blue). Middle, clusters of cancer region (yellow), pancreatic tissue (blue), duct epithelium (green), and stroma (dark grey) from the spatial transcriptomics data. Bottom, spatial deconvolution result of the bulk PDAC data by Bulk2Space with each spot displaying the composition of cell types in the pie chart. **g**, Spatial abundance of acinar cells, cancer clone A cells, cancer clone B cells, and ductal cells in each spot on the tissue section predicted by Bulk2Space. **h**, Spatial expression of the marker genes in acinar cells, cancer clone A cells, cancer clone B cells, and ductal cells predicted by Bulk2Space at spot (left) and cellular (right) resolution.

c. In addition, the authors perform an analysis wherein a single-cell dataset is split into a “reference” subset and a “bulk data synthesis” subset. They then aggregate the “bulk synthesis” subset to generate pseudo-bulk transcriptome data, and then evaluate correlation between the generated and true single-cell data on a held-out set. With this setup, questions about generalizability remain. What about training on data from studies that have independent, paired bulk and single-cell data? Evaluation on these datasets will help verify the robustness of the method.

Response: Thanks for your valuable suggestion. In the revised version, a comprehensive experimental design for the performance evaluation of Bulk2Space was conducted, including paired simulations (bulk and reference data were generated from the same datasets), unpaired simulations (bulk and reference data were generated from the same datasets), biological datasets (Figure 10), etc.

Deconvolution performance evaluation of Bulk2Space using paired biological bulk and single-cell datasets

Based on the robust performance of Bulk2Space on simulated data, we further validated it with biological data. In biological circumstances, bulk and single-cell RNA-seq data from the same tissue can be treated as paired datasets because they share identical conditions. In this study, three paired bulk and single-cell RNA-seq data of different mice livers (GSE119340) fed with standard chow were collected to evaluate the performance of Bulk2Space. As shown in Figure 10, three paired mice bulk and

single-cell profiles were used as input bulk data and single-cell reference data. Through Bulk2Space deconvolution, the correlation of the marker gene expression for different cell types between generated and the reference single-cell profiles showed that Bulk2Space could be well applied in biological scenarios.

Figure 10. Deconvolution of bulk data using biological datasets by Bulk2Space. a, b, and c, Correlation of the marker gene expression for different clusters between Bulk2Space generated single-cell data and the reference data. Right, the scale bar.

d. Similarly, what about analyses when the scRNAseq and spatial datasets are decoupled, i.e., not from the same tissue specimen or source? This evaluation is also requested to verify robustness and generalizability.

Response: Thanks for your constructive suggestion. In this study, most scRNA-seq and spatial datasets are decoupled. Moreover, as shown in the examples of the reconstruction of the mouse hippocampus (Reviewer #1, Question 2a) and hypothalamus (Figure 11), we have addressed this issue in particular.

Spatial reconstruction of the mouse hypothalamus by Bulk2Space

The second spatial mapping strategy uses image-based targeted methods. A scRNA-seq data (GSE113576) and MERFISH data (image-based reference) of the mouse hypothalamus tissue from the same experiment were used as paired datasets to test the performance of the Bulk2Space algorithm for the second spatial mapping strategy. Another scRNA-seq data derived from a separated experiment was used as unpaired single-cell data. The results showed that spatially, the gene expression pattern of the matched cells was highly correlated with that of the reference, and the predicted spatial expression patterns across all 150 targeted genes are strongly correlated with the ground truth with *Pearson* correlation coefficients (PCCs) over 0.9 for both paired

and unpaired datasets (Figure 11).

Figure 11. Spatial expression of genes in the ground truth and predicted by Bulk2Space using paired and unpaired single-cell RNA-seq data. PCC for each gene was shown.

5. Other major comments

a. The authors describe the Spatial-seq method to generate new spatial transcriptomics data and evaluate Bulk2Space's performance using Spatial-seq data as a reference. When deployed for the same application or on the same tissue type, how well does the method generalize to other spatial transcriptomic methods for generating the spatial reference? Greater discussion on modularity and generalizability in this respect is warranted.

Response: Thanks for your comments. Sorry that we had not clearly described the detailed information of the Spatial-seq method in the original manuscript. In this study, the Spatial-seq can be regarded as a multiplexed RNA-seq method because it was only used to provide two sets of bulk RNA-seq data, the mouse isocortex, and hypothalamus. In the revised version, we have stated the data acquisition of bulk transcriptomics by Spatial-seq (Figure 12).

Workflow of our in-house developed Spatial-seq technology

The Spatial-seq workflow is shown in Figure 12a, the laser capture microdissection (LCM) was used to isolate brain regions and a spatial barcoding strategy was utilized for multiplexed RNA-seq. Specifically, the mouse brain was sliced into 14- μ m sections from the coronal and sagittal directions. Each tissue slice was registered to a reference brain template provided by the Allen Brain Atlas (<https://portal.brain-map.org/>). After spatial registration, the anatomical regions of the mouse brain were delineated and annotated. The outlines, dissection sequences, and collectors of the brain regions were specified from the annotated data and then imported to an LCM instrument. LCM is a microscope-guided powerful cutting system incorporating UV light for contact- and contamination-free isolation of areas of interest from tissue sections. Brain regions were dissected from the tissue by LCM according to the imported files and fell into the collector loaded with barcoded beads in advance. Subsequently, the tissue was lysed in the barcoded well, allowing mRNA to be captured by polyT tail on the surface of the magnetic beads. The captured mRNA was reversely transcribed to construct cDNA library and a paired-end sequencing was conducted to decode the spatial barcodes in the 3' end and detect RNA species in the 5' end of the cDNA.

In this study, 214 brain regions from three mice were sequenced following the Spatial-seq procedure. Two out of the 214 bulk RNA-seq data, the mouse isocortex and hypothalamus, were used as two examples for Bulk2Space applications (Figure 12b).

Figure 12. Workflow of Spatial-seq and two isolated regions of the mouse brain were used in this study. **a**, Spatial-seq workflow. The mouse brain was sectioned continuously from coronal and sagittal directions. For each slice, spatial registration was carried out to annotate each brain region to be isolated. Brain regions are dissected by LCM and labeled with a unique barcode and then followed with RNA-seq. **b**, Two isolated brain regions from two mice were used in the main text. The

hypothalamus region from a coronal section and the isocortex region from a sagittal section. The scale bar for each slice is shown in the figure.

b. When describing the generated single-cell reconstructions, the authors often use the language “deconvolved single-cell profiles” or “after deconvolution, input bulk data is transformed into single-cell transcriptomics”. This language is a bit misleading, as it gives the impression that these outputs are reflective of real data, when in truth the output data is a generative reconstruction. There are certain instances when the authors correctly state “generated single cells”; however, this is not consistent. Please correct.

Response: Thanks for your kind suggestion. We have thoroughly revised the manuscript and corrected the inappropriate expressions.

c. It is not clear that the requirement for a spatial reference will ultimately be needed. In training, it seems that the spatial reference supervises a mapping from reconstructed single-cell output (after bulk deconvolution) to spatial coordinates. It seems then that after training, it would be possible to go directly from bulk RNA-seq to spatial coordinates without a reference but with fully trained models. Could the authors please clarify this point? Analyses without spatial reference would significantly improve the broad utility of the method.

Response: Thanks for your comments. We have considered your concerns carefully and take a different view. In our opinion, the spatial reference is still required because the single cells generated from bulk data do not contain spatial information, and as we know, there is currently no such computational approach that can directly predict the spatial location of cells without spatial reference. It is a tremendous challenge in this area.

d. Ideally in deployment, one would want to use Bulk2Space in a new setting where they may not have information on the cell types present. Namely, one could envision performing the analysis and spatial mapping and then annotating the cell types. The package should support this ability and support the option to not provide marker

genes as input.

Response: Thanks for your comments, we fully considered the suggestion. we have performed the deconvolution step of Bulk2Space using single-cell reference without cell-type annotations. As shown in Figure 13, we first clustered the reference single-cell profiles into five clusters using the existing clustering algorithm, the Louvain of Seurat. Based on the clustering space of the five resulting clusters, single cells were generated from simulated bulk data by Bulk2Space. Compared with the test set, the single-cell RNA-seq data generated by Bulk2Space showed consistent distribution (Figure 13a). The clustering space of generated single-cell transcriptomics data and the reference data were exhibited in Figure 13b. And the correlation of marker gene expression for different cell clusters showed that the corresponding cell clusters had higher correlations, suggesting that Bulk2Space can robustly generate single cells using unannotated single-cell reference (Figure 13c).

Figure 13. Deconvolution of bulk data using unannotated single-cell reference by Bulk2Space. **a**, t-SNE layout showed the distribution of Bulk2Space generated single cells (blue) and the test sets (grey). **b**, t-SNE plots of single cells generated by Bulk2Space demonstrated the clustering space of different cell clusters. **c**, Correlation of the marker gene expression for different clusters between Bulk2Space generated single-cell data and the ground truth. Right, the scale bar.

Minor Comments

1. Grammatical and language errors can be found throughout the text. Please correct these in revision. Two examples are included here. First, line 58-59: “The in silico method has great potential to predict spatial heterogeneity from bulk RNA-seq data at single-cell resolution by integrating cutting-edge technologies”, should be corrected to something like “In silico methods have the potential to...”. Second, line 62: “The emerging of an approach that can efficiently convert...”. Please correct grammatical issues. This will help improve the readability of the paper.

Response: Thanks for your kind suggestion. We have thoroughly revised the manuscript and corrected the inappropriate expressions to improve the readability of the paper.

2. Figure 1b: In the “Single-cell reference” sub-region, there are two “C4” annotations, when it seems that one should be labeled C2. Please correct. Additionally, if the organization numbers (1)-(6) are not referenced in the figure caption, they should be eliminated; otherwise, they may be referenced in the figure caption.

Response: Thanks for your correction. We have thoroughly revised the figures and corresponding descriptions. The organization numbers have been removed in the revised Fig. 1 in the main text.

3. Figure 2: please clarify what tissue type(s) are used for the analyses in subpanels (d)-(i).

Response: Thanks for your comments. We have clarified the information and included these two subfigures in the supplementary materials.

4. Line 237: The use of “good correlation” is inexact and nonspecific. What is a good correlation? Please provide the value of the correlation coefficient and an associated metric of statistical significance, both here and elsewhere throughout the text.

Response: Thanks for your kind suggestion. We have thoroughly revised the manuscript and provided the corresponding statistical metric.

5. Line 368-372: “The Bulk2Space results indicated that...CAFs accumulated in normal glands at the early stage...promoting the occurrence of local inflammation, a fundamental innate immune response to perturbed tissue homeostasis, thus leading to tissue cancerization.” This is an overstated claim and an over-inflated interpretation of the results. The authors have performed a correlative analysis, but have not proven anything about the causal source or mechanisms of tissue cancerization in this context. Please temper the claims.

Response: Thanks for your comments and we agree with you that the sentence is an overstated claim. We have thoroughly revised the manuscript and corrected the inappropriate claims.

6. Line 411: “Bulk2Space can assign single cells to optimal spots, and these mixtures are split into collections of individual cells, thus achieving single-cell resolution.” The claim to have achieved single-cell resolution is overstated. Bulk2Space describes a generative process that involves the generation of synthetic data (in converting bulk RNA-seq to estimate single-cell profiles); based on this the resulting assignment to spatial coordinates is simply a prediction – the statement of achieving single-cell resolution implies some sort of measurement, which is not the case here. Please correct.

Response: Thanks for your comments and we agree with you that the statement should be modest. We have thoroughly revised the manuscript and corrected the inappropriate statements in this version.

7. Line 412-415: “...uncovering the molecular mechanism underlying the progression of the disease”. Like the comment regard Line 368-372, this is an overstated claim. The authors have not proven anything about the causal source or mechanisms of tumor progression in this context; please temper the claims.

Response: Thanks for your constructive comments and suggestions. We have thoroughly revised the manuscript and corrected the inappropriate statements to avoid misleading.

8. Code: The code needs to be cleaned and better commented. To promote use by the community, there should be more extensive documentation and explanation such that people will know how to use it. In particular, the `bulk2space.py` file is a single script – there are no other functions provided, and it is not well commented – this is not modular and usable coding practice. Please improve the clarity, readability, and usability of the codebase in order to promote its use by the community.

Response: Thanks for your valuable suggestion, we fully agree with you that the code

needs to be cleaned and better commented. Therefore, we have made efforts to clean our codes, add more comments, and write step-by-step tutorials for Bulk2Space (<https://github.com/ZJUFanLab/bulk2space/tree/main/bulk2space/tutorial>).

Hopefully, this will make our project and codes more readable and easier to use.

Reviewer #3 (Remarks to the Author):

Major comments

1. As a central part of the work, the description of the deconvolution procedure (Fig 1b, [117-125]) is rather succinct and it is current from difficult to follow. It would benefit for more explanation.

Response: Thanks for your constructive comments, we agree that the description of Bulk2Space should be detailed for readers to follow up. We have included more explanation of the deconvolution procedure and other information about Bulk2Space in the revised manuscript. We expect the revised version to increase readability.

2. How does bulk2space methods of deconvoluting spatial barcoding methods compare to some already published methods such as Stereoscope, SpatialDWLS or Spotlight?

Response: Thanks for your comments. In the revised manuscript, we have compared the Bulk2Space with these four spatial deconvolution methods. As shown in Figure 14, a comprehensive comparison of six machine learning or deep learning approaches and 4 published methods, RCTD, SpatialDWLS, stereoscope, and SPOTlight was then conducted using 40 paired simulation datasets (single-cell and spatial data were derived from the same datasets). As shown in Figure 14a, compared with these methods, Bulk2Space showed higher correlations in gene expression and lower RMSEs than other methods. Next, we benchmarked the spatial mapping step of Bulk2Space using unpaired simulation data of mice pancreas (single-cell and spatial data were derived from different datasets). As shown in Figure 14b, Bulk2Space showed the highest correlations in gene expression and the lowest RMSEs.

Figure 14. Benchmark test for Bulk2Space. **a**, Benchmark test for 10 spatial mapping methods (from left to right, Bulk2Space (deep forest), gradient boosting decision tree, decision tree, multi-layer perceptron, DeepGBM, linear regression, RCTD, spatialDWLS, stereoscope, and SPOTlight) using paired simulation data. Left, *Pearson* correlation of gene expression, Middle, *Spearman* correlation of gene expression, Right, gene expression variation (RMSE). **b**, Benchmark test for 10 spatial mapping methods using unpaired simulation data.

3. To confirm the robustness of bulk2space, it would be ideal to see how for the second spatial mapping strategy (MERFISH) bulk2space performs when spatial data deconvolution is carried out when using scRNA-Seq data from a similar tissue from a separate experiment.

Response: Thanks for your kind suggestion. As answered in Question 4d raised by Reviewer #2, we have performed the spatial mapping step of Bulk2Space with scRNA-seq from a separated experiment. The results have been included in the revised manuscript. Specifically, a set of scRNA-seq data (GSE113576) and MERFISH data (image-based reference) of the mouse hypothalamus tissue from the same experiment were used as paired datasets to test the performance of the Bulk2Space algorithm for the second spatial mapping strategy. Another scRNA-seq data derived from a total separated experiment was used as unpaired single-cell data. The results showed that spatially, the gene expression pattern of the matched cells was highly correlated with that of the reference, and the predicted spatial expression patterns across all 150 targeted genes were strongly correlated with the ground truth with average *Pearson* correlation coefficients (PCCs) over 0.9 for both paired and unpaired datasets (Figure 11, Page 23).

4. In validation of MERFISH data with reference genes, a split of 80% for reference and 20% for validation would help further validate performance of the algorithm.

Response: Thanks for your constructive suggestion and we fully agree with you. We have included five-fold cross-validation for the spatial mapping of Bulk2Space (Figure 15). Specifically, we randomly split the 150 target genes identified in MERFISH into five folds using R package caret (version 6.0-92), a split of 80% for reference and 20% for validation. As shown, the spatial expressions of the top 25 genes predicted by Bulk2Space for the paired and unpaired single-cell data were compared with MERFISH.

Figure 15. Comparison of the spatial expression of 25 genes between MERFISH (Top) and Bulk2Space results with the paired (Middle) and unpaired (Bottom) single-cell data using five-fold cross-validation. PCCs of the spatial gene expressions were listed.

5. Performance of Bulk2Space should be further verified using bulk transcriptomics data not generated from the reference scRNA-Seq data but from a separate bulk transcriptomics experiment, as this will more closely represent how users will employ the algorithm.

Response: Thanks for your constructive suggestion we agree with you that biological data should be included. As answered above (Reviewer #2, Question 4c), we evaluated the deconvolution performance of Bulk2Space using biological bulk and single-cell datasets. In biological circumstances, bulk and single-cell RNA-seq data from the same tissue can be treated as paired datasets because they share identical conditions. In this study, three paired bulk and single-cell RNA-seq data of different mice livers (GSE119340) were collected. As shown in Figure 10 (Page 22), after deconvolution, the cell-type-specific marker gene expression of single-cell profiles generated by Bulk2Space was highly correlated with the reference, showing that Bulk2Space could be well applied in biological scenarios in addition to simulated data.

6. Availability of the algorithm: could the authors provide a detailed protocol for installation including necessary dependencies. We have manage to run bulk2Space, not without difficulties:

Response: Thanks for your constructive comments and we agree that the installation instructions should be more readable. The necessary dependencies were distributed in (<https://github.com/ZJUFanLab/bulk2space/blob/main/requirements.txt>). We have provided a detailed installation protocol and other details can be found in our tutorial (<https://github.com/ZJUFanLab/bulk2space/blob/main/bulk2space/README.md>). Hopefully, this will work for you.

a. We tried to set up conda environment on two separate systems with GPUs to run demo data, setting up the conda environment failed on both systems. Further information on the system the conda environment was tested on would be beneficial, to see if it could be reproduced by users. Additionally, if programme is able to be run without GPUs, that option should be made more explicit to broaden the scope of potential users and make tool more accessible.

Response: Thanks for your useful comments and we are sorry that didn't have such experience. We have tested Bulk2Space on different platforms and it could just work. Although the necessary dependencies were provided, we are willing to afford remote assistance if similar situation happens again. In addition, in the revised version, we have added the option to use CPU in this program. Set the parameter `gpu_id` to `-1`, then you can run it without GPU.

b. More information should be provided on the Github page on the programmes parameters.

Response: Thanks for your comments, we fully agree with you. We have provided step-by-step analysis tutorial of several examples and detailed instructions on GitHub (<https://github.com/ZJUFanLab/bulk2space/blob/main/bulk2space/README.md>). The adjustable program parameters are fully commented on in the tutorials and codes.

c. Additional specifications required on the computing power required to run the programme and GPUs/CUDA system it is compatible with. Bulk2Space currently only works on certain Linux servers with specific CUDA version support which is not mentioned in the paper or the Github. This should be made more explicit. More relaxed constraints would be ideal as not all users will have access to a similar environment.

Response: Thanks for your comments. Actually, Bulk2Space is not difficult to install. We don't need a specific version of CUDA, but you are recommended to install the corresponding version of PyTorch according to the CUDA version of your Linux server. This website can help you find the corresponding relationship between the two: https://download.pytorch.org/whl/torch_stable.html. Also, we can provide remote assistance if necessary.

d. Test run using provided demo based on spatial-barcoding data ran successfully on one of the systems tested. Additional documentation should be provided on Github page with regards to the significance of the output files to give the user a clearer understanding of what they correspond to.

Response: Thanks for your comments, we fully agree with you. We have provided step-by-step analysis tutorial of several examples and detailed instructions on GitHub (<https://github.com/ZJUFanLab/bulk2space/blob/main/bulk2space/README.md>), which is easier to get started in this revision.

Minor comments

1. The value of Bulk2Space in being able to deconvolve multiple RNA-Seq datasets when a single scRNA-Seq and spatial transcriptomics reference are provided needs to be stated more explicitly in the abstract and throughout the main body of the paper. The submitted abstract reads as Bulk2Space would be a deep learning framework that allows spatial deconvolution of bulk data without the necessary references required.

Response: Thanks for your constructive suggestion and we fully agree with you. We have thoroughly revised the manuscript based on your advice. The statement of the use of single-cell and spatial reference is stated more explicitly in the abstract and throughout the main text of the revised manuscript.

2. For instance, in line 85: 'prevailing spatially resolved transcriptomics methods can provide reference coordinates for single cells' as an explanation for their method yet in their introduction the authors state that they will incorporate data from spatial methods that have not achieved single-cell resolution. Keeping consistent across spatial methods will make the method easier to follow for readers.

Response: Thanks for your comments. We have checked the inconsistent statements and unified the expression throughout the manuscript.

3. Check grammar on paragraphs 58-64, eg "the emerging of an approach"

Response: Thanks for your comments, we have checked the grammar.

4. Sentence in line 68 could be made clearer.

Response: Thanks for your useful comments. The detailed explanation of the

hypothesis is described in the design concept of Bulk2Space in the 'Result' section. Compared with the original version, substantial descriptions were included in the revised manuscript to make it clearer.

5. Fig 2b color for generated and ground truth data is hard to distinguish

Response: Thanks for your comments, we have revised the figure and changed the color of the generated data (Figure 16). This sub figure (Fig. 2b in the original manuscript) has been included in the supplementary information in the revised version.

Figure 16. The distribution of single cells generated by Bulk2Space (blue) and the test sets (grey).

6. In figure 2F it is difficult to distinguish from this visualisation the exact similarity and differences between the ground truth and bulk2space generated data.

Response: Thanks for your comments, we have revised the figure and included the spatial deconvolution pie charts in the supplementary information (Figure 17).

Figure 17. Comparison of the cell-type composition of each simulated spot between

Bulk2Space and the spatial reference.

7. Figure 3D histological annotation needs to be clearly – difficult to see contrasting colours on top of H&E

Response: Thanks for your comments. The histological annotations were provided by the original article and we had not modified it. In the revised version, this figure has been included in the supplementary information. As shown in Figure 18, we have made clearer annotation marks on the replacement figure in the main text.

Figure 18. Histological annotation for cancer region (red), duct epithelium (yellow), and normal pancreatic tissue (light blue).

8. In Figure 3c and 3d the colours chosen for cancer cells and erythrocytes need to be more contrasting as it is currently challenging to distinguish between them in the figures.

Response: Thanks for your comments. The original Figure 3 have been included in the supplementary information.

9. In figure 3G the highest correlation between expression of markers is ~ 0.5 , an explanation for this result should be offered.

Response: Thanks for your constructive comments. The task for Bulk2Space was different. Because we map single cells to spatial coordinates, there would be batch effect of gene expression between the single-cell and spatial data. Although the correlation of the marker gene expression was around 0.5, the corresponding cell types were highly correlated between the two heterogeneous data.

10. Explanation of how histological annotations were performed is needed.

Response: Thanks for your constructive comments. The histological annotations were provided by the author of the cited article not by us. In the original article, the authors annotated the slide for distinct histological features according to the H&E staining and brightfield imaging.

11. In Figures 4B and 4E the expression correlation of marker genes between bulk2space and the slices is not particularly strong, could an explanation be offered for this?

Response: Thanks for your comments. As explained above, there is batch effect of gene expression between single-cell and spatial data, thus lower the correlation coefficient of two heterogeneous datasets. However, the correlation of pairwise cell-type correspondence was considerable.

12. Figure 4G legends needs quantitative value instead of just 'high'.

Response: Thanks for your comments, we fully agree with you. We have provided statistical metric in the revised manuscript. However, in some cases, the quantitative range of each subfigure is different from each other. Therefore, we used 'high' and 'low' instead.

13. Figure 6B and 6E B cells, monocytes and NK cells need to be in more contrasting colours to be able to more easily distinguish between them.

Response: Thanks for your valuable comments, we have revised the Figure 6 in the

manuscript (Figure 19) with more contrasting colors.

Figure 19. Revised figure 6B and 6E with more contrasting colors.

14. It would be ideal if all t-SNE plots in paper should be updated to UMAPs.

Response: Thanks for your suggestion, we have changed most of the t-SNE plots to UMAPs in the revised version of the manuscript.

REVIEWER COMMENTS

Reviewer #1 (Remarks to the Author):

Liao et al. have made a tremendous effort to make the manuscript much better than the previous version. They have added the complete evaluation of Bulk2Space step by step with simulated and real data, specified model configuration, and added new datasets as examples. I'm satisfied with the current manuscript.

Minor Comment:

1. In Figure 4a, the authors compared the deconvolution performance between different methods. There are three metrics being computed: Pearson's correlations, Spearman's correlations, and RMSE. How are they calculated? You need to specify that you combined the gene expression from all cell types and then compute the three metrics.
2. How about the cell type-specific gene expression evaluations? Do we have the same pattern in Pearson's correlation, Spearman's correlation, and RMSE?

Reviewer #2 (Remarks to the Author):

The authors have provided additional details and new analyses, which have improved the quality of the manuscript, the robustness of the results, and the comprehensibility of the methods. Most of my concerns have been addressed in this revision. However, some lingering concerns remain, most significantly with respect to the quality of the source code.

1. The Abstract provides a better and more accurate description of the contributions of the work. However, the syntax and language is not concise, particularly the last sentence of the Abstract. Please address.
2. Major issues still exist with the code. There is only one main script, `bulk2space.py`, and it is very lengthy, not well-commented, and does not follow coding best practices in terms of division into composable functions. In general the code organization is very poor – for example there is no dedicating training or testing script; classes and variables are often not named properly (e.g., “myDataset”), etc. This undercuts the quality and technical rigor of the work. With this quality of the code, I have doubts as to the suitability of the work, in terms of its technical quality and polish, for publication in Nature

Communications. Additionally, while the provided Tutorials are appreciated, they are not really tutorials as they do not provide step-by-step guidance for the user. At the least, the authors must improve the quality of their scripts and code from a software / package perspective. Relatedly, parameters for running the Bulk2Space model are currently provided as dictionaries; there is no use of .config files. .config files as well as publication of saved model weights would support the reproducibility of the results and the utility of the method. In its current form this work is not ready for publication as a usable package.

3. Please add explicit description of the computing environments (e.g., GPUs used for training) used into a section within the Methods.

4. Please provide descriptive titles for each of the Supplementary Data files. The main Supplementary Information should also be labeled / titled accordingly.

Reviewer #3 (Remarks to the Author):

The revisions were satisfactory and the manuscript and workflow have substantially improved in clarity. I have only 2 minor remarks:

1- Bulk2Space was successfully ran on PDAC data however this was not easy with the raw hypothalamus data, please provide detailed instructions on how to use that data (i.e. to produce the bulk.csv, sc_data.csv etc.).

2- Page 10 of comments to reviewers under section 'Paired and unpaired simulations for evaluation of the spatial mapping step', for simulating pseudo spatial transcriptomics data 100-1000 spot numbers were used. The upper boundary seems low as Visium (which is specifically referenced in the paper) has up to 5000 spots per capture area. I suggest for this analysis to be extended to 5000 spots.

Response to reviewers

Overview of Changes

We sincerely appreciate the reviewers' constructive comments and positive feedback. Our work has been much improved based on their valuable suggestions. We have tried our best to address the concerns raised by the reviewers point by point. Compared with the previous version, the revised source code focuses more on readability, specification, and operability. We responded to all comments with detailed figures and experimental results. The main changes are summarized as follows.

1. We have specified the detailed calculation process of the *Pearson* correlation, *Spearman* correlation, and RMSE and compared these metrics of cell-type-specific maker gene expression by Bulk2Space with other methods.
2. We have re-organized the source code based on the composable function of Bulk2Space, specified the naming of all classes and variables canonically, commented on the source code in detail, wrapped the Bulk2Space code as a package, and provided step-by-step tutorials for users.
3. We have provided the computing environment of Bulk2Space and descriptive information of Supplementary data files.
4. We have added detailed instructions on the formats and descriptions of the key parameters in Bulk2Space.
5. We have extended the spot number of the simulated spatial transcriptomics data to 5000 to match mainstream ST technologies such as Visium.

We hope this edition will satisfy the reviewers and address all the concerns to win their approval for the publication of our manuscript. Please find detailed responses to each concern below.

Reviewer #1 (Remarks to the Author):

Minor Comment

1. In Figure 4a, the authors compared the deconvolution performance between different methods. There are three metrics being computed: Pearson's correlations, Spearman's correlations, and RMSE. How are they calculated? You need to specify that you combined the gene expression from all cell types and then compute the three metrics.

Response: Thank you very much for your positive comments and valuable suggestion. In the revised manuscript, we have described the detailed calculation process (Highlighted in blue, Page 8, Lines 202-204) as 'The *Pearson* correlation, *Spearman* correlation, and RMSE were calculated by combining the gene expression across all cell types and averaging these metrics for each cell type'.

2. How about the cell type-specific gene expression evaluations? Do we have the same pattern in Pearson's correlation, Spearman's correlation, and RMSE?

Response: Thanks for your comments. The expression correlation for cell-type-specific marker genes between the generated single-cell data and the ground truth of the constructed bulk data was calculated to evaluate and compare the four candidate algorithms, namely Bulk2Space (β -VAE), GAN, CGAN, and bMIND. The results are shown in Figure 1. The same pattern in *Pearson* correlation, *Spearman* correlation, and root mean squared error (RMSE) was confirmed.

Bulk2Space outperformed GAN and CGAN by the *Pearson* correlation, *Spearman* correlation, and RMSE of cell-type-specific marker gene expression with paired simulation data (Figure 1a).

Because the bMIND algorithm is not applicable to paired simulation data, thus we compared the four methods with unpaired simulation data. As shown in Figure 1b, the same pattern was also obtained. Bulk2Space exhibited higher *Pearson* and *Spearman* correlation and lower RMSE than the other three methods. In addition, we have previously compared the cell-type-specific marker gene correlations of these four methods in the *Supplementary Information* (Fig. S3b, Pages 8-9).

Figure 1. Performance evaluation of four methods using paired and unpaired simulation data. **a**, Expression correlation and variation of cell-type-specific marker genes (*Pearson* correlations, left; *Spearman* correlations, middle; and RMSE, right) were calculated for Bulk2Space (β -VAE), GAN, and CGAN, using paired simulation data. Each point represented a simulated dataset. **b**, Expression correlation and variation of cell-type-specific marker genes for Bulk2Space (β -VAE), GAN, CGAN, and bMIND with unpaired simulation data.

Reviewer #2 (Remarks to the Author):

Comments

1. The Abstract provides a better and more accurate description of the contributions of the work. However, the syntax and language is not concise, particularly the last sentence of the Abstract. Please address.

Response: Thanks for your valuable reminder. We have gone through the abstract and addressed the issue immediately. We have modified the last sentence to 'Moreover, Bulk2Space was utilized to perform spatial deconvolution analysis on bulk transcriptome data from two different mouse brain regions derived from our in-house developed sequencing approach termed Spatial-seq. We not only reconstructed the hierarchical structure of the mouse isocortex but also further annotated cell types that were not identified by original methods in the mouse hypothalamus' (Highlighted in blue, Page 2 Lines 39-44).

2. Major issues still exist with the code. There is only one main script, `bulk2space.py`, and it is very lengthy, not well-commented, and does not follow coding best practices in terms of division into composable functions. In general the code organization is very poor – for example there is no dedicating training or testing script; classes and variables are often not named properly (e.g., “myDataset”), etc. This undercuts the quality and technical rigor of the work. With this quality of the code, I have doubts as to the suitability of the work, in terms of its technical quality and polish, for publication in Nature Communications. Additionally, while the provided Tutorials are appreciated, they are not really tutorials as they do not provide step-by-step guidance for the user. At the least, the authors must improve the quality of their scripts and code from a software / package perspective. Relatedly, parameters for running the Bulk2Space model are currently provided as dictionaries; there is no use of .config files. .config files as well as publication of saved model weights would support the reproducibility of the results and the utility of the method. In its current form this work is not ready for publication as a usable package.

Response: Thank you very much for your professional comments and constructive advice, which greatly enhances the quality of the Bulk2Space algorithm. We fully agree with you that we should wrap the code in software or package form, comment the

source code in detail for better interpretation, and provide step-by-step tutorials to fully explain our algorithm.

1) *Re-organization of the source code of Bulk2Space.*

We are very sorry that the original ‘bulk2space.py’ was lengthy, so we devoted a great effort to improving the code quality and technical rigor. As a result, the original main script was separated into four scripts (Figure 2) based on their functions with systematic and explanatory comments (Figure 3) to achieve better practice for users. In addition, we have thoroughly re-organized the Bulk2Space code to provide dedicating training and testing script. In the updated version, the naming of all classes and variables are specified in a canonical manner (Figure 4).

Figure 2. The division of the scripts for Bulk2Space is based on its composable functions. 1) ‘bulk2space.py’, The main script of the Bulk2Space algorithm with all functions encapsulated; 2) ‘utils.py’, The script for data loading and preprocessing; 3) ‘vae.py’, The deep learning model (β -VAE) for training, testing, and generation of single-cell data for the deconvolution step of Bulk2Space; 4) ‘map_utils.py’, The script for simulation, prediction, and processing of the spot-based spatial data for the spatial mapping step of Bulk2Space.

```

8 def load_data(input_bulk_path,
9               input_sc_data_path,
10              input_sc_meta_path,
11              input_st_data_path,
12              input_st_meta_path):
13     input_sc_meta_path = input_sc_meta_path
14     input_sc_data_path = input_sc_data_path
15     input_bulk_path = input_bulk_path
16     input_st_meta_path = input_st_meta_path
17     input_st_data_path = input_st_data_path
18     print("loading data.....")
19     input_data = {}
20     # load sc_meta.csv file, containing two columns of cell name and cell type
21     input_data["input_sc_meta"] = pd.read_csv(input_sc_meta_path, index_col=0)
22     # load sc_data.csv file, containing gene expression of each cell
23     input_sc_data = pd.read_csv(input_sc_data_path, index_col=0)
24     input_data["sc_gene"] = input_sc_data._stat_axis.values.tolist()
25     # load bulk.csv file, containing one column of gene expression in bulk
26     input_bulk = pd.read_csv(input_bulk_path, index_col=0)
27     input_data["bulk_gene"] = input_bulk._stat_axis.values.tolist()
28     # filter overlapping genes.
29     input_data["intersect_gene"] = list(set(input_data["sc_gene"]).intersection(set(input_data["bulk_gene"])))
30     input_data["input_sc_data"] = input_sc_data.loc[input_data["intersect_gene"]]
31     input_data["input_bulk"] = input_bulk.loc[input_data["intersect_gene"]]
32     # load st_meta.csv and st_data.csv, containing coordinates and gene expression of each spot respectively.
33     input_data["input_st_meta"] = pd.read_csv(input_st_meta_path, index_col=0)
34     input_data["input_st_data"] = pd.read_csv(input_st_data_path, index_col=0)
35     print("load data done!")
36
37     return input_data

```

Figure 3. An example to illustrate the detailed comments for Bulk2Space. The comments in the updated version of Bulk2Space are detailed and informative.

Parameter description

- Decompose bulk transcriptomics data into single-cell transcriptomics data:

```

from bulk2space import Bulk2Space
model = Bulk2Space()

# Decompose bulk transcriptomics data into single-cell transcriptomics data
generate_sc_meta, generate_sc_data = model.train_vae_and_generate(
    input_bulk_path,
    input_sc_data_path,
    input_sc_meta_path,
    input_st_data_path,
    input_st_meta_path,
    ratio_num=1,
    top_marker_num=500,
    gpu=0,
    batch_size=512,
    learning_rate=1e-4,
    hidden_size=256,
    epoch_num=5000,
    vae_save_dir='save_model',
    vae_save_name='vae',
    generate_save_dir='output',
    generate_save_name='output')

```

Parameter	Description	Default Value
input_bulk_path	Path to bulk-seq data files (.csv)	None
input_sc_data_path	Path to scRNA-seq data files (.csv)	None
input_sc_meta_path	Path to scRNA-seq annotation files (.csv)	None
input_st_data_path	Path to ST data files (.csv)	None
input_st_meta_path	Path to ST metadata files (.csv)	None
ratio_num	The multiples of the number of cells of generated scRNA-seq data	(int) 1
top_marker_num	The number of marker genes of each celltype used	(int) 500
gpu	The GPU ID. Use cpu if <code>--gpu < 0</code>	(int) 0
batch_size	The batch size for β -VAE model training	(int) 512
learning_rate	The learning rate for β -VAE model training	(float) 0.0001
hidden_size	The hidden size of β -VAE model	(int) 256
epoch_num	The epoch number for β -VAE model training	(int) 5000
vae_save_dir	Path to save the trained β -VAE model	(str) save_model
vae_save_name	File name of the trained β -VAE model	(str) vae
generate_save_dir	Path to save the generated scRNA-seq data	(str) output
generate_save_name	File name of the generated scRNA-seq data	(str) output

Figure 4. An example to illustrate the naming of all classes and variables. In the revised version, the meanings are easier to understand based on the names of classes, functions, and variables.

2) Encapsulation of the Bulk2Space code as a package.

The original Bulk2Space code was not organized so well that it needed to be run command by command. In the updated version, as shown in Figure 5, the code is packaged so that you can call training, testing, simulation, prediction, and other functions directly by importing the bulk2space package ('import bulk2space'). We sincerely appreciate your suggestions on improving code quality, which can in return help Bulk2Space to be used and promoted by more users. Therefore, we have wrapped the code and added more comments in this version from a software or package perspective.

Requirements and Installation

Dependencies: deepforest 0.4.9, numpy 1.19.2, pandas 1.1.3, scikit-learn 1.0.1, scipy 1.5.2, scanpy 1.8.1, easydict 1.9, scdm 4.50.2, Livecode 1.3.0

Create and activate Python environment

For Bulk2Space, the python version need is over 3.8. If you have installed Python3.6 or Python3.7, consider installing Anaconda, and then you can create a new environment.

```
conda create -n bulk2space python=3.8
conda activate bulk2space
```

Install pytorch

The version of pytorch should be suitable to the CUDA version of your machine. You can find the appropriate version on the [PyTorch website](https://download.pytorch.org/whl/cu116). Here is an example with CUDA11.6:

```
pip install torch --extra-index-url https://download.pytorch.org/whl/cu116
```

Install other requirements

```
cd bulk2space-main
pip install -r requirements.txt
```

Install Bulk2Space

```
python setup.py build
python setup.py install
```

Python Code:

```
import scanpy
import pandas as pd
import numpy as np
import matplotlib.colors as clr
import matplotlib.pyplot as plt
from scipy.stats import pearsonr

import bulk2space
from bulk2space import Bulk2Space
model = Bulk2Space()

# load input sc data
input_data = bulk2space.utils.load_data(
    input_bulk_path='tutorial/data/hypothalamus/lcm_bulk.csv',
    input_sc_data_path='tutorial/data/hypothalamus/lcm_sc_dt.csv',
    input_sc_meta_path='tutorial/data/hypothalamus/lcm_sc_ct.csv',
    input_st_data_path='tutorial/data/hypothalamus/lcm_st_data.csv',
    input_st_meta_path='tutorial/data/hypothalamus/lcm_st_meta.csv'
)

df_meta, df_data = model.spatial_mapping(
    generate_sc_meta,
    generate_sc_data,
    input_st_data_path='tutorial/data/hypothalamus/lcm_st_data.csv',
    input_st_meta_path='tutorial/data/hypothalamus/lcm_st_meta.csv')
```

Figure 5. Illustration of installing (left) and importing (right) bulk2space.

3) Providing step-by-step tutorials.

We appreciate your constructive suggestion. Therefore, we have updated the Bulk2Space to version 1.0.0, and provided a detailed tutorial handbook (<https://github.com/ZJUFanLab/bulk2space/blob/main/tutorial/handbook.md>) to use the Bulk2Space, including the format of input files (bulk, single-cell, and spatial datasets) and description of parameters in the algorithm (Figure 6).

Tutorial Handbook

Input data format

Bulk2Space requires five formatted data as input:

1. Bulk-seq Normalized Data: a `.csv` file with genes as rows and one sample as column

	Sample
Gene1	5.22
Gene2	3.67
...	...
GeneN	15.76

2. Single Cell RNA-seq Normalized Data: a `.csv` file with genes as rows and cells as columns

	Cell1	Cell2	Cell3	...	CellN
Gene1	1.05	2.31	1.72	...	0
Gene2	4.71	1.07	0	...	4.22
...
GeneN	0.55	0	1.48	...	0

3. Single Cell RNA-seq Annotation Data: a `.csv` file with cell ID and celltype annotation columns.

- o The column containing cell ID should be named `Cell`
- o the column containing the labels should be named `Cell_type`

	Cell	Cell_type
Cell1	Cell1	T cell
Cell2	Cell2	B cell
...
CellN	CellN	Monocyte

4. Spatial Transcriptomics Normalized Data: a `.csv` file with genes as rows and cells (or spots) as columns

	Cell1 / Spot1	Cell2 / Spot2	...	CellN / SpotN
Gene1	3.22	4.71	...	1.01
Gene2	0	2.17	...	2.20
...
GeneN	0	0.11	...	1.61

5. Spatial Transcriptomics Coordinates Data: a `.csv` with cell/spot ID and coordinates columns.

- o The column containing the coordinates should be named `xcoord` and `ycoord`
- o For spot-based data, the column containing spot ID should be named `Spot`
- o For image-based data, the column containing cell ID should be named `Cell`

	Spot (or Cell)	xcoord	ycoord
Cell_1 / Spot_1	Cell_1 / Spot_1	1.2	5.2
Cell_2 / Spot_2	Cell_1 / Spot_1	5.4	4.3
...
Cell_n / Spot_n	Cell_1 / Spot_1	11.3	6.3

Figure 6. Screenshots of the handbook of Bulk2Space.

In addition, we have provided step-by-step guidance for users in the updated tutorials to ensure that Bulk2Space works without obstacles on analyzing the bulk RNA-seq data (<https://github.com/ZJUFanLab/bulk2space/tree/main/tutorial>). The descriptive comments are also included to make it easy for users to understand the rationality of the parameter selection in each step (Figure 7).

Demonstration of Bulk2Space on demo1 dataset

Import Bulk2Space

```
In [1]: from bulk2space import Bulk2Space
model = Bulk2Space()
```

Decompose bulk-seq data into scRNA-seq data

Train β -VAE model to generate scRNA-seq data

```
In [2]: generate_sc_meta, generate_sc_data = model.train_vae_and_generate(
input_bulk_path='tutorial/data/example_data/demo1/demo1_bulk.csv',
input_sc_data_path='tutorial/data/example_data/demo1/demo1_sc_data.csv',
input_sc_meta_path='tutorial/data/example_data/demo1/demo1_sc_meta.csv',
input_st_data_path='tutorial/data/example_data/demo1/demo1_st_data.csv',
input_st_meta_path='tutorial/data/example_data/demo1/demo1_st_meta.csv',
ratio_num=1,
top_marker_num=500,
gpu=0,
batch_size=512,
learning_rate=1e-4,
hidden_size=256,
epoch_num=20,
vae_save_dir='tutorial/data/example_data/demo1/predata/save_model',
vae_save_name='demo1_vae',
generate_save_dir='tutorial/data/example_data/demo1/predata/output',
generate_save_name='demo1')

loading data.....
load data done!
begin vae training...
Train Epoch: 19: 100% | 20/20 [00:01<00:00, 15.25tit/s, loss=0.5252, min_lo
min loss = 0.5252203941345215
vae training done!
generating: 100% | 249/249.0 [00:00<00:00, 12262.75tit/s]
save trained vae in tutorial/data/example_data/demo1/predata/save_model/demo1_vae.pth.
generating...
generated done!
saving to tutorial/data/example_data/demo1/predata/output/demo1_celltype_pred.csv and tutorial/data/example_data/demo1/predata/output/demo1_data_pred.csv.
```

```
In [3]: generate_sc_meta
```

```
Out [3]:
```

	Cell	Cell_type
0	C_1	T cell
1	C_2	Dendritic cell
2	C_3	Monocyte
3	C_4	T cell
4	C_5	T cell
...
244	C_245	Dendritic cell
245	C_246	Dendritic cell
246	C_247	Dendritic cell
247	C_248	Dendritic cell
248	C_249	Dendritic cell

249 rows x 2 columns

```
In [4]: generate_sc_data
```

```
Out [4]:
```

	C_1	C_2	C_3	C_4	C_5	C_6	C_7	C_8	C_9	C_10	...	C_240	C_241	C_242	C_243	C_249
AP2B1	0.626690	0.613905	0.713269	0.560173	0.669750	0.590490	0.650834	0.531286	0.680752	0.544410	...	0.546440	0.510871	0.619274	0.640492	0.641
ZCCHC10	0.303225	0.286319	0.403583	0.225366	0.358619	0.394472	0.405909	0.260239	0.429008	0.250870	...	0.340657	0.292923	0.313093	0.289614	0.36
ARF3	0.528737	0.505512	0.564077	0.443419	0.434200	0.591801	0.448409	0.476640	0.524426	0.486490	...	0.536649	0.435973	0.506175	0.491610	0.58
ELOA	0.114385	0.158002	0.205178	0.122117	0.243502	0.107239	0.199565	0.116714	0.168558	0.185430	...	0.138138	0.111177	0.192235	0.204547	0.16
TNRC6C	0.000000	0.000000	0.000000	0.000000	0.000000	0.000000	0.000000	0.000000	0.000000	0.000000	...	0.000000	0.000000	0.000000	0.000000	0.00
...
SLC25A24	0.506901	0.664801	0.735292	0.502766	0.558032	0.590491	0.610373	0.525088	0.850940	0.512380	...	0.614196	0.544365	0.670417	0.585746	0.75
INTS13	0.065622	0.000000	0.082370	0.072101	0.026869	0.034036	0.070905	0.000000	0.063085	0.016441	...	0.082803	0.075601	0.062031	0.084564	0.00
TIMM17A	0.000000	0.000000	0.000000	0.000000	0.000000	0.000000	0.000000	0.018179	0.000000	0.000000	...	0.040960	0.110996	0.068087	0.000000	0.00
PSMC1	0.306773	0.298457	0.408020	0.217292	0.305006	0.387245	0.435506	0.268307	0.421335	0.280133	...	0.255140	0.390590	0.282728	0.367992	0.46
TAF2	0.111933	0.086029	0.189509	0.139594	0.107850	0.000000	0.023603	0.033048	0.071503	0.062142	...	0.016894	0.009459	0.013466	0.134478	0.02

6588 rows x 249 columns

Figure 7. Demonstration of Bulk2Space on demo 1 datasets

Furthermore, since many researchers require the use of Bulk2Space via email and other social media, there will be continuous and long-term updates, and the parameters for running Bulk2Space are comprehensively described on the handbook. The modified tutorial is implemented step by step, and users can always view the parameter selection and intermediate variables along the analysis (Figure 8). We hope the revised source code and tutorials will address your concerns and satisfy you.

Bulk2Space integrates spatial gene expression and histomorphology in pancreatic ductal adenocarcinoma (PDAC)

In this tutorial, we will show you an application of Bulk2Space to integrate spatial gene expression and histomorphology in pancreatic ductal adenocarcinoma (PDAC). The processed PDAC bulk-seq data (GSE171485) by Wu et.al., the PDAC scRNA-seq data and ST data (GSE111672) by Moncada et.al. can be found here.

Figure 8. Screenshots of the step-by-step tutorial for the analysis of PDAC data.

3. Please add explicit description of the computing environments (e.g., GPUs used for training) used into a section within the Methods.

Response: Thanks for your valuable suggestion, we fully agree with you. We have

revised the Methods section and included the detailed computing environments (Highlighted in blue, Page 25 Lines 633-645). The computing environments are listed as follows.

Computing Environment #1

Workstation 1: Dell Precision Tower 7820 Workstation, CPU (Intel Xeon Gold 5118, 2.3 GHz × 2), RAM (64 GB, 16 GB × 4, DDR4, 2933 MHz), Hard Drive (SSD, SATA Class 20, 512 GB; HDD, 7200 rpm, SATA), Graphics Card (NVIDIA, Quadro P4000, 8 GB), Operating System (Ubuntu 16.04), Running Environment (CUDA 11.6, Torch 1.12.1, Python 3.8.5, deep-forest 0.1.5, easydict 1.9, numpy 1.19.2, pandas 1.1.3, scanpy 1.8.1, scikit-learn 1.0.1, scipy 1.5.2, tqdm 4.50.2, Unidecode 1.3.0).

Computing Environment #2

Workstation 2: Dell Precision Tower 7920 Workstation, CPU (Intel Xeon Gold 6230, 2.1 GHz × 2), RAM (192 GB, 16 GB × 12, DDR4, 2933 MHz), Hard Drive (SSD, SATA Class 20, 512 GB; HDD, 7200 rpm, SATA), Graphics Card (NVIDIA, RTX2080Ti VIDEO CARD V2 × 2, 22 GB), Operating System (Ubuntu 18.04), Running Environment (CUDA 11.0, Torch 1.7.1, Python 3.8.5, deep-forest 0.1.5, easydict 1.9, numpy 1.19.2, pandas 1.1.3, scanpy 1.8.1, scikit-learn 1.0.1, scipy 1.5.2, tqdm 4.50.2, Unidecode 1.3.0).

4. Please provide descriptive titles for each of the Supplementary Data files. The main Supplementary Information should also be labeled / titled accordingly.

Response: Thanks for your suggestion, we fully agree with you. In the revised version, we have included a detailed description in the *Supplementary Information* for the *Supplementary Data* files (Highlighted in blue, Page 3). Moreover, we have added a table of contents to guide readers (Highlighted in blue, Pages 1-2) in the *Supplementary Information* file.

Reviewer #3 (Remarks to the Author):

Minor comments

1. Bulk2Space was successfully ran on PDAC data however this was not easy with the raw hypothalamus data, please provide detailed instructions on how to use that data (i.e. to produce the bulk.csv, sc_data.csv etc.).

Response: Thanks for your positive comments and valuable advice, which extensively improved the quality of the manuscript. According to your suggestion, we have updated the Bulk2Space to version 1.0.0, and provided a detailed tutorial handbook (<https://github.com/ZJUFanLab/bulk2space/blob/main/tutorial/handbook.md>) to use the Bulk2Space, including the format of input files (bulk, single-cell, and spatial datasets) and description of parameters in the algorithm.

In addition, we provided a step-by-step tutorial on analyzing the hypothalamic data (<https://github.com/ZJUFanLab/bulk2space/blob/main/tutorial/hypothalamus.ipynb>) so that you can follow the process to reproduce the results. The bulk hypothalamus data of the mouse brain used in this study can be downloaded from Google Drive (https://drive.google.com/file/d/1ZGstNzVX-YxofrPP8ZVmr0Zu4nd_O_bZ/view).

Should you have any questions, please feel free to leave a comment on the GitHub website (<https://github.com/ZJUFanLab/bulk2space/issues>).

2. Page 10 of comments to reviewers under section 'Paired and unpaired simulations for evaluation of the spatial mapping step', for simulating pseudo spatial transcriptomics data 100-1000 spot numbers were used. The upper boundary seems low as Visium (which is specifically referenced in the paper) has up to 5000 spots per capture area. I suggest for this analysis to be extended to 5000 spots.

Response: Thanks for your constructive suggestion and we fully agree with you that the analysis could be extended to 5000 spots. Therefore, in the revised manuscript, we have evaluated the performance of Bulk2Space using simulated spatial data with 5000 spots and obtained same patterns on benchmark test (Figure 9). The results were included in Fig. 2 in the manuscript (Page 11).

Figure 9. Performance evaluation of Bulk2Space and other methods with 100, 200, 500, 1000, and 5000 spots using paired and unpaired simulation data.

REVIEWERS' COMMENTS

Reviewer #2 (Remarks to the Author):

Thank you for carefully addressing all reviewers' concerns. I have no further comments.

Response to reviewers

REVIEWERS' COMMENTS

Reviewer #2 (Remarks to the Author):

Thank you for carefully addressing all reviewers' concerns. I have no further comments.

Response: Thanks for your positive comment.